# REPRESENTATIONAL KNOWLEDGE DISTILLATION ACROSS WEARABLE BIOSIGNALS

## ABSTRACT

Modern wearable devices can conveniently and continuously record various biosignals in the many different environments of daily living, ultimately enabling a rich view of individual health. However, not all biosignals are the same: high-fidelity measurements, such as photoplethysmography (PPG), contain more physiological information, but require optical sensors with a high power footprint. In a resource-constrained setting, such high-fidelity biosignals may be unavailable. Alternatively, a lower-fidelity biosignal, such as those from an accelerometer, has a significantly smaller power footprint and is available in almost any wearable device. While multi-modal modeling and cross-modal reconstruction of biosignals have been done before, here, we demonstrate that we can distill representational knowledge across biosignals with different levels of fidelity, i.e., from PPG to accelerometer, using 20 million minutes of unlabeled data collected from ∼172K participants in the Apple Heart and Movement Study under informed consent. Our knowledge distillation framework does not require labels; we pre-train PPG encoders via self-supervised learning, and then distill the representational knowledge from the PPG encoders to accelerometer encoders. We first demonstrate strong cross-modal alignment on unseen data, e.g., $99.2\%$ top-1 accuracy for retrieving PPG embeddings from accelerometer embeddings. We show that distilled accelerometer encoders have significantly more informative representations compared to self-supervised or supervised encoders trained on accelerometer data for downstream targets, observed by at least 23%-49% improved performance for predicting heart rate and heart rate variability, and are readily predictive of a wide array of downstream targets including demographic variables, health conditions, use of medications, and lifestyle habits. We also demonstrate that our framework can be applied to different encoder architectures with different pre-training strategies of the strong encoder, and can be used to simultaneously do cross-modality distillation and model compression. Additionally, we perform various ablations for augmentations, hyperparameters and multi-modal training. We believe our proposed representational knowledge distillation framework may unlock new opportunities for developing digital biomarkers from any wearable device with lower-fidelity biosignals, and help individuals track their health more frequently and conveniently.

## 1 INTRODUCTION

The recent growth of wearable devices has empowered individuals to track their health more conveniently and frequently. Wearable devices can record various biosignals and provide individuals with a continuous view of their health metrics in daily environments, a view which is hard to achieve in clinical settings. While this unlocks new exciting opportunities for wearable devices, challenges remain. High-fidelity biosignals that have substantial physiological information content tend to require specific hardware and sensors for recording, therefore making them unavailable in some wearable devices. Even when available, they may be difficult to collect frequently due to power constraints of small wearable devices, or they may require active user engagement. An example of such is photoplethysmography (PPG), which measures volumetric changes in arterial blood flow and contains diverse physiological information but requires power-consuming optical sensors for recording. On the other hand, lower-fidelity biosignals, such as those from an accelerometer, while capable of detecting certain minute physiological movements (Inan et al., 2015; Kim et al., 2016), are

known to be more susceptible to motion artifacts and thus more challenging to extract physiological information from, compared to PPG. However, they are available on almost all wearable devices and have a significantly smaller power footprint. Independently, another challenge for developing digital biomarkers for wearable devices is the lack of large labeled medical datasets because they are much more expensive and time-consuming to collect unlike in other domains. Given these challenges, the question that remains is whether we can train powerful deep neural networks for lower-fidelity biosignals, such as the accelerometer, without requiring labels and by leveraging information in high-fidelity biosignals, such as PPG. In this work, we aim to address this question.

Unsupervised representation learning techniques, such as self-supervised learning, have been proven successful in training deep encoders without requiring any explicit labels during training in various domains of deep learning such as natural language processing (Devlin et al., 2019; OpenAI, 2023), computer vision (Chen et al., 2020; 2021; Oquab et al., 2023), and speech recognition (Baevski et al., 2020; 2022). In addition, there has been several works for training encoders for health and wellness applications using biosignals (Cheng et al., 2020; Kostas et al., 2021; Sarkar & Etemad, 2022; Mohsenvand et al., 2020; Gopal et al., 2021; Kiyasseh et al., 2021; Mehari & Strodthoff, 2022; Wu et al., 2020; Spathis et al., 2021; Tang et al., 2021; Yuan et al., 2023; Lai et al., 2023; Abbaspourazad et al., 2024; Liu et al., 2024). Independent to representation learning, knowledge distillation has been extensively used for transferring knowledge from a neural network (teacher) to another neural network (student) (Hinton et al., 2015), which can also unlock cross-modal distillation (Gupta et al., 2015; Aytar et al., 2016; Tian et al., 2022). While knowledge distillation has been traditionally often used in supervised settings (Hinton et al., 2015; Gupta et al., 2015; Aytar et al., 2016; Tian et al., 2022), there is a growing interest in combining it with self-supervised learning via self-distillation (Xie et al., 2020; Fang et al., 2021; Caron et al., 2021) or distillation from multiple foundation models into one to improve and agglomerate their representations (Wei et al., 2022; Wang et al., 2024; Ranzinger et al., 2024). See Section 2 for how our work relates to the existing work.

In this paper, using unlabeled sensor data collected under informed consent from the large longitudinal Apple Heart and Movement Study (AHMS) (MacRae, 2021; Truslow et al., 2024), we demonstrate cross-modal representational knowledge distillation to train accelerometer encoders distilled from PPG teacher encoders. Our contributions are: 1) **Representational knowledge distillation across wearable biosignals in a large-scale dataset**: While multi-modal modeling and cross-modal reconstruction of biosignals have been done before, we study representational knowledge distillation across wearable biosignals using a dataset at such large scale with 20M minutes of multi-modal sensor data from ∼172K participants. 2) **Fully unsupervised representational knowledge distillation framework**: We combine and adopt techniques inspired by uni-modal and multi-modal pre-training frameworks from other domains of deep learning to create a fully unsupervised distillation framework for biosignal time-series, which is particularly crucial for health applications where labeled data is limited. 3) **Studying representational alignment**: We study representational alignment of PPG and accelerometer encoders, by doing retrieval analysis. 4) **Studying representational information**: We study the representational power of PPG and accelerometer encoders for critical health targets such as heart rate and heart rate variability, across different available labeled data regimes, as well as for demographic variables and 46 health targets including health conditions, use of medications and lifestyle habits from AHMS survey questions. In addition to these contributions, to support the robustness of the findings, we perform ablation studies to evaluate the efficacy of the encoder architecture (Transformer and EfficientNet), the pre-training strategy of the PPG teacher model (masked autoencoding and contrastive learning), the augmentations, and other training choices.

## 2 RELATED WORK

**Uni-modal and multi-modal representation learning**: Unsupervised representation learning techniques, e.g., self-supervised learning, have been proven successful in training generalist models, also known as foundation models, without requiring any explicit labels during training in various domains of deep learning such as natural language processing (Devlin et al., 2019; OpenAI, 2023), computer vision (Chen et al., 2020; 2021; He et al., 2021; Oquab et al., 2023), speech recognition (Baevski et al., 2020; 2022), and health (Cheng et al., 2020; Kostas et al., 2021; Sarkar & Etemad, 2022; Mohsenvand et al., 2020; Gopal et al., 2021; Kiyasseh et al., 2021; Mehari & Strodthoff, 2022; Wu et al., 2020; Spathis et al., 2021; Tang et al., 2021; Yuan et al., 2023; Lai et al., 2023; Abbaspourazad et al., 2024; Liu et al., 2024). While most of these works have been primarily on

training uni-modal foundation models, there has been a recent shift in training multi-modal foundation models to allow for leveraging information from multiple modalities, either to train a model that simultaneously processes multiple modalities (Mizrahi et al., 2023; Meta, 2024), or to train and bind multiple modality-specific foundation models (Radford et al., 2021; Girdhar et al., 2023; Thapa et al., 2024), particularly with a contrastive objective. Similarly for health applications, there has been a growing interest in cross-modal reconstruction of biosignals (Sarkar & Etemad, 2020), or modeling and pre-training multiple biosignal modalities simultaneously (Deldari et al., 2022; Liu et al., 2024; Deldari et al., 2024; Thapa et al., 2024).

**Knowledge distillation**: Knowledge distillation has been extensively used for transferring knowledge from a neural network (teacher) to another neural network (student) in other domains (Hinton et al., 2015; Tian et al., 2022), traditionally often in supervised settings to transfer knowledge from a large neural network to a small neural network (Hinton et al., 2015; Tian et al., 2022) or from a high-fidelity modality to a low-fidelity modality (Gupta et al., 2015; Aytar et al., 2016; Tian et al., 2022), or from an ensemble network into a single one (Tian et al., 2022), by using the teacher's output logits as soft labels of the student model. Alternatively, knowledge distillation can also be performed using intermediate representations. With the recent emergence of foundation models, there has been several works for combining self-supervised learning and knowledge distillation via self-distillation (Xie et al., 2020; Fang et al., 2021; Caron et al., 2021), or distilling one or several existing foundation models to a single foundation model to improve and agglomerate their representations (Wei et al., 2022; Wang et al., 2024; Ranzinger et al., 2024).

In this work, we adopt ideas from different related work in uni-modal and multi-modal representation learning and knowledge distillation, for biosignal time-series. Similar to uni-modal foundation models, we train our PPG teacher encoder using masked autoencoding (He et al., 2021) and contrastive learning (Chen et al., 2020), with our own variation of these frameworks. We then transfer knowledge from our PPG teacher encoders to accelerometer student encoders via a cross-modal knowledge distillation framework (Tian et al., 2022) that bears similarity to multi-modal contrastive learning (Radford et al., 2021; Girdhar et al., 2023) and self-distillation for model compression (Fang et al., 2021). There has been prior work on training uni-modal foundation models for PPG and accelerometer (Spathis et al., 2021; Yuan et al., 2023; Abbaspourazad et al., 2024) or multi-modal foundation models for other biosignals (Deldari et al., 2022; Liu et al., 2024; Deldari et al., 2024; Thapa et al., 2024). These multi-modal biosignal foundation models either take multiple modalities as input (Liu et al., 2024; Deldari et al., 2024); or they "bind" multi-modality embeddings into a shared subspace (Deldari et al., 2022; Thapa et al., 2024) similar to (Radford et al., 2021) from randomly-initialized encoders, which is primarily different for our motivation of cross-modal representational knowledge distillation and model compression from an existing high-fidelity teacher encoder. In fact, we show that training both encoders of high-fidelity and low-fidelity modalities together degrades the quality of low-fidelity embeddings (see Section 5.5). In line with our motivation, there has been prior work on leveraging asymmetric information in biosignals, by cross-modal reconstruction of electrocardiogram (ECG) from PPG, for a more accurate estimation of heart rate (Sarkar & Etemad, 2020). Also, our work focuses on accelerometer during low-motion periods where it captures minute cardiovascular signals such as the ballistocardiogram, which is distinct from modeling slower changes in accelerometer during gross motion for health and fitness (Hallgrímsson et al., 2018; Ni et al., 2019; Spathis et al., 2021). All in all, while multi-modal modeling and cross-modal reconstruction of biosignals have been done before, our work studies unsupervised representational knowledge distillation across wearable biosignals using large-scale data collected from wearable consumer devices, to develop a single encoder for accelerometer that is readily predictive of a wide array of downstream health targets.

# 3 METHODS AND IMPLEMENTATION DETAILS

Our representational knowledge distillation framework is fully unsupervised and consists of two steps: teacher pre-training and cross-modal representational knowledge distillation as depicted in Figure 1. Below, we go over the details of each of these steps as well as their implementation details.

## 3.1 TEACHER PRE-TRAINING

The first step of our representational knowledge distillation framework is to pre-train the PPG teacher encoder via self-supervised learning without any labels. To show that our knowledge distillation

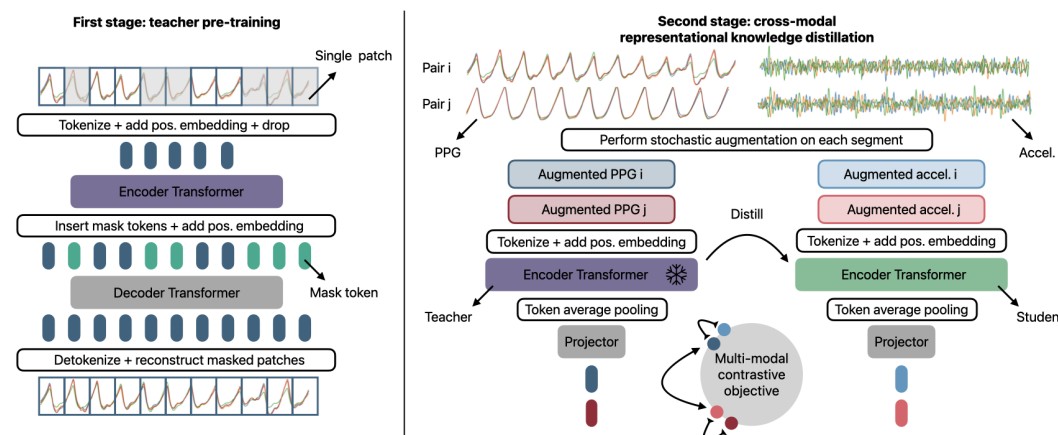

Figure 1: Overview of our representational knowledge distillation framework, in two major steps. **Teacher pre-training**: we first pre-train an encoder for the high-fidelity biosignal, PPG, using masked autoencoding, this step can be alternated with other self-supervised learning techniques, such as contrastive learning, for which we do an ablation in the text. We turn the multi-channel PPG input into tokens from non-overlapping sequential patches. The encoder Transformer processes only a small proportion of the input tokens (20%), and the decoder Transformer takes the encoder's output tokens interleaved with learnable mask tokens to reconstruct the PPG signal at masked patches. **Cross-modal representational knowledge distillation**: we then distill the representational knowledge from the PPG teacher encoder to an accelerometer student encoder using a paired PPG-accelerometer dataset. In each mini-batch of data, we first augment PPG and accelerometer signals independently, the teacher (frozen) and student encoders process these augmented views into embeddings, which are then down-projected to a lower-dimensional subspace. The down-projected embeddings are then aligned using a multi-modal contrastive loss.

framework is agnostic to the pre-training of the PPG teacher encoder and study its efficacy, we investigated two popular pre-training strategies: masked autoencoding (MAE) as the main framework and contrastive learning (CL) as an ablation, which we explain in detail below.

**Masked autoencoding**: Our masked autoencoding pre-training framework is adopted from the prior work on images (He et al., 2021), for time-series. We turn the multi-channel PPG input (4-channels and 60s-long, as explained in Section 4.1) into patches using non-overlapping fixed-length windows, and then project the patches with a learnable linear tokenizer into tokens, which results in 192 256-D tokens. Sinusoidal positional embeddings are added to the input tokens, then 80% of input tokens are randomly dropped and the 38 kept tokens are passed through the encoder Transformer to get encoder output tokens. Learnable mask tokens, initialized with a 256-D token drawn from a uniform distribution $\sim U(0, 1)$, are added back to the encoder output tokens at positions where input tokens where dropped, followed by adding sinusoidal positional embeddings. These 192 new tokens are then processed by the decoder Transformer to generate the decoder output tokens. Finally, with a linear projection, the decoder output tokens are projected to the multi-channel PPG output, with the objective of reconstructing PPG "pixel" values in those indices whose patches/tokens were dropped in the encoder's input. For maximum learning rate, we used 2e-4 and batch size was set to 512. A complete list of other architectural and training hyperparameters that are shared across methods using Transformers can be found in Appendix Table 4. As a baseline for our distilled accelerometer encoders, we also train masked autoencoders for accelerometer in the same way explained above.

**Contrastive learning**: While our main pre-training framework for the teacher encoder is masked autoencoding, we perform ablation in regards to the teacher pre-training method and architecture via contrastive learning with EfficientNets. We would like to emphasize that contrastive learning can also be done with Transformer models without any meaningful difference (Appendix Table 13), but we choose EfficientNets for the main results to simultaneously demonstrate an ablation on changing the teacher pre-training strategy and student/teacher architecture. Our contrastive pre-training framework closely follows a prior work for PPG signals (Abbaspourazad et al., 2024), with the major difference that the positive pairs are selected as two augmented views of the same sample. This choice is made to

enforce the encoders to contain more segment-level information necessary for the main downstream targets used in this study, as opposed to participant-level positive pair selection which was investigated in the prior work (Appendix Table 14). Our stochastic augmentation module consists of a stochastic cascade of several standard individual augmentations (Iwana & Uchida, 2021) including {cut out: 0.4, magnitude warp: 0.25, add Gaussian noise: 0.25, channel permute: 0.25, time warp: 0.15}, where the values are the assigned probabilities of whether the augmentation is applied for each segment or not. During pre-training, we augment each sample twice; the two augmented views are then passed through a joint-embedding architecture, where one encoder is an exponential moving average ($m = 0.99$ for momentum updates) of the other encoder that is updated via backpropagation. The 256-D embeddings of each encoder are then projected to a lower dimensional subspace (128-D) via multi-layer perceptron projection heads. The training objective is maximizing the mutual information of the down-projected embeddings of the two views of the same PPG segment, and minimizing that for different PPG segments, implemented by a regularized InfoNCE loss. We use Kozachenko-Leonenko (KoLeo) differential entropy estimator (Sablayrolles et al., 2019; Oquab et al., 2023) with the weight of 0.1 for regularization, and temperature of 0.04 for scaling similarity scores in the InfoNCE loss. For the maximum learning rate, we used 1e-3, while batch size was set to 256. Other common hyperparameters for training EfficientNets is available in Appendix Table 5. As a baseline for our distilled accelerometer encoders, we also train contrastive learned encoders for accelerometer in the same way explained above.

## 3.2 CROSS-MODAL REPRESENTATIONAL KNOWLEDGE DISTILLATION

After pre-training the PPG encoder in Section 3.1, we distill its representational knowledge to an accelerometer encoder in a dataset of paired PPG-accelerometer segments (Section 4.1). This second stage is also fully unsupervised without requiring any explicit labels. We perform representational knowledge distillation via multi-modal contrastive learning similar to a technique used previously to weakly supervise an image encoder with text (CLIP) (Radford et al., 2021), but here we use it to transfer knowledge. To do this, unlike standard approaches (Radford et al., 2021), we first perform augmentations on both modalities, PPG and accelerometer, using our stochastic augmentation (see Section 3.1), where the augmentations are independently drawn for each modality in a PPG-accelerometer pair. We found that augmentations were crucial for the quality of the embeddings (see Ablation 5.5). The augmented PPG and accelerometer signals are then processed by the PPG teacher encoder (frozen) and accelerometer student encoder, respectively, to get 256-D output embeddings. To calculate the objective, we first down-project the embeddings to 128-D with trainable multi-layer perceptron projection heads (one 1024-D hidden layer) for both the student and teacher encoders. We found separate learnable projection heads to a smaller subspace was necessary to avoid representation collapse. The student encoder is trained to generate embeddings similar to the teacher encoder, where the objective is contrastive and maximizes the mutual information of paired PPG and accelerometer embeddings, while minimizing the mutual information of an accelerometer embedding with other PPG embeddings. For each batch of embeddings $h$ from N positive pairs for student and teacher ($h_t$, $h_s$), we define multi-modal InfoNCE, where the teacher embeddings are selected as anchors:

$$L_{\text{contrastive}}^{(t \to s)} = -\frac{1}{N} \sum_{i=1}^{N} \log \frac{\exp(sim(h_t^i, h_s^i)/\tau)}{\sum_{j=1}^{N} \exp(sim(h_t^i, h_s^j)/\tau)}. \tag{1}$$

Here, $sim(\cdot, \cdot)$ is the cosine similarity function and $L_{\text{contrastive}}^{(t \to s)}$ can be viewed as a $N$-way classification problem ($N$ = batch size), such that $h_s^i$ is the correct pair to $h_t^i$ compared to all other potential pairs in the batch $\{h_s^j | 1 \le j \le N, j \ne i\}$. The final objective is computed as the weighted sum of InfoNCE, from teacher to student and from student to teacher:

$$L = \lambda L_{\text{contrastive}}^{(t \to s)} + (1 - \lambda) L_{\text{contrastive}}^{(s \to t)}, \tag{2}$$

where $\lambda$ is the scalar weight between 0 and 1. In our experiments, the temperature parameter is set to 0.04, and unless otherwise specified we set $\lambda$ to 1, to emphasize more on alignment when PPG embeddings are anchors (see Ablation on $\lambda$ in Section 5.5). For the maximum learning rate, we used 1e-3, while batch size was set to 256. As mentioned above, our main encoders are with Transformers whose common hyperparameters are in Appendix Table 4, and we perform ablations with EfficientNets whose common hyperparameters are in Appendix Table 5.

Table 1: Retrieval analysis for PPG embeddings from accelerometer demonstrates near perfect alignment. Numbers are reported as average (std) across 100 bootstrap candidate pools.

| Embedding | Top-1 Acc. ↑ | Top-3 Acc. ↑ | Top-5 Acc. ↑ | Mean Rank ↓ |
|---|---|---|---|---|
| Accel-KD via PPG-MAE | **99.17 (0.23)** | **99.72 (0.08)** | **99.98 (0.02)** | **1.02 (0.01)** |
| Accel-MAE (+ Procrustes alignment) | 0.18 (0.03) | 0.47 (0.05) | 0.73 (0.06) | 2808.86 (68.95) |
| Chance-level performance | 0.01 | 0.02 | 0.03 | 9551.64 |

## 4 EXPERIMENTS

### 4.1 DATASETS

We used the PPG and accelerometer signals recorded on Apple Watch from participants in the Apple Heart and Movement Study (AHMS) (MacRae, 2021; Truslow et al., 2024). AHMS is an ongoing research study designed to explore the links between physical activity and cardiovascular health, which is sponsored by Apple and conducted in partnership with the American Heart Association and Brigham and Women's Hospital. To be eligible for the Study, participants must be of legal age to provide informed consent (18, 19, or 21 based on location), reside in the United States, have access to an iPhone with the Research app, have an Apple Watch, and provide informed consent within the Research app to participate (Shapiro et al., 2023).

We created a paired PPG-accelerometer pre-training dataset that we used for pre-training uni-modal and distilled models. Apple Watch intermittently and passively records simultaneous green PPG and accelerometer signals during low-motion periods multiple times per day. PPG and accelerometer signals are recorded simultaneously at a 256Hz or 64Hz sampling rate and are 60 seconds in duration. PPG signals consist of four optical channels corresponding to different combinations of transmitting and receiving diodes, and accelerometer signals consist of 3 channels corresponding to 3 spatial dimensions (see Figure 1 for signal examples). We curated a pre-training dataset of paired PPG-accelerometer segments from ~172K participants, where ~20M paired segments were randomly drawn from the full dataset given two conditions: 1) each participant had at least four segments in the pre-training dataset, and 2) the number of segments per participant was as uniform as possible. PPG segments were pre-processed using dark subtraction (to reject signals introduced by ambient light). Both PPG and accelerometer segments were further pre-processed by bandpass filtering, down-sampling to 64Hz if needed, and temporal channel-wise z-scoring for each segment. For PPG teacher pre-training, we use only the PPG segments of the same dataset. Brief statistics of our curated dataset are in Table 6, and AHMS demographics can be found in prior publications (Shapiro et al., 2023; Abbaspourazad et al., 2024; Truslow et al., 2024). The training and evaluation splits were stratified based on participants such that there were no overlapping participants in these two splits.

### 4.2 EVALUATION METRICS

**PPG-accelerometer pair embeddings retrieval:** To assess the quality of the PPG-distilled accelerometer embeddings (256-D representations after the encoder), we perform a retrieval experiment to see how well the accelerometer embeddings can retrieve their corresponding matched PPG segment on unseen test data. For a batch of paired PPG and accelerometer segments on held-out test participants, we compute the cosine similarity between each distilled accelerometer embedding and PPG embedding, which generates a ranked list that can be used for retrieval. We evaluate retrieval quality using top-K accuracy for $K = 1, 3, 5$ (i.e., for a given query accelerometer embedding, how often is its paired PPG embedding in the top $K$ most similar embeddings). We also report the mean rank, i.e., the mean position of the true paired PPG segment in the rankings; smaller values are better, with 1 indicating perfect retrieval as the correct segment is always ranked first. As an ablation, we repeated the same retrieval analysis from PPG to accelerometer embeddings.

**Linear probing for downstream targets:** As our main targets, we perform linear probing for predicting heart rate (HR), and two popular measures of heart rate variability: standard deviation of normal-to-normal intervals (SDNN) and root mean square of successive differences (RMSSD) (Natarajan et al., 2020; Shaffer & Ginsberg, 2017). In addition, we perform linear probing for predicting self-reported age, body mass index (BMI), biological sex, and 46 health targets including

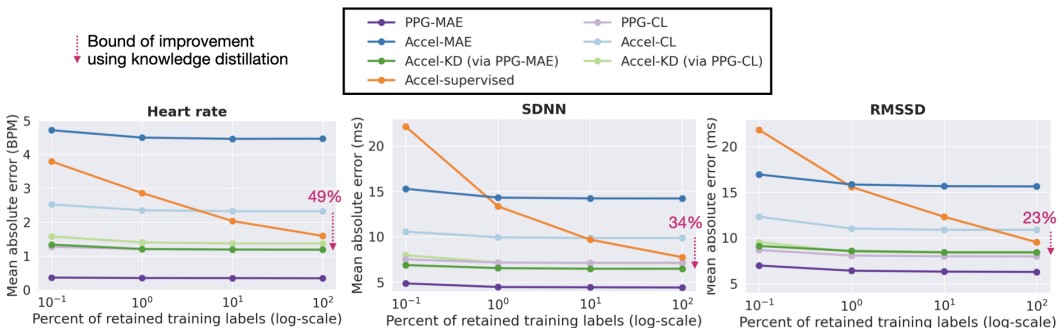

Figure 2: Cross-modal representational knowledge distillation improves the quality of accelerometer embeddings. We compare the representational quality of accelerometer encoders via their downstream prediction of heart rate, SDNN and RMSSD. We sweep the number of training segments/labels for supervised training and linear probing training, from 0.1% to 100% in the x axis. Distilled accelerometer encoders, "Accel-KD (via PPG MAE)" and "Accel-KD (via PPG-CL)", are better than their baseline uni-modal accelerometer encoders, "Accel-MAE" and "Accel-CL", and better than a supervised encoder ("Accel-supervised"). The bound of improvement with the dotted arrows is determined as the difference between the best uni-modal vs. the best distilled accelerometer encoder (Appendix Tables 9, 10 and 11). Compared to the supervised encoder, all pre-trained accelerometer encoders, including distilled ones, demonstrated robustness to the number of available training labels.

health conditions, medication use, and lifestyle habits, for participants in AHMS. Detailed explanation of our linear probing evaluation metrics is explained in Appendix A.2.

## 5 RESULTS

### 5.1 REPRESENTATIONAL KNOWLEDGE DISTILLATION RESULTS IN STRONG ACCELEROMETER-PPG EMBEDDING ALIGNMENT

We first visually inspected the embeddings of 200 random PPG-accelerometer pairs (20 from 10 participants) from our test split, embedded via PPG teacher encoder trained with MAE ("PPG-MAE") and 2 accelerometer encoders: 1) accelerometer encoder trained with MAE ("Accel-MAE"), 2) distilled accelerometer encoder ("Accel-KD via PPG-MAE") from "PPG-MAE", by projecting them into 2D t-sne (Maaten & Hinton, 2008) representation subspace as shown in Appendix Figure 4. We observed a marked difference in the alignment of PPG-accelerometer embeddings for PPG teacher encoder and distilled accelerometer encoder (Appendix Figure 4-right) compared to the uni-modal encoders, visually validating the knowledge transfer from PPG to accelerometer via our representational knowledge distillation framework.

To quantify the quality of the cross-modality embedding alignment, we performed retrieval analysis from accelerometer embeddings to PPG embeddings (Section 4.2), where PPG embeddings are from the PPG teacher encoder ("PPG-MAE") and accelerometer embeddings are from the distilled accelerometer encoder ("Accel-KD via PPG-MAE"). To do this, we sample 1,000 random participants from the ~17.5K participants in the test set (no overlap with training), each with an average of 19 PPG and accelerometer segments, and we repeat this procedure 100 times to report standard deviations of our metrics (Table 1). We observed near-perfect alignment of the PPG and accelerometer embeddings after distillation, indicated by the results in Table 1: we achieve an average 99.17 top-1 accuracy, 99.98 top-5 accuracy, and 1.02 mean rank in random batches of $\sim 19K$ candidates, orders of magnitude better than the baseline encoders. We obtain similarly strong results when repeating this experiment but switching the retrieval task to be instead from PPG embeddings to accelerometer embeddings (see Appendix Table 8). To rule out that this near perfect retrieval performance was simply achievable with uni-modal encoder embeddings, we repeated the retrieval analysis using the uni-modal accelerometer embeddings from "Accel-MAE" by applying optimal translation, rotation and scaling via Procrustes alignment (Krzanowski, 2000) to make them as close as possible to "PPG-MAE" embeddings, and observed marked difference in the retrieval performance (Table 1 and Appendix Table 8). Overall, very high retrieval performance (e.g., 99.17 top-1 accuracy) demonstrates the effectiveness of our

Table 2: Distilled accelerometer encoders are predictive of age, BMI and biological sex with high accuracy, and better than baseline accelerometer encoders. Age and BMI metrics are reported with mean absolute error and biological sex with ROC AUC.

| Encoder | Age (years) ↓ | BMI (kg/m$^2$) ↓ | Biological Sex ↑ |
|---|---|---|---|
| Accel-KD via PPG-MAE | **4.04** | **2.48** | **0.99** |
| Accel-KD via PPG-CL | 4.74 | 2.82 | 0.97 |
| Accel-MAE | 7.73 | 3.84 | 0.87 |
| Accel-CL | 4.96 | 2.62 | 0.98 |

representational knowledge distillation framework and how well the distilled accelerometer encoder embeddings match with PPG teacher encoder embeddings. Importantly, these results indicate that distilled accelerometer embeddings may achieve improved performance for predicting downstream targets due to their high alignment with the high-fidelity PPG embeddings, which we will investigate in the next two sections.

### 5.2 REPRESENTATIONAL KNOWLEDGE DISTILLATION FROM PPG IMPROVES THE QUALITY OF ACCELEROMETER EMBEDDINGS

To gain additional insight about the quality of accelerometer embeddings after knowledge distillation, we compared several encoders in terms of their downstream performance: 1) "Accel-MAE", 2) "Accel-KD via PPG-MAE", and 3) supervised encoder trained directly on each target from scratch ("Accel-supervised"). In the meantime, to study the label efficiency of these models, we sweep the proportion of available training segments/labels for linear probing and supervised training from 0.1% to 100% while keeping the number of test segments/labels the same. Figure 2a represents the performance of these encoders, as well as the performance of the "PPG-MAE" teacher encoder (see Appendix Tables 9, 10 and 11 for extended numbers). We observed that the distilled accelerometer encoder ("Accel-KD via PPG-MAE") not only outperformed "Accel-MAE", but also was better than the supervised encoder in all targets and all label availability regimes, being particularly robust to the amount of training labels (Figure 2a across x axis) as it retained strong performance even with $1000\times$ smaller-sized labeled data. This robustness to the amount of labeled data is a major motivation behind keeping our knowledge distillation framework fully unsupervised.

Additionally, to show that our representational knowledge distillation framework is not unique to the pre-training of the teacher encoder or architecture of the teacher/student encoders, we repeated the pre-training and downstream performance comparison between: 1) accelerometer encoder trained with contrastive learning ("Accel-CL"), 2) distilled accelerometer encoder ("Accel-KD via PPG-CL") from a PPG teacher encoder trained with CL ("PPG-CL"). To simultaneously show robustness to the architecture, in these experiments, we also changed the teacher and student architectures to EfficientNet (Section 3.1). We made similar observations with EfficientNet encoders trained via CL: the distilled accelerometer encoder not only outperformed "Accel-CL" encoder, but also was better than the supervised encoder in all targets and all label availability regimes (Figure 2a; see Appendix Tables 9, 10 and 11 for extended numbers). Interestingly, we made the observation that in uni-modal pre-training, "PPG-MAE" was better than "PPG-CL", while "Accel-CL" was better than "Accel-MAE" (Figure 2a; see Appendix Tables 9, 10 and 11 for extended numbers), where we further discuss in Appendix A.4.

### 5.3 DISTILLED ACCELEROMETER ENCODERS ARE SIGNIFICANTLY MORE PREDICTIVE OF DEMOGRAPHIC VARIABLES AND HEALTH TARGETS

We next questioned whether the distilled accelerometer encoders are predictive of other health-related targets that require capturing waveform information as opposed to pulse timing information that may be sufficient for heart rate and heart rate variability. To this end, we evaluated their downstream prediction performance of a wide array of targets including age, biological sex, BMI and 46 binary health targets derived from AHMS self-reported questionnaires (see Appendix Section A.3). We observed that distilled accelerometer encoders are better predictive of the demographic variables (Table 2) and health targets (Appendix Table 12), compared to the baseline uni-modal accelerometer

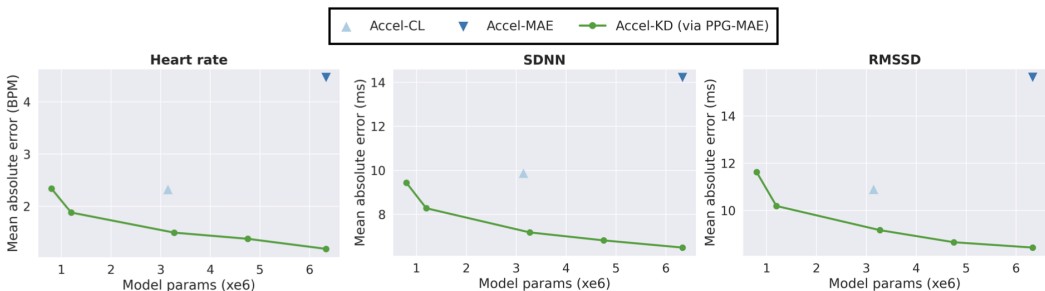

Figure 3: Cross-modal representational knowledge distillation can be used for model compression. We show the downstream prediction of heart rate, SDNN and RMSSD while compressing the distilled accelerometer encoder. We observe that even small accelerometer encoders maintain information and are still even better than the baseline accelerometer encoders, while being $\sim 5\times$ smaller.

encoders. This indicates the generalizability of the distilled accelerometer encoders to a wide range of tasks. To the best of our knowledge, this is the first work demonstrating that a single accelerometer encoder is predictive of demographic variables with remarkably high accuracy (Table 2), and is readily predictive of variety of health targets (Appendix Table 12).

## 5.4 CROSS-MODAL REPRESENTATIONAL KNOWLEDGE DISTILLATION CAN SIMULTANEOUSLY ENABLE MODEL COMPRESSION

One important aspect of models for wearable devices is that they should be compact in size for on-device inference with a minimal power footprint. This allows more frequent estimation of digital biomarkers throughout the day, and makes them available from resource-constrained wearable devices. Therefore, we questioned whether we can perform model compression during the representational knowledge distillation from PPG to accelerometer. To show this, we reduced the size of accelerometer student encoder to 4 new smaller sizes by shrinking the depth and width of the Transformer backbone during the distillation (Figure 3 and Table 7). We observed that our distillation framework robustly maintains the information quantified by downstream performance, even in encoders with significantly smaller sizes. Particularly, our small encoder ("S" in Table 7) is still better than the baseline uni-modal accelerometer encoders ("Accel-MAE" and "Accel-CL"), while having $\sim 5\times$ less parameters.

## 5.5 ADDITIONAL ABLATION STUDIES

Unless otherwise specified, we performed the following ablation studies with "Accel-KD via PPG-MAE" as it is our main method (Figure 1), without loss of generalization given similar qualitative conclusions to "Accel-KD via PPG-CL" (Section 5.2 and Appendix Table 13). We also performed several other ablations presented in Appendix A.3.

**Augmentations:** During our cross-modal knowledge distillation, both pairs of signals, PPG and accelerometer, are augmented with our stochastic cascade augmentation module (Section 3.2). While prior works have investigated augmentations for uni-modal contrastive learning of biosignals, the effect of augmentations for multi-modal knowledge distillation of biosignals remains unknown. Therefore, we investigated the importance of augmentations and observed that they were critical for our cross-modal knowledge distillation as shown in Appendix Table 17. For instance, we observed 45% higher mean absolute error for heart rate prediction when the augmentations were absent during the distillation stage. In addition, we investigated the importance of individual augmentation functions during knowledge distillation (Appendix A.3 and Appendix Table 17).

**Multi-modal pre-training of both PPG and accelerometer encoders simultaneously results in significantly reduced performance:** As discussed in Section 2, there are prior works for multi-modal pre-training of biosignals where they bind different modality embeddings via multi-modal contrastive learning (Deldari et al., 2022; Thapa et al., 2024), which does not involve pre-training a uni-modal teacher encoder (stage 1 in Figure 1). While, our motivation here is different for we use multi-modal contrastive learning to enable unsupervised representational knowledge distillation, we questioned whether we get the same improvement for the accelerometer encoders when binding

Table 3: Ablation on simultaneous PPG-accelerometer multi-modal pre-training as opposed to two-stage representational knowledge distillation.

| Pre-training framework | Eval. ↓ | $\lambda = 0$ | $\lambda = 0.5$ | $\lambda = 1$ |
|---|---|---|---|---|
| Simultaneous multi-modal contrastive learning | Heart rate (BPM) | 2.81 | 2.59 | 2.36 |
| | SDNN (ms) | 11.01 | 10.37 | 9.68 |
| | RMSSD (ms) | 12.55 | 11.90 | 11.30 |
| Cross-modal knowledge distillation (ours, Accel-KD via PPG-MAE) | Heart rate (BPM) | 1.25 | 1.25 | **1.21** |
| | SDNN (ms) | 6.80 | 6.77 | **6.58** |
| | RMSSD (ms) | 8.78 | 8.68 | **8.40** |

PPG and accelerometer embeddings in a multi-modal pre-training setup. Therefore, we did an ablation of multi-modal contrastive learning where both PPG/accelerometer encoders are trainable during training, and to do a fair comparison, we experimented with different $\lambda$ values. For all different $\lambda$ values, we observed that the learned accelerometer encoder had significantly lower quality embeddings (see Table 3) as quantified by a significant drop of performance in all downstream targets (95%, 47% and 35% higher mean absolute error for heart rate, SDNN, and RMSSD, respectively). In addition, to emphasize the importance of freezing the teacher encoder in knowledge distillation, we did the multi-modal training where we initialized the PPG encoder using the pre-trained weights of "PPG-MAE" encoder, and still observed significant degrade in performance (Appendix Table 19). We believe that this is due to the asymmetric amount of information present in the PPG and accelerometer, such that allowing the PPG encoder to update results in more trivial embeddings and degraded performance for the accelerometer encoder. Overall, this observation further demonstrates the importance of two-stage representational knowledge distillation and freezing the teacher encoder to allow maximal knowledge transfer.

# 6 DISCUSSION, LIMITATIONS AND FUTURE WORK

Here, we introduced a fully unsupervised representational knowledge distillation framework to transfer knowledge from high-fidelity to low-fidelity biosignals. We showed that our knowledge distillation framework is not unique to the teacher/student architecture, or teacher pre-training method, and it results in near-perfect alignment of PPG and accelerometer embeddings as quantified by our retrieval analysis. In addition, it improves the quality of the representations in the low-fidelity biosignal compared to self-supervised and supervised baselines, while maintaining label efficiency for downstream targets. We demonstrated that a single accelerometer encoder is predictive of heart rate, heart rate variability, demographic variables and a wide array of downstream health targets. We also showed that it can be used to simultaneously compress the student model while transferring knowledge across modalities. We also performed ablations with respect to several training choices. Our work primarily focused on accelerometer signals during low-motion and sedentary periods (Section 4.1), where accelerometer captures ballistocardiogram and therefore minute cardiovascular-related information (Inan et al., 2015; Kim et al., 2016). Future work can investigate models that take slower-scale activity metrics on wearable devices such as steps, speed, sleep, and slow changes in accelerometer, as well as minute changes in accelerometer at sedentary settings to improve the performance for downstream targets (Hallgrímsson et al., 2018; Ni et al., 2019; Spathis et al., 2021). In addition, another interesting area of investigation for future work could be experimenting with modality specific augmentations (Qian et al., 2022; Demirel & Holz, 2023). One caveat of our work is that it currently supports two modalities, while future work can consider statistical objectives for mutual information maximization, when there are more than one teacher or student modalities (Shidani et al., 2024), or by binding all student modalities to a single teacher modality (Girdhar et al., 2023). Another caveat is that while this work can be used for knowledge transfer or retrieval of the high-fidelity modality embeddings, it does not provide a generative model across modalities (Sarkar & Etemad, 2020); future work can consider recent techniques to incorporate generative capabilities using unified encoder and decoder Transformers (Mizrahi et al., 2023; Meta, 2024). Ideally, future work can also consider modeling other modalities such as text, images and videos to leverage information from these other input sources and weakly supervise biosignal representations, similar to prior work for modeling accelerometer during motion and other modalities such as video and text (Moon et al., 2022; Tan et al., 2023).

REPRODUCIBILITY STATEMENT

The aggregated data that support the findings of this study can be made available on request from the corresponding author. Request for data will be evaluated and responded to in a manner consistent with the specific language in the study protocol and informed consent form (ICF). Similarly, code for all data analyses may be available upon request from the corresponding author. Requests for code will be evaluated and responded to in a manner consistent with policies intended to protect participant confidentiality and language in the study protocol and ICF.

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

# A APPENDIX

## A.1 IMPLEMENTATION DETAILS

The common architectural and training hyperparameters for training Transformer and EfficientNet models, can be found in Tables 4 and 5, respectively. The changes in Transformer model architecture for ablation on compressing the Transformer model size (Section 5.4) can be found in Table 7.

## A.2 DATASET AND EVALUATIONS

**Dataset statistics**: Brief statistics for our curated PPG-accelerometer dataset from AHMS is available in Table 6.

**Linear probing for heart rate and heart rate variability:** We perform linear probing for predicting heart rate (HR), and two popular measures of heart rate variability: standard deviation of normal-to-normal intervals (SDNN) and root mean square of successive differences (RMSSD). These targets are from Apple Watch's generated values during low-motion periods where PPG peaks are reliably detected, resulting in accurate prediction of heart rate and heart rate variability. These targets are chosen because they are widely used in wearable devices (Natarajan et al., 2020) and are indicative of health status (Shaffer & Ginsberg, 2017), training load in athletes (Plews et al., 2013) and stress levels (Kim et al., 2018). Being able to predict them via low-fidelity biosignals will enable a more frequent prediction of such targets, giving the users a broader view of their health-related changes in different scenarios of their daily life. We formulate this problem as a regression task, where we use ridge regression to predict the continuous value of these targets and we use mean absolute error to quantify performance. For heart rate, we report error in beats per minute (BPM), and for SDNN and RMSSD, we report error in milliseconds (ms). For these targets given that they change from segment to segment, we perform the linear probing at segment granularity: each segment contributes one and only one sample in the downstream training/evaluation. However, the downstream training/evaluation splits are stratified based on participants such that the evaluation split's participants have no overlap with those in the training split of the linear probing or pre-training.

**Linear probing for age, BMI, biological sex, and health targets from AHMS survey questions:** We perform linear probing for predicting self-reported age, body mass index (BMI), biological sex (sex assigned at birth), and health targets from survey questions for participants in AHMS. During AHMS, participants fill out multiple questionnaires containing various questions regarding their historical health record and demographics (Truslow et al., 2024; Abbaspourazad et al., 2024). The response to these questions are usually in form of 'yes' or 'no' for whether the participant has had a history of a health condition (e.g., asthma), or whether they take specific medications (e.g., anti-depressants), or regarding their lifestyle habits (e.g., smoking). For the classification tasks, we use ridge regression to predict scores for binarized targets (0/1) and we quantify the performance with area under curve of receiver's operating curve (AUC). For biological sex, we classify male versus female, and for health targets we classify 'yes' versus 'no'. For regression tasks (age and BMI), we use ridge regression to predict continuous targets and we use mean absolute error to quantify performance. Age is reported in years and BMI is reported in $kg/m^2$. In all these subject-related targets that do not vary from segment to segment, we perform the linear probing at participant granularity: we mean-aggregate all the embeddings associated to each participant so that each participant contributes one and only one sample in the downstream training/evaluation. Similar to the heart rate and heart rate variability linear probing, the downstream training/evaluation splits for these targets are stratified based on participants such that the evaluation split's participants have no overlap with those in the training split of the linear probing or pre-training.

AHMS survey is formed of multiple questionnaires which participants fill out over the course of their participation in the study. Tables 20 and 21 contain AHMS survey questions about medical conditions and medications, respectively, in addition to the corresponding target labels used in Appendix Table 12. Table 22 includes AHMS survey questions about drinking and smoking habits, and Table 23 defines our logic to summarize these questions into binary labels for the related targets used in Appendix Table 12.

## A.3 RESULTS AND ABLATION STUDIES

**Visual inspection of the T-SNE representations**: The visual inspection of the T-SNE embeddings for the random, uni-modal and distilled accelerometer encoder as well as the PPG teacher encoder is shown in 4.

**Retrieval analysis**: Retrieval analysis for accelerometer embeddings from PPG embeddings is shown in Table 8.

**Extended numbers for linear probing of heart rate, SDNN and RMSSD**: Linear probing and supervised evaluation performance numbers at 0.1% and 100% data availability in Figure 2, as well as root mean squared error and Pearson's R metrics, can be found in Appendix Tables 9, 10 and 11.

**Ablation on choice of positive pairs in teacher pre-training with contrastive learning**: We selected the positive pairs as two augmented views of the same sample, as opposed to two different segments of the same participants proposed in (Abbaspourazad et al., 2024). This choice was made to enforce the encoders to contain more segment-level information necessary for the main downstream targets used in this study (heart rate and heart rate variability). Appendix Table 14 demonstrates the performance of heart rate, SDNN and RMSSD for the "PPG-CL" trained with two different positive pair selection strategies.

**Ablation on number of PPG channels in teacher pre-training**: We performed an ablation about the effect of the number of PPG channels in teacher pre-training on downstream evaluations in Table 15, where we only kept one of the PPG channels for modeling and observed a drop in performance.

**Ablation on larger model sizes for teacher pre-training**: We made several optimizations to keep our model sizes small for feasibility on running wearable devices with power and resource constraints. Interestingly, we observed signs of overfitting as we increased our encoder size, which is why our encoder sizes are not larger. This could be due to the fact that one needs to scale the model and data size simultaneously to gain benefits of scaling laws (Kaplan et al., 2020; Zhai et al., 2022). As an example, when we increased the encoder size in "PPG-MAE" (from 6.3M to 12.7M) by increasing the number of layers from 8 to 16, we observed initial signs of overfitting as shown in Table 16. We believe future work can investigate the scaling laws for encoder models of wearable biosignals by growing the encoder and data size simultaneously (Kaplan et al., 2020; Zhai et al., 2022).

**Ablation on augmentations**: In addition to comparing knowledge distillation with and without augmentations (Section 5.5), we studied the importance of individual augmentations during knowledge distillation. To this end, we performed the knowledge distillation from PPG to accelerometer, where we only kept one of the augmentation functions during distillation (applied in every forward pass), one at a time. This was done while maintaining all other training choices the same to control for the effect of augmentations. We observed that: 1) our stochastic augmentation module achieved the highest accuracy (Appendix Table 17), likely because it captures more diverse range of distortions during training, 2) among the individual augmentations, "add Gaussian noise" and "Cut out" had the highest importance, while "Time warp" had the least importance.

In general, our hypothesis for why augmentations are important for knowledge distillation across PPG and accelerometer is that given the relationship between arterial blood flow present in PPG and ballistocardiogram in accelerometer (Inan et al., 2015; Kim et al., 2016), particularly for aligned PPG-accelerometer segments during low-motion periods which is the focus of our work, knowledge distillation without augmentations is a relatively easier pre-training task compared to that with augmentations. Therefore, we think distillation without augmentations, and even very simple augmentations as shown by individual augmentations results in Appendix Table 17, leads to capturing less minute information, which is relatively similar to why and how the amount and type of augmentations in uni-modal contrastive learning is critical as demonstrated in prior work (Chen et al., 2020).

**Ablation on impact of $\lambda$ in the multi-modal contrastive objective** $\lambda$: We studied the impact of $\lambda$ in Equation 2 for our representational knowledge distillation. We observed that while the improvements of accelerometer embeddings via distillation were robust to $\lambda$, higher values of $\lambda$ ($\lambda = \{0.75, 1\}$) were the most optimal (Appendix Table 18), indicating that keeping PPG embeddings as anchor embeddings provided the most knowledge transfer, perhaps due to the fact that PPG is the higher-fidelity modality.

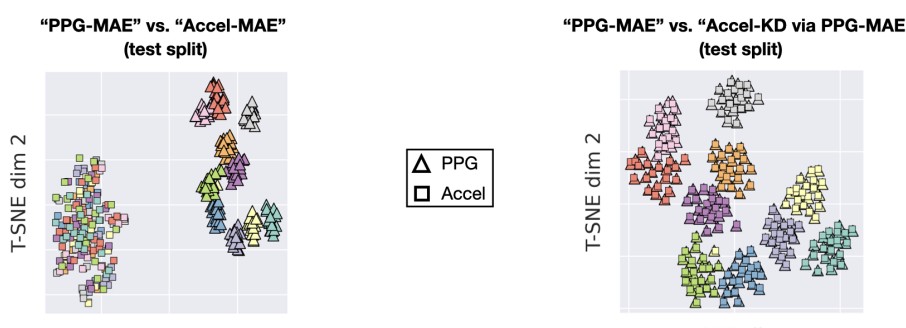

Figure 4: 2D T-SNE projections of embeddings for PPG pre-trained teacher encoder ("PPG-MAE") and 2 accelerometer encoders: 1) uni-modal encoder pre-trained with masked autoencoding ("Accel-MAE", left), 3) distilled encoder from the PPG teacher encoder ("Accel-MAE via PPG-KD", right). We can visually see marked alignment after distillation in the right panel. Each marker represents an individual segment, where markers are colored based on participants and segments are identical across panels. See retrieval analysis numbers in Table 1.

**Ablation on initialization of the PPG encoder in multi-modal pre-training**: In multi-modal pre-training mentioned in Section 5.5, even if we initialize the PPG encoder with PPG-MAE (Table 19), the downstream predictions are not as good as knowledge distillation in Table 3, indicating the importance of freezing the teacher encoder.

## A.4 DISCUSSION

**Discussion for why in uni-modal pre-training with contrastive learning was better for accelerometer, and with masked autoencoding was better for PPG**: Our hypothesis for this is that given that accelerometer is noisier than PPG (see Figure 1 for examples), reconstructing accelerometer in the output space via MAE is a rather difficult pre-training method that bottlenecks the quality of extracted representations, as opposed to that for a signal such as PPG which is more structured and less noisier to reconstruct. Therefore, we think MAE pre-training may be more suitable for less noisy biosignals, and CL pre-training is more suitable for noisier biosignals. This, in fact, is a major motivation and difference for pre-training strategies that reconstruct in the representation space versus output space (Littwin et al., 2024). All in all, for both pre-training frameworks and Transformer/EfficientNet architectures, our representational knowledge distillation framework robustly distills the information from PPG to accelerometer, and improves the information in the accelerometer embeddings.

Table 4: Common hyperparameters for pre-training experiments involving Transformers.

| Hyperparameters | Value |
|---|---|
| Patch window | 0.3125s (non-overlapping) |
| Tokenizer | Learnable linear |
| Token dim | 256 |
| Number of layers | 8 |
| Attention heads | 8 |
| MLP feedforward hidden dim | 1024 |
| Normalization | Layer norm (pre) |
| Positional embedding | Sinusoidal |
| Activation | GeLU |
| Output aggregation | Global average pooling |
| Optimizer | AdamW |
| Max learning rate | Variable (refer to text) |
| Weight decay | 1e-5 |
| Learning rate scheduling | Warmup for 20K iters, then exp. decay with $\gamma = 0.985$ every 1K iters |
| Gradient clipping | Max norm at 3 |

Table 5: Common hyperparameters for pre-training experiments involving EfficientNet (Tan & Le, 2020), adapted for time-series as proposed in (Abbaspourazad et al., 2024). Here, MBConv1D refers to the 1D version of Mobile Inverted Bottleneck, the standard building block of EfficientNet.

| Hyperparameters | Value |
|---|---|
| **Encoder Architecture** | |
|     Input layer | Conv1D, BN, Swish |
|     Number of MBConv1D blocks | 16 |
|     Output layer | Conv1D, BN, Swish, Avg. pooling |
| **MBConv1D** | |
|     Number of Conv1D layers | 5 |
|     Expansion factor | 7 |
|     Activation | Swish |
|     Normalization | Batchnorm (BN) |
|     Kernel size | 3 (first 8 blocks), 5 (last 8 blocks) |
| Optimizer | Adam |
| Max learning rate | 1e-3 |
| Weight decay | 1e-5 |
| Learning rate scheduling | Step decay with $\gamma = 0.5$ every 125K iters |
| Gradient clipping | - |

Table 6: Number of participants/segments, average number of calendar days per participant, and total dataset time span (time between the earliest to the latest recorded segment) in our pre-training datasets. We used the data from 80% of the participants for training, 10% for validation and 10% for test.

| | |
|---|---|
| Number of participants | 172,318 |
| Number of segments | 19,993,427 |
| Average number of calendar days per participant | 84.12 |
| Total dataset time span (days from 2019-09-21 to 2023-08-29) | 1,439 |

Table 7: Changes in Transformer architecture parameters when scaling its size. Size names ("XS" to "XL") are picked arbitrarily and are just relative.

| Transformer size | XS | S | M | L | XL |
|---|---|---|---|---|---|
| Token dim | 128 | 128 | 192 | 256 | 256 |
| Number of layers | 4 | 6 | 6 | 6 | 8 |
| MLP feedforward hidden dim | 512 | 512 | 1024 | 1024 | 1024 |
| Attention heads | 4 | 4 | 6 | 8 | 8 |
| Number of parameters | 800K | 1.2M | 3.3M | 4.8M | 6.3M |

Table 8: Retrieval analysis for accelerometer embeddings from PPG demonstrates near perfect alignment. Numbers are reported as average (std) across 100 bootstrap candidate pools. We would like to note that due to strong alignment of PPG and accelerometer embeddings in "Accel-KD via PPG-MAE", the statistics of retrieval analysis for accelerometer embeddings from PPG embeddings in this table is very similar to that in Table 1 for retrieval analysis of PPG embeddings from accelerometer embeddings.

| Embedding | Top-1 Acc. ↑ | Top-3 Acc. ↑ | Top-5 Acc. ↑ | Mean Rank ↓ |
|---|---|---|---|---|
| Accel-KD via PPG-MAE | **99.17 (0.23)** | **99.72 (0.08)** | **99.98 (0.02)** | **1.02 (0.01)** |
| Accel-MAE (+ Procrustes alignment) | 0.31 (0.04) | 0.80 (0.07) | 1.22 (0.09) | 2311.50 (60.74) |
| Chance-level performance | 0.01 | 0.02 | 0.03 | 9551.64 |

Table 9: Linear probing and supervised evaluation for 100% (0.1%) data availability of heart rate (reported in BPM) in Figure 2a for accelerometer and PPG encoders. Best performance based on 0.1% data availability is shown in bold, separately for each modality.

| Encoder | Mean absolute error ↓ | Root mean squared error ↓ | Pearson's R ↑ |
|---|---|---|---|
| Accel-supervised | 1.60 (3.94) | 4.28 (7.76) | 0.87 (0.30) |
| Accel-MAE | 4.47 (4.72) | 7.63 (7.90) | 0.70 (0.67) |
| Accel-CL | 2.33 (2.54) | 4.58 (4.80) | 0.89 (0.88) |
| Accel-KD via PPG-MAE | **1.19 (1.34)** | **3.12 (3.20)** | **0.96 (0.95)** |
| Accel-KD via PPG-CL | 1.37 (1.58) | 3.45 (3.56) | 0.94 (0.93) |
| PPG-MAE | **0.34 (0.36)** | **0.55 (0.59)** | **0.99 (0.98)** |
| PPG-CL | 1.20 (1.28) | 1.63 (1.74) | 0.98 (0.97) |

Table 10: Linear probing and supervised evaluation for 100% (0.1%) data availability of SDNN (reported in ms) in Figure 2a for accelerometer and PPG encoders. Best performance based on 0.1% data availability is shown in bold, separately for each modality.

| Encoder | Mean absolute error ↓ | Root mean squared error ↓ | Pearson's R ↑ |
|---|---|---|---|
| Accel-supervised | 7.82 (22.38) | 14.86 (32.30) | 0.77 (0.12) |
| Accel-MAE | 14.31 (15.38) | 22.74 (23.80) | 0.47 (0.42) |
| Accel-CL | 9.93 (10.64) | 16.47 (17.23) | 0.72 (0.70) |
| Accel-KD via PPG-MAE | **6.54 (6.95)** | **12.62 (13.11)** | **0.84 (0.83)** |
| Accel-KD via PPG-CL | 7.16 (8.04) | 13.21 (14.04) | 0.82 (0.80) |
| PPG-MAE | **4.46 (4.92)** | **7.51 (8.07)** | **0.94 (0.93)** |
| PPG-CL | 7.17 (7.55) | 11.09 (11.72) | 0.87 (0.86) |

Table 11: Linear probing and supervised evaluation for 100% (0.1%) data availability of RMSSD (reported in ms) in Figure 2a for accelerometer and PPG encoders. Best performance based on 0.1% data availability is shown in bold, separately for each modality.

| Encoder | Mean absolute error ↓ | Root mean squared error ↓ | Pearson's R ↑ |
|---|---|---|---|
| Accel-supervised | 9.61 (22.60) | 18.74 (36.01) | 0.70 (0.08) |
| Accel-MAE | 15.70 (17.00) | 27.03 (28.17) | 0.40 (0.35) |
| Accel-CL | 10.95 (12.39) | 19.83 (21.01) | 0.68 (0.64) |
| Accel-KD via PPG-MAE | **8.47 (9.17)** | **16.70 (17.33)** | **0.77 (0.75)** |
| Accel-KD via PPG-CL | 8.42 (9.57) | 16.82 (17.63) | 0.76 (0.74) |
| PPG-MAE | **6.34 (7.05)** | **11.18 (12.00)** | **0.90 (0.88)** |
| PPG-CL | 8.02 (8.71) | 13.67 (14.86) | 0.85 (0.83) |

Table 12: Downstream target evaluations reported in ROC AUC for AHMS survey questions. We observed that the distilled accelerometer encoders consistently better predicted these targets compared to the uni-modal accelerometer encoders. Asterisks indicate statistical significance for the comparison between the best distilled accelerometer encoder ("Accel-KD via PPG-MAE") versus the best uni-modal accelerometer encoder ("Accel-CL"). For statistical significance, we calculated 200 bootstrapped ROC AUC values for each evaluation, and then performed one-sided Wilcoxon Rank-Sum test to compute the p-value of the statistical comparison. We report "***" for $P < 5e-4$, "**" for $5e-4 \leq P < 5e-3$, "*" for $5e-3 \leq P < 5e-2$ and "n.s." for $P \geq 5e-2$, where $P$ is the p-value of the comparison.

| Name | Accel-KD via PPG-MAE | Accel-KD via PPG-CL | Accel-MAE | Accel-CL |
|---|---|---|---|---|
| ACE-inhibitors | **0.802** (***) | 0.794 | 0.731 | 0.791 |
| Active alcohol user | **0.681** (***) | 0.675 | 0.616 | 0.665 |
| Active smoker | **0.810** (***) | 0.801 | 0.735 | 0.784 |
| Afib | **0.816** (***) | 0.804 | 0.765 | 0.798 |
| Allergy | **0.652** (***) | 0.648 | 0.619 | 0.644 |
| Anti-anxiety | **0.713** (***) | 0.707 | 0.641 | 0.696 |
| Anti-psychotics | **0.796** (***) | 0.785 | 0.705 | 0.767 |
| Anticoagulants | **0.818** (***) | 0.809 | 0.759 | 0.801 |
| Antidepressants | **0.795** (***) | 0.782 | 0.685 | 0.761 |
| Antiplatelets | **0.784** (***) | 0.781 | 0.732 | 0.776 |
| Anxiety | **0.767** (***) | 0.759 | 0.679 | 0.747 |
| Artery disease | **0.880** (*) | 0.869 | 0.822 | 0.873 |
| Arthritis | **0.781** (***) | 0.773 | 0.733 | 0.774 |
| Asthma | **0.634** (***) | 0.630 | 0.596 | 0.621 |
| Beta-blockers | **0.759** (***) | 0.747 | 0.690 | 0.736 |
| Blood pressure | **0.798** (***) | 0.789 | 0.732 | 0.787 |
| Blood pressure med. | **0.710** (***) | 0.697 | 0.651 | 0.694 |
| Calcium-channel blockers | **0.772** (***) | 0.759 | 0.703 | 0.757 |
| Cancer | **0.800** (***) | 0.791 | 0.743 | 0.793 |
| Chemotherapy | **0.735** (***) | 0.704 | 0.626 | 0.714 |
| Cholesterol | **0.755** (***) | 0.747 | 0.703 | 0.746 |
| Chronic bronchitis | **0.725** (***) | 0.725 | 0.683 | 0.714 |
| Depression | **0.740** (***) | 0.735 | 0.665 | 0.722 |
| Diabetes | **0.829** (***) | 0.818 | 0.767 | 0.810 |
| Diuretics | **0.756** (***) | 0.747 | 0.701 | 0.743 |
| Hearing | **0.719** (***) | 0.713 | 0.676 | 0.709 |
| Heart attack | **0.835** (*) | 0.832 | 0.771 | 0.831 |
| Heart disease | **0.857** (***) | 0.845 | 0.801 | 0.843 |
| Heart failure | **0.857** (n.s.) | 0.838 | 0.789 | 0.855 |
| Heart rhythm | **0.678** (***) | 0.664 | 0.634 | 0.663 |
| Hip/Knee | **0.844** (*) | 0.842 | 0.790 | 0.841 |
| Kidney | **0.694** (***) | 0.687 | 0.646 | 0.678 |
| Liver | **0.729** (***) | 0.713 | 0.615 | 0.696 |
| Lower back | **0.685** (***) | 0.681 | 0.651 | 0.674 |
| Neck disorder | **0.724** (***) | 0.717 | 0.676 | 0.714 |
| Neuropathy | **0.802** (***) | 0.793 | 0.747 | 0.791 |
| Opioid painkillers | **0.769** (***) | 0.763 | 0.667 | 0.748 |
| Osteoporosis | **0.854** (***) | 0.849 | 0.807 | 0.851 |
| Pacemaker | **0.910** (***) | 0.884 | 0.835 | 0.885 |
| Painkillers | **0.602** (***) | 0.599 | 0.578 | 0.597 |
| Sleep apnea | **0.798** (***) | **0.798** | 0.729 | 0.783 |
| Sleep medication | **0.673** (***) | 0.661 | 0.622 | 0.652 |
| Stroke or TIA | **0.790** (n.s.) | 0.779 | 0.743 | 0.789 |
| Thyroid | **0.750** (***) | 0.746 | 0.712 | 0.743 |
| Urinary | **0.799** (***) | 0.790 | 0.748 | 0.787 |
| Vision | **0.657** (***) | 0.655 | 0.627 | 0.651 |

Table 13: Ablation on training contrastive learning based models with Transformer. Here, we report linear probing performance (mean absolute error) for 100% (0.1%) data availability equivalent to Figure 2, and Appendix Tables 9, 10 and 11 when the encoder architecture in "PPG-CL", "Accel-CL" and "Accel-KD via PPG-CL" is Transformer. We observed no meaningful difference between these methods when we replaced the architecture and all of our conclusions remains the same.

|  | Heart rate (BPM) ↓ | SDNN (ms) ↓ | RMSSD (ms) ↓ |
|---|---|---|---|
| Accel-CL (EfficientNet) | 2.33 (2.54) | 9.93 (10.54) | 10.95 (12.39) |
| PPG-CL (EfficientNet) | 1.20 (1.28) | 7.17 (7.55) | 8.02 (8.71) |
| Accel-KD via PPG-CL (EfficientNet) | 1.37 (1.58) | 7.16 (8.04) | 8.42 (9.57) |
| Accel-CL (Transformer) | 2.28 (2.48) | 9.81 (10.43) | 10.87 (12.09) |
| PPG-CL (Transformer) | 1.13 (1.20) | 7.23 (7.58) | 8.08 (8.73) |
| Accel-KD via PPG-CL (Transformer) | 1.41 (1.59) | 7.17 (7.92) | 8.67 (9.67) |

Table 14: Ablation on the choice of positive pair selection in teacher pre-training with contrastive learning ("PPG-CL").

| Eval. ↓ | Participant-level positive pairs | Segment-level positive pairs |
|---|---|---|
| Heart rate (BPM) | 2.51 | **1.21** |
| SDNN (ms) | 12.24 | **6.58** |
| RMSSD (ms) | 11.76 | **8.40** |

Table 15: Ablation on number of PPG channels in teacher pre-training ("PPG-MAE").

| Eval. ↓ | PPG-MAE w/ 1 PPG channels | PPG-MAE w/ 4 PPG channels |
|---|---|---|
| Heart rate (BPM) | 0.39 | **0.34** |
| SDNN (ms) | 5.04 | **4.46** |
| RMSSD (ms) | 7.30 | **6.34** |

Table 16: Ablation on larger model size for "PPG-MAE".

| Eval. ↓ | PPG-MAE (12.7M) | PPG-MAE (6.3M) |
|---|---|---|
| Heart rate (BPM) | **0.34** | **0.34** |
| SDNN (ms) | 5.05 | **4.46** |
| RMSSD (ms) | 7.52 | **6.34** |

Table 17: Ablation on augmentations during cross-modal knowledge distillation.

| Eval. ↓ | W/o augs. | W augs. | Cut out | Gauss. noise | Mag. warp | Channel perm. | T. warp |
|---|---|---|---|---|---|---|---|
| Heart rate (BPM) | 1.76 | **1.21** | 1.38 | 1.29 | 1.76 | 1.79 | 1.84 |
| SDNN (ms) | 6.98 | **6.58** | 6.96 | 6.66 | 6.97 | 6.77 | 10.47 |
| RMSSD (ms) | 8.82 | **8.40** | 8.51 | 8.84 | 8.96 | 8.67 | 12.83 |

Table 18: Ablation on $\lambda$ in the multi-modal contrastive objective (Equation 2).

| Eval. ↓ | $\lambda = 0$ | $\lambda = 0.25$ | $\lambda = 0.5$ | $\lambda = 0.75$ | $\lambda = 1$ |
|---|---|---|---|---|---|
| Heart rate (BPM) | 1.25 | 1.26 | 1.25 | **1.19** | 1.21 |
| SDNN (ms) | 6.80 | 6.69 | 6.77 | 6.80 | **6.58** |
| RMSSD (ms) | 8.78 | 8.59 | 8.68 | 8.78 | **8.40** |

Table 19: Ablation on not freezing the PPG encoder in cross-modal knowledge distillation in Table 3.

| Pre-training framework | Eval. ↓ | $\lambda = 0$ | $\lambda = 0.5$ | $\lambda = 1$ |
|---|---|---|---|---|
| Simultaneous multi-modal contrastive learning (PPG encoder initialized with "PPG-MAE" weights) | Heart rate (BPM) | 2.73 | 2.61 | 2.23 |
|  | SDNN (ms) | 10.97 | 10.29 | 9.63 |
|  | RMSSD (ms) | 12.56 | 11.89 | 11.06 |

Table 20: AHMS survey questions about medical conditions. The main question is in form of 'Have you ever been diagnosed with any of the following conditions?' and participants can answer 'Yes' or 'No' or 'I prefer not to answer' or 'I don't know'. The question for vision and hearing loss is different, which we explicitly mention in the corresponding rows. Third column indicates the number of left out participants for evaluation – the reason for variations is that for each target we exclude participants whose answers were 'I prefer not to answer' or 'I don't know' or missing.

| Target label | Medical condition | N (test) |
|---|---|---|
| Heart attack | Heart attack (myocardial infarction) | 26,806 |
| Heart disease | Coronary heart disease or angina pectoris | 26,584 |
| Blood pressure | High blood pressure (hypertension) | 26,326 |
| Stroke or TIA | Stroke (cerebral hemorrhage, cerebral thrombosis) or transient ischemic attack (ministroke) | 26,805 |
| Afib | Atrial fibrillation | 26,342 |
| Heart rhythm | Heart rhythm problem other than atrial fibrillation | 26,068 |
| Pacemaker | Pacemaker | 26,916 |
| Artery disease | Peripheral artery disease | 26,429 |
| Heart failure | Heart failure | 26,843 |
| Diabetes | Diabetes | 26,661 |
| Cholesterol | High cholesterol | 26,238 |
| Arthritis | Arthritis | 26,429 |
| Hip/Knee | Hip or knee replacement | 26,948 |
| Lower back | Low back disorder or other chronic back defect | 26,480 |
| Neck disorder | Neck disorder or other chronic neck defect | 26,628 |
| Osteoporosis | Osteoporosis | 26,554 |
| Asthma | Asthma | 26,762 |
| Chronic bronchitis | Chronic bronchitis, chronic obstructive pulmonary disease, or emphysema | 26,707 |
| Allergy | Rhinitis, hay fever, eye inflammation, dermatitis, food allergy or other allergy (allergic asthma excluded) | 26,626 |
| Kidney | Kidney problems | 26,640 |
| Thyroid | Thyroid disease | 26,502 |
| Cancer | Cancer | 26,783 |
| Liver | Cirrhosis of the liver | 26,784 |
| Urinary | Urinary incontinence | 26,728 |
| Neuropathy | Neuropathy | 26,370 |
| Depression | Depression | 26,110 |
| Anxiety | Anxiety disorder | 25,989 |
| Hearing | Do you have hearing loss? | 25,633 |
| Vision | Do you have vision loss? | 25,895 |

Table 21: AHMS survey questions about medications. The main question is in form of 'Do you currently take any of the following types of medications?' and participants can answer 'Yes' or 'No' or 'I prefer not to answer'. The formatting for the medications is similar to their presentation in the study, but may not exactly match the format in the study application. Third column indicates the number of left out participants for evaluation – the reason for variations is that for each target we exclude participants whose answers were 'I prefer not to answer' or missing. Third party trademarks used herein are trademarks of their respective owners.

| Target label | Medications | N (test) |
|---|---|---|
| ACE-inhibitors | ACE-inhibitors or ARBs (for blood pressure) such as captopril, enalapril, lisinopril, losartan, ramipril, or valsartan | 11,043 |
| Anti-anxiety | Anti-anxiety aids such as alprazolam (Xanax®), clonazepam (Klonopin®), clorazepate (Tranxene®), diazepam (Valium®), or lorazepam (Ativan®) | 15,876 |
| Anti-psychotics | Anti-psychotics such as haloperidol (Haldol®), aripiprazole (Abilify®), risperidone (Risperdal®), quetiapine (Seroquel®), olanzapine (Zyprexa®), clozapine (Clozaril®), or lurasidone (Latuda®) | 15,914 |
| Anticoagulants | Anticoagulants (blood thinners) such as warfarin (Coumadin®), apixaban (Eliquis®), betrixaban (Bevyxxa®), dabigatran (Pradaxa®), edoxaban (Lixiana®), or rivaroxaban (Xarelto®) | 15,906 |
| Antidepressants | Antidepressants such as amitriptyline (Elavil®), bupropion (Wellbutrin®), citalopram (Celexa®), duloxetine (Cymbalta®), escitalopram (Lexapro®), fluoxetine (Prozac®), paroxetine (Paxil®), mirtazapine (Remeron®), sertraline (Zoloft®), or venlafaxine (Effexor®) | 15,919 |
| Antiplatelets | Antiplatelets (blood thinners) such as aspirin, clopidogrel (Plavix®), prasugrel (Effient®), or ticagrelor (Brilinta®) | 15,891 |
| Beta-blockers | Beta-blockers (for blood pressure or heart rhythm) such as atenolol (Tenormin®), bisoprolol (Zebeta®), carvedilol (Coreg®), labetalol, metoprolol (Lopressor®, Toprol-XL®), nadolol (Corgard®), nebivolol (Bystolic®), propranolol (Inderal®), or sotalol (Betapace®) | 15,868 |
| Blood pressure med. | Other medications for lowering blood pressure such as clonidine, hydralazine, minoxidil, or sacubitril/valsartan (Entresto®) | 15,835 |
| Calcium-channel blockers | Calcium-channel blockers (for blood pressure or heart rhythm) such as amlodipine (Norvasc®), diltiazem, or verapamil | 15,812 |
| Chemotherapy | Certain types of chemotherapy such as carboplatin, cisplatin, oxaliplatin, vincristine, or vinblastine | 11,084 |
| Diuretics | Diuretics (water pills) such as chlorthalidone, furosemide (Lasix®), hydrochlorothiazide, or spironolactone | 15,929 |
| Opioid painkillers | Opioid painkillers such as codeine, fentanyl, hydrocodone, hydromorphone (Dilaudid®), meperidine (Demerol®), morphine, oxycodone, Percocet®, or Vicodin® | 15,941 |
| Painkillers | Non-steroidal anti-inflammatories (painkillers) such as aspirin, celecoxib (Celebrex®), diclofenac (Cambia®), ibuprofen (Motrin®/Advil®), or naproxen (Aleve®) | 15,919 |
| Sleep medication. | Sleeping aids such as eszopiclone (Lunesta®), zaleplon (Sonata®), or zolpidem (Ambien®) | 15,892 |

Table 22: AHMS survey questions about drinking and smoking habits. These are standardized questions from the AUDIT-C questionnaire (Bush et al., 1998) and All of US research program (Denny et al., 2019; Ramirez et al., 2022).

| # | Question | Answer choices |
|---|----------|----------------|
| Q1 | In your entire life, have you had at least 1 drink of any kind of alcohol, not counting small tastes or sips? | 'Yes'/'No'/'I don't know'/'I prefer not to answer' |
| Q1b | [If yes to Q1] How often did you have a drink containing alcohol in the past year? | 'Never'/'Monthly or less'/'Two to four time a month'/'Two to three times a week'/'Four or more times a week'/'I prefer not to answer' |
| Q1c | [If yes to Q1] On a typical day when you drink, how many drinks do you have? | '1 or 2'/'3 or 4'/'5 or 6'/'7 to 9'/'10 or more'/'I prefer not to answer' |
| Q2 | Have you smoked at least 100 cigarettes in your entire life? | 'Yes'/'No'/'I don't know'/'I prefer not to answer' |
| Q2b | [If Yes or Do not know to Q2] Do you now smoke cigarettes every day, some days, or not at all? | 'Every day'/'Some days'/'Not at all'/'I prefer not to answer' |

Table 23: Our logic for defining drinking and smoking targets from questions in Table 22. Third column indicates the number of left out participants for evaluation – the reason for variations is that for each target we exclude participants whose answers did not conclude in our binary 'yes' or 'no' mappings or were missing.

| Target label | Logic | N (test) |
|--------------|-------|----------|
| Active alcohol user | 'Yes': answer to Q1 is 'Yes' and answer to Q1b is 'Two to three times a week'/'Four or more times a week' 
 'No': answer to Q1 is 'No', or answer to Q1b is 'Never'/'Monthly or less'/'Two to four time a month' | 23,472 |
| Active smoker | 'Yes': answer to Q2 is 'Yes' and answer to Q2b is 'Every day'/'Some days' 
 'No': answer to Q2 is 'No', and answer to Q2b is 'Not at all' | 23,452 |

