# OpenReview forum: "Representational Knowledge Distillation Across Wearable Biosignals"
_ICLR.cc/2025/Conference — Submitted to ICLR 2025_

### Official Review · Reviewer_9r1K · 2024-10-27

**Soundness:** 1
**Presentation:** 2
**Contribution:** 2
**Rating:** 3
**Confidence:** 4

**Summary:**

The paper proposes a cross-modal representational knowledge distillation framework that aims to transfer knowledge between high-fidelity PPG and low-fidelity accelerometer signals. The encoders for the two modalities are trained using two self-supervised methods, and the distillation process is achieved through cross-modal contrastive learning. While the application of this work has practical value, there are several limitations.

**Strengths:**

This work has practical value.

**Weaknesses:**

1.Limited Methodological Innovation: The knowledge transfer between modalities is performed using basic cross-modal contrastive learning, a widely used approach. Many recent works combine both intra-modal and cross-modal self-supervised contrastive learning to learn representations, making the approach in this paper less novel.
2.Lack of Comprehensive Comparative Experiments: The paper lacks comparisons with state-of-the-art (SOTA) cross-modal distillation methods, as well as accelerometer-based prediction methods, which limits the evaluation of the proposed method's performance.
3.Unclear Impact of Data Augmentation: While the paper highlights the importance of data augmentation in cross-modal distillation, it does not provide a detailed comparison or analysis of different augmentation strategies. Additionally, wearable device data is likely to contain noise and artifacts, and although the authors mention that augmentation can help the model adapt to noisy environments, no thorough analysis is provided.
4.Limited and Unlabeled Dataset, Insufficient Downstream Tasks: Although the framework demonstrates the advantages of unsupervised learning, it lacks detailed experiments on labeled datasets, limiting the validation of its performance on tasks such as classification. The downstream tasks explored in the paper are not sufficient to fully showcase the framework's potential.

**Questions:**

Please see Weaknesses.

---

> ### Author Response · Authors · 2024-11-20
> **Response to [R-9r1K] (Part 1/n)**
>
> > Summary
>
> We thank the reviewer for reviewing our manuscript and pointing out a set of important modifications. We have responded to the reviewer’s comments below (**W**: **W**eakness, **Q**: **Q**uestion), and have addressed them accordingly in the uploaded revision of the paper. **We are looking forward to the reviewer’s revised assessment of our manuscript given the improvements suggested by all the reviewers, which we summarized in the "Summary of revision" on top.**

---

> > ### Author Response · Authors · 2024-11-20
> > **Response to [R-9r1K] (Part 2/n)**
> >
> > > Re W1
> >
> > This is an important point. We want to clarify that:
> >
> > * **We respectfully believe our methodology, our application, our dataset, and most importantly our empirical findings significantly differ from any prior work on wearable biosignals, and our manuscript provides transformative evidence of using accelerometer for diverse health applications/diagnoses to the wearable research community. We are not familiar with any prior work on representational knowledge distillation across wearable biosignals, but if the reviewer is aware of such work, we would appreciate any pointers they have so that we can update our related work and be sure to better contextualize our contributions.**
> >     * **Methodology**: While cross-modal self-supervised contrastive learning has been done in other domains of deep learning, there is no prior work on wearable biosignals that has investigated creating **generalist/foundation** embedding models via representational knowledge distillation from a high-fidelity to low-fidelity modality. Also, in details, we are not aware of any work that has our training details for wearable biosignals including the high-level method (Fig.1, training a masked autoencoder on teacher modality, and distilling it to a student modality with cross-modal contrastive learning, as well as low-level training details.
> >     * **Application**: We have repeatedly mentioned in our original submission that our goal is to build models for low-cost power-efficient biosignals that are available in any wearable without requiring any optical sensors, which is why we used accelerometer. **This is a transformative application/finding for wearable research and industry to use accelerometer for a wide array of health/diagnosis applications, and has not been shown in any prior work for accelerometer biosignals.**
> >     * **Dataset**:  Our dataset includes 172,318 participants with the data collected under natural daily life conditions covering 4 years of data. We hope that the reviewer, like we do, appreciates the challenges of modeling large-scale datasets and the generalizability of the findings in our manuscript to a broad range of demographics, which is important for health applications.
> >     * **Findings**: In our revised manuscript, we show that **a single distilled accelerometer encoder is readily (without fine-tuning) predictive of heart rate, heart rate variability, age, body mass index (BMI), biological sex, as well as 46 other health related targets spanning targets for underlying health conditions, the use of medications and lifestyle habits.** This is all done by only linear probing of the embeddings of single accelerometer encoder, which means a single compact embedding vector is predictive of all these targets. We respectfully believe, this is significantly different from any prior work. In addition, we have done several ablation studies for sweeping the number of labels, compressing the model sizes, different architecture, teacher pre-training framework, and etc. These make our manuscript of interest for practitioners.
> > * We agree that the individual modeling components used in our paper are not themselves novel in the deep learning domain, and we never claimed we created new individual components (e.g. masked autoencoding, contrastive learning, knowledge distillation, knowledge distillation across pre-trained models) — in fact, even in our original manuscript we said: “*We combine and adopt techniques inspired by uni-modal and multi-modal pre-training frameworks from other domains of deep learning to create a fully unsupervised distillation framework for biosignal time-series, which is particularly crucial for health applications where labeled data is limited*”. **That said, we respectfully believe figuring out how best to combine and adopt existing techniques in order to solve real-world problems is extremely non-trivial, particularly in such large-scale datasets and novel applications for accelerometer signals. We are not aware of any prior work that does the modeling similar to how we do for wearable biosignals.**
> > * We believe our technical implementation, solution to the problem and empirical findings are valuable to researchers and engineers in research and industry labs working with data from wearable devices. Practical applications of how to combine such techniques are non-trivial, and we would like to respectfully state that according to ICLR reviewer guidelines (https://iclr.cc/Conferences/2025/ReviewerGuide), “Submissions bring value to the ICLR community when they convincingly demonstrate new, relevant, impactful knowledge (incl., empirical, theoretical, for practitioners, etc).”, and **we believe our manuscript does fit well in this category, which is why we originally submitted our manuscript to the “Applications” track.**

---

> ### Author Response · Authors · 2024-11-20
> **Response to [R-9r1K] (Part 3/n)**
>
> > Re W2
>
> We thank the reviewer for raising this important point. **While we are not sure what SOTA methods the reviewer is referring to for unsupervised representational knowledge distillation across wearable biosignals, we respectfully want to mention that our original submission already had several comparisons versus competitive baseline methods including “*accelerometer-based*” prediction methods that the reviewer mentions**: 1) self-supervised accelerometer encoders trained with contrastive learning and masked auto encoding, with two different architectures of Transformer and EfficientNet (now even more evidence in Figure 2, Table 2, Appendix Tables 9, 10, 11, 12, 13), 2) supervised accelerometer encoders (now even more evidence in Figure 2, Appendix Tables 9, 10, 11), 3) simultaneous multi-modal contrastive learning of PPG and accelerometer (as an alternative multi-modal method, Table 3). **This is while we never claimed that the message of our paper was proposing a novel method but rather the application of transferring knowledge from PPG to accelerometer and we used our 2-stage distillation method (Fig. 1) as a means to achieve this.** We want to further clarify:
>
> * **Our original submission already included comparison with simultaneous multi-modal contrastive learning as a multi-modal method to bind cross-modal representations**: To the best of our knowledge, there’s no prior work on cross-modal representational knowledge distillation in wearable biosignals, particularly PPG to accelerometer in a dataset at such scale. **However, there’s prior work on multi-modal contrastive learning of biosignals, for which, we had a comparison in our original manuscript and demonstrated that our method achieves a significantly higher accuracy (Table 3).** In addition, in our experiments, we had tried techniques similar to self-distillation (e.g., where the embeddings are turned into a probability distribution, such as DINO [1]), and the knowledge is transferred via a cross-entropy loss, and despite various tuning attempts, we observed representational collapse in that technique. **Moreover, while we have this comparison, the message of our manuscript is the efficacy and application of knowledge distillation from PPG to accelerometer on such large-scale dataset, and not necessarily the exact means to do it.**
> * **Our original submission already included various competitive baselines for accelerometer-based prediction methods**: The reviewer comment about “*[lack] of comparisons with accelerometer-based prediction methods*” is inaccurate. We originally had detailed comparisons across various available labeled data regimes compared to 1) self-supervised accelerometer encoders trained with contrastive learning and masked auto encoding, with two different architectures of Transformer and EfficientNet (now even more evidence in Figure 2, Table 2, Appendix Tables 9, 10, 11, 12, 13), 2) supervised accelerometer encoders (now even more evidence in Figure 2, Appendix Tables 9, 10, 11). This is in contrast with the reviewer’s comment about “*[lack] of comparisons with accelerometer-based prediction methods*”; **we have covered a diverse set of ways to predict downstream targets from accelerometer with unsupervised (1) and supervised methods (2).**
> * **Our manuscript contribution is not directly the methodology but demonstrating the application**: As mentioned above in the previous weakenss, our goal is to demonstrate the efficacy and application of transferring knowledge from a PPG encoder to an accelerometer encoder, and creating one accelerometer encoder that is predictive of a wide array of downstream health targets. While we have already provided several comparisons, we respectfully do not believe additional comparisons are required to support the paper’s message because our main message is not about the method itself.
>
> [1] Caron, M., Touvron, H., Misra, I., Jégou, H., Mairal, J., Bojanowski, P., & Joulin, A. (2021). Emerging properties in self-supervised vision transformers. In Proceedings of the IEEE/CVF international conference on computer vision (pp. 9650-9660).

---

> > ### Author Response · Authors · 2024-11-20
> > **Response to [R-9r1K] (Part 4/n)**
> >
> > > Re W3
> >
> > We would like to thank the reviewer for raising this important point. We agree with the reviewer that it is necessary to investigate the importance of data augmentations in more details. To this end and to provide more detailed information about the importance of individual augmentations, we have now performed a new ablation analysis to quantify the importance of single augmentation functions in our stochastic augmentation module: “*In addition to comparing knowledge distillation with and without augmentations (Section 5.5), we studied the importance of individual augmentations during knowledge distillation. To this end, we performed the knowledge distillation from PPG to accelerometer, where we only kept one of the augmentation functions during distillation (applied in every forward pass), one at a time. This was done while maintaining all other training choices the same to control for the effect of augmentations. We observed that: 1) our stochastic augmentation module achieved the highest accuracy (Appendix Table 17), likely because it captures more diverse range of distortions during training, 2) among the individual augmentations, ”add Gaussian noise” and ”Cut out” had the highest importance, while ”Time warp” had the least importance.*”. **We have now added this ablation to our Appendix Table 17, with its corresponding text in Appendix A3.**

---

> > > ### Author Response · Authors · 2024-11-20
> > > **Response to [R-9r1K] (Part 5/n)**
> > >
> > > > Re W4 (part 1)
> > >
> > > We would like to thank the reviewer for pointing this out. We want to emphasize that our knowledge distillation framework is by design unsupervised to address the limitations of collecting large-scale labeled datasets in health domain unlike other domains of deep learning: Medical datasets are typically collected in lengthy and expensive health studies, require domain expertise for annotation, and are usually collected from limited number of participants, which can make the learned models less generalizable to broad, demographically varied populations. Therefore, we believe knowledge distillation of supervised PPG models to accelerometer models are out of scope of this manuscript, and in fact what we demonstrate is way stronger: we show a single distillation step, in an unsupervised manner, can create a single accelerometer encoder that is readily predictive of a wide array of downstream conditions with no further fine-tuning. We would like to clarify for the reviewer that:
> > >
> > > * **We originally used HR/HRV as example downstream targets because of their significance in wearable devices due to their information about an individual’s of health status, training loads and stress level, and because wearable devices typically predict HR/HRV frequently throughout the day**: We originally used HR/HRV as our downstream targets due to the importance of predicting them frequently using wearable devices throughout the day and in various environments, which matched well with our motivation of creating powerful encoders for low-fidelity but easy-to-collect biosignals such as accelerometer that are available on all wearables (see our Introduction). **These targets, particularly HRV (SDNN/RMSSD) are widely used in wearable devices [1], and are known to be indicative of health status [2], training load in athletes [3] and stress levels [4]. We have now clarified further significance of HR/HRV in the manuscript**: “*We perform linear probing for predicting heart rate (HR), and two popular measures of heart rate variability: standard deviation of normal-to-normal intervals (SDNN) and root mean square of successive differences (RMSSD). These targets are chosen because they are widely used in wearable devices (Natarajan et al., 2020) and are indicative of health status (Shaffer & Ginsberg, 2017), training load in athletes (Plews et al., 2013) and stress levels (Kim et al., 2018).*”.
> > >
> > > * **Our alignment analysis was agnostic to downstream targets and we observed near perfect alignment between accelerometer and PPG embeddings, i.e., our accelerometer encoder can capture PPG-like embeddings irrespective of the downstream target**: In addition, we had performed an alignment analysis in an unsupervised manner, demonstrating that we can match accelerometer embeddings to their correct PPG embedding pair with 99.2% top-1 accuracy. This analysis is agnostic to the downstream target and holistically estimates how well these embeddings are aligned irrespective of the downstream target. If the distilled accelerometer embeddings are aligned with the PPG embeddings, it means we can expect improved performance for distilled accelerometer embeddings (as the low-fidelity modality) for any generic downstream target due to alignment to PPG (as the high-fidelity modality).

---

> > > > ### Author Response · Authors · 2024-11-20
> > > > **Response to [R-9r1K] (Part 6/n)**
> > > >
> > > > > Re W4 (part 2)
> > > >
> > > > * **Most importantly, to mitigate the concerns regarding the downstream targets, we have added 49 new downstream targets: age, body mass index (BMI) and biological sex prediction as well as 46 health targets derived from AHMS survey questions**: We totally agree with the reviewer that additional downstream evaluations provide more context about the information in the embeddings and shed more light on their generalizability. **To this end, we added various new downstream evaluations including age, biological sex, body mass index (BMI), and a wide array of health targets collected from surveys in Apple Heart and Movement Study (AHMS) containing targets such as health conditions, use of medications, and lifestyle habits, to provide more context.** These targets evaluate different aspects in the embeddings, e.g., morphological information (as opposed to pulse timing information that may be enough for heart rate and heart rate variability), just as an example, it is well-known that PPG morphological information changes with age [5]. **These new evaluation targets also include classification tasks (biological sex and all the other 46 binary health targets) mentioned by the reviewer.** We observed that our distilled accelerometer encoder is significantly better predictive of demographic variables (age, BMI, and biological sex) and health targets compared to the baseline uni-modal accelerometer encoders, indicating its generalizability. **In addition, to the best of our knowledge, this is the first work demonstrating that a single accelerometer encoder can predict demographic variables with remarkably high accuracy, e.g., 4.04 years of error for age, 0.99 classification ROC AUC for sex, 2.46 $\text{kg}/\text{m}^2$ error for BMI, and is readily predictive of a variety of health targets. We have now added these results to the paper in new Table 2, new Appendix Table 12, new Results Section 5.3, and new supporting Appendix Tables 20, 21, 22, and 23.**
> > > >
> > > > [1] A. Natarajan, A. Pantelopoulos, H. Emir-Farinas, and P. Natarajan, “Heart rate variability with photoplethysmography in 8 million individuals: a cross-sectional study,” The Lancet Digital Health, vol. 2, no. 12, pp. e650–e657, Dec. 2020, doi: 10.1016/S2589-7500(20)30246-6.
> > > >
> > > > [2] F. Shaffer and J. P. Ginsberg, “An Overview of Heart Rate Variability Metrics and Norms,” Front. Public Health, vol. 5, p. 258, Sep. 2017, doi: 10.3389/fpubh.2017.00258.
> > > >
> > > > [3] D. J. Plews, P. B. Laursen, J. Stanley, A. E. Kilding, and M. Buchheit, “Training Adaptation and Heart Rate Variability in Elite Endurance Athletes: Opening the Door to Effective Monitoring,” Sports Med, vol. 43, no. 9, pp. 773–781, Sep. 2013, doi: 10.1007/s40279-013-0071-8.
> > > >
> > > > [4] H.-G. Kim, E.-J. Cheon, D.-S. Bai, Y. H. Lee, and B.-H. Koo, “Stress and Heart Rate Variability: A Meta-Analysis and Review of the Literature,” Psychiatry Investig, vol. 15, no. 3, pp. 235–245, Mar. 2018, doi: 10.30773/pi.2017.08.17.
> > > >
> > > > [5]: Charlton, P. H., Paliakaitė, B., Pilt, K., Bachler, M., Zanelli, S., Kulin, D., ... & Marozas, V. (2022). Assessing hemodynamics from the photoplethysmogram to gain insights into vascular age: a review from VascAgeNet. American Journal of Physiology-Heart and Circulatory Physiology, 322(4), H493-H522.

---

> ### Comment · Reviewer_9r1K · 2024-11-21
>
> There are already too many cross-modal distillation methods to reduce model parameters and improve the accuracy of small models. For example:
> [1] Cross-modal Knowledge Distillation for Vision-to-Sensor Action Recognition.
> [2] Capturing causality and bias in human action recognition.
> [3] Brain-Machine Machine Coupled Learning Method for Facial Emotion Recognition
> [4] Emotionkd: a cross-modal knowledge distillation framework for emotion recognition based on physiological signals
> [5] Multi Teacher Privileged Knowledge Distillation for Multimodal Expression Recognition
> [6] Distilling privileged multimodal information for expression recognition using optimal transport
>
> In terms of experiments, I understand that only one large-scale data set is used for pre-training. But since it is representation learning, why not use data from multiple different tasks for downstream tasks? For example, fall detection, action recognition, emotion recognition, etc.

---

> ### Author Response · Authors · 2024-11-21
> **Response to [R-9r1K] (Part 7/n)**
>
> > There are already too many cross-modal distillation methods to reduce model parameters and improve the accuracy of small models. For example: [1] Cross-modal Knowledge Distillation for Vision-to-Sensor Action Recognition. [2] Capturing causality and bias in human action recognition. [3] Brain-Machine Machine Coupled Learning Method for Facial Emotion Recognition [4] Emotionkd: a cross-modal knowledge distillation framework for emotion recognition based on physiological signals [5] Multi Teacher Privileged Knowledge Distillation for Multimodal Expression Recognition [6] Distilling privileged multimodal information for expression recognition using optimal transport
>
> Dear reviewer,
>
> None of these papers are directly related to our work. Our work focuses on improving **health** applications via cross-modal distillation **across** wearable biosignals, where **both** high-fidelity and low-fidelity modalities are **physiological signals collected from wearable devices**. Examples of wearable biosignals are ECG, PPG, IMU/Accelerometer, etc that are commonly collected from consumer wearable devices such Apple Watch, Fitbit, Oura ring, Whoop, etc. In addition, as we mention multiple times in the paper, we model accelerometer during low-motion periods where it captures minute fluctuation of the body which are a direct result of the heart function, which is distinct from activity recognition from gross motion. For example, we say “*Also, our work focuses on accelerometer during low-motion periods where it captures minute cardiovascular signals such as the ballistocardiogram, which is distinct from modeling slower changes in accelerometer during gross motion for health and fitness (Hallgrimsson et al., 2018; Ni et al.,2019; Spathis et al.,2021).*”. Therefore, using accelerometer for applications such as activity recognition or fall detection is not relevant to our work. For health applications and at low-motion periods, we show that a single accelerometer foundation model is predictive of 51 health conditions, which has never been shown before. **Below is a brief summary of the references you mentioned, none of them: 1) model teacher and student modalities that are both wearable biosignals, 2) focus on health/diagnosis applications.**
>
> [1]: Video to motion accelerometer for activity recognition
>
> [2]: (Not cross-modal) Motion accelerometer for fall detection
>
> [3]: (Not knowledge distillation) Multi-modal image-EEG for emotion recognition
>
> [4]: EEG to Galvanic Skin Response for emotion recognition
>
> [5]: Image (visual frames) to audio/EDA/EMG for expression recognition
>
> [6]: Same as 5
>
> > In terms of experiments, I understand that only one large-scale data set is used for pre-training. But since it is representation learning, why not use data from multiple different tasks for downstream tasks? For example, fall detection, action recognition, emotion recognition, etc.
>
> As we mentioned above, activity recognition/fall detection is not relevant to our work as we model accelerometer during low-motion periods where it captures minute fluctuation of the body which are a direct result of the heart function. However, **we indeed included data from 51 downstream tasks for health applications in our paper (please take a look at Figure 2, Table 2, Appendix Table 12), and several of them are related to mental health / mood.** We list them here in case you missed them but please take a look at our “summary of revision” and revised paper:
>
> **Downstream targets/tasks in our paper are**: 1) Heart rate, 2) Heart rate variability (SDNN, RMSSD), 3) Age, 4) Biological Sex, 5) BMI, 6) ACE-inhibitors meds, 7) Active alcohol use, 8) Active smoking, 9) Afib, 10) Allergy, 11) Anti-anxiety meds, 12) Anti-psychotics meds, 13) Anticoagulants meds, 14) Antidepressants meds, 15) Antiplatelets meds, 16) Anxiety, 17) Artery disease, 18) Arthritis, 19) Asthma, 20) Beta-blockers meds, 21) Blood pressure, 22) Blood pressure meds, 23) Calcium-channel blockers meds, 24) Cancer, 25) Chemotherapy, 26) Cholesterol, 27) Chronic bronchitis, 28) Depression, 29) Diabetes, 30) Diuretics meds, 31) Heart attack, 32) Heart disease, 33) Heart failure, 34) Heart rhythm, 35) Hip/Knee, 36) Kidney, 37) Liver, 38) Lower back, 39) Neck disorder, 40) Neuropathy, 41) Opioid painkillers, 42) Osteoporosis, 43) Pacemaker, 44) Painkillers, 45) Sleep apnea, 46) Sleep meds, 47) Stroke or TIA, 48) Thyroid, 49) Urinary, 50) Vision, 51) Hearing.

---

> > ### Author Response · Authors · 2024-11-27
> > **Follow up**
> >
> > Dear reviewer,
> >
> > Thanks again for your feedback regarding our paper. We have responded to your initial and follow up comments / questions and have incorporated them accordingly into our revised manuscript.
> >
> > We would greatly appreciate you taking a moment to review our response, reassess our revised manuscript from all aspects, share your thoughts, and update your score accordingly. We, along with other reviewers, sincerely believe that our paper has been strengthened a lot thanks to the feedback. We look forward to hearing from you.

---

> ### Author Response · Authors · 2024-12-02
> **A gentle reminder**
>
> Dear reviewer,
>
> Hope this message finds you well and thanks again for your feedback regarding our paper. Just a friendly reminder that we are approaching the end of the discussion period.
>
> We, along with other reviewers, sincerely believe that our paper has been significantly strengthened since your pre-rebuttal evaluation. We would greatly appreciate it if you could take a moment to review our response, reassess our revised manuscript from all aspects in light of the rebuttal, and consider updating your score to reflect the improvements. We look forward to hearing from you. Thank you!

---

### Official Review · Reviewer_3EkY · 2024-10-31

**Soundness:** 2
**Presentation:** 2
**Contribution:** 3
**Rating:** 6
**Confidence:** 3

**Summary:**

This work presents a cross-modality knowledge distillation method between PPG and Accelerometry data. They show that one can predict HR features from the accelerometry data by aligning the accelerometry data to the pre-trained PPG feature space.

**Strengths:**

Although cross-modality knowledge distillation has been studied before, applying this onto biosignals is a novel and interesting problem. The problem framing is well justified, in terms of using lower cost accelerometry signal to model PPG signals. The plots showing how distilling ppg encoder information into the accelerometry encoder improves performance is compelling, especially in regards to label efficiency compared to the supervised model.

**Weaknesses:**

* Potential overclaiming on distillation and representation capabilities
	* Results are based upon linear probing on HR, SDNN, and RMSSD. Each of these metrics are heart-rate-derived statistics, which somewhat form a limited evaluation on the capabilities of the knowledge distillation. Heart rate is trivially captured from PPG by measuring peak-to-peak differences, so these results only show that frequency information was distilled to the accelerometry encoder. I would be interested in understanding what other representation information from PPG data could be reflected in the accelerometry data after distillation, for example predicting a participants specific health diagnoses. Alternatively, the authors can argue how each of these 3 metrics are distinctive and capture unique information, such that the representation needs to represent 3 distinctive different ideas after distillation. This is my biggest concern with this work.


* Unclear ablation study approach
	* in Section 3.1.2 "we perform ablation in regards to the teacher pre-training method and architecture via contrastive learning with EfficientNets ". Why does it make sense to conduct ablation studies of architecture with the contrastive pre-training objective rather than the teacher pre-training MAE objective?
	* Do the ablation studies in Sec. 5.5 use a contrastive learning objective for the teacher?

**Questions:**

* Are there any hypothesis or further studies as to why augmentations lead to a stronger performance?
* Without any alignment between a PPG encoder and an Accel encoder, it should be obvious that the two respective embedding dimensions exist in different feature spaces. Therefore, rows 2-4 in Table 1 and Figure 2 seem to present trivial results. It is okay to keep as is, but marking these as a main claim seems to be a reach.

POST REBUTTAL EDIT: Changed Score from 5 -> 6

---

> ### Author Response · Authors · 2024-11-20
> **Response to [R-3EkY] (Part 1/n)**
>
> > Summary
>
> We thank the reviewer for their kind mention of our manuscript strengths and pointing out a set of important modifications. We have responded to the reviewer’s comments below (**W**: **W**eakness, **Q**: **Q**uestion), and have addressed them accordingly in the uploaded revision of the paper. We believe the reviewer’s comments have positively improved our manuscript. **We are looking forward to the reviewer’s revised assessment of our manuscript given the improvements suggested by all the reviewers, which we summarized in the "Summary of revision" on top.**
>
> > Re W1 (part 1)
>
> We would like to thank the reviewer for raising this important point and we agree with the reviewer that a more diverse set of evaluations sheds more light on the generalizability of the transferred knowledge and our distilled accelerometer encoders. **To this end and in line with the reviewer’s suggestion, we have added 49 new downstream targets: age, body mass index (BMI) and biological sex prediction as well as 46 health targets derived from AHMS survey questions (see last item below).** We would like to emphasize that:
>
> * **We originally used HR/HRV as example downstream targets because of their significance in wearable devices due to their information about an individual’s of health status, training loads and stress level, and because wearable devices typically predict HR/HRV frequently throughout the day**: We originally used HR/HRV as our downstream targets due to the importance of predicting them frequently using wearable devices throughout the day and in various environments, which matched well with our motivation of creating powerful encoders for low-fidelity but easy-to-collect biosignals such as accelerometer that are available on all wearables (see our Introduction). **These targets, particularly HRV (SDNN/RMSSD) are widely used in wearable devices [1], and are known to be indicative of health status [2], training load in athletes [3] and stress levels [4]. We have now clarified further significance of HR/HRV in the manuscript**: “*We perform linear probing for predicting heart rate (HR), and two popular measures of heart rate variability: standard deviation of normal-to-normal intervals (SDNN) and root mean square of successive differences (RMSSD). These targets are chosen because they are widely used in wearable devices (Natarajan et al., 2020) and are indicative of health status (Shaffer & Ginsberg, 2017), training load in athletes (Plews et al., 2013) and stress levels (Kim et al., 2018).*”. We want to clarify that, for estimating HRV, traditional algorithms operate by first estimating locations of peaks corresponding to individual heart beats, followed by generating statistics over inter-beat intervals [2]. While this may be lot easier to achieve with high-fidelity modalities (such as ECG or PPG), it is an extremely challenging problem with wearable accelerometer signals. We believe that to be able to estimate HRV metrics, the model has to develop some understanding of morphological features corresponding to the heart beat in time domain. This is very different from capturing frequency information that is captured in the heart rate.
> * **Our alignment analysis was agnostic to downstream targets and we observed near perfect alignment between accelerometer and PPG embeddings, i.e., our accelerometer encoder can capture PPG-like embeddings irrespective of the downstream target**: In addition, we had performed an alignment analysis in an unsupervised manner, demonstrating that we can match accelerometer embeddings to their correct PPG embedding pair with 99.2% top-1 accuracy. This analysis is agnostic to the downstream target and holistically estimates how well these embeddings are aligned irrespective of the downstream target. If the distilled accelerometer embeddings are aligned with the PPG embeddings, it means we can expect improved performance for distilled accelerometer embeddings (as the low-fidelity modality) for any generic downstream target due to alignment to PPG (as the high-fidelity modality).

---

> > ### Author Response · Authors · 2024-11-20
> > **Response to [R-3EkY] (Part 2/n)**
> >
> > > Re W1 (part 2)
> >
> > * **Most importantly, to mitigate the concerns regarding the downstream targets, we have added 49 new downstream targets: age, body mass index (BMI) and biological sex prediction as well as 46 health targets derived from AHMS survey questions**: We totally agree with the reviewer that additional downstream evaluations provide more context about the information in the embeddings and shed more light on their generalizability. **To this end, we added various new downstream evaluations including age, biological sex, body mass index (BMI), and a wide array of health targets collected from surveys in Apple Heart and Movement Study (AHMS) containing targets such as health conditions, use of medications, and lifestyle habits, to provide more context.** These targets evaluate different aspects in the embeddings, e.g., morphological information (as opposed to pulse timing information that may be enough for heart rate and heart rate variability), just as an example, it is well-known that PPG morphological information changes with age [5]. We observed that our distilled accelerometer encoder is significantly better predictive of demographic variables (age, BMI, and biological sex) and health targets compared to the baseline uni-modal accelerometer encoders, indicating its generalizability and that waveform information are also transferred to accelerometer embeddings during knowledge distillation. **In addition, to the best of our knowledge, this is the first work demonstrating that a single accelerometer encoder can predict demographic variables with remarkably high accuracy, e.g., 4.04 years of error for age, 0.99 classification ROC AUC for sex, 2.46 $\text{kg}/\text{m}^2$ error for BMI, and is readily predictive of a variety of health targets. We have now added these results to the paper in new Table 2, new Appendix Table 12, new Results Section 5.3, and new supporting Appendix Tables 20, 21, 22, and 23.**
> >
> > [1] A. Natarajan, A. Pantelopoulos, H. Emir-Farinas, and P. Natarajan, “Heart rate variability with photoplethysmography in 8 million individuals: a cross-sectional study,” The Lancet Digital Health, vol. 2, no. 12, pp. e650–e657, Dec. 2020, doi: 10.1016/S2589-7500(20)30246-6.
> >
> > [2] F. Shaffer and J. P. Ginsberg, “An Overview of Heart Rate Variability Metrics and Norms,” Front. Public Health, vol. 5, p. 258, Sep. 2017, doi: 10.3389/fpubh.2017.00258.
> >
> > [3] D. J. Plews, P. B. Laursen, J. Stanley, A. E. Kilding, and M. Buchheit, “Training Adaptation and Heart Rate Variability in Elite Endurance Athletes: Opening the Door to Effective Monitoring,” Sports Med, vol. 43, no. 9, pp. 773–781, Sep. 2013, doi: 10.1007/s40279-013-0071-8.
> >
> > [4] H.-G. Kim, E.-J. Cheon, D.-S. Bai, Y. H. Lee, and B.-H. Koo, “Stress and Heart Rate Variability: A Meta-Analysis and Review of the Literature,” Psychiatry Investig, vol. 15, no. 3, pp. 235–245, Mar. 2018, doi: 10.30773/pi.2017.08.17.
> >
> > [5]: Charlton, P. H., Paliakaitė, B., Pilt, K., Bachler, M., Zanelli, S., Kulin, D., ... & Marozas, V. (2022). Assessing hemodynamics from the photoplethysmogram to gain insights into vascular age: a review from VascAgeNet. American Journal of Physiology-Heart and Circulatory Physiology, 322(4), H493-H522.

---

> > > ### Author Response · Authors · 2024-11-20
> > > **Response to [R-3EkY] (Part 3/n)**
> > >
> > > > Re W2.1
> > >
> > > We would like to thank the reviewer for pointing this out. The analysis for contrastive learning and EfficientNet was done to provide an ablation to the generalizability of the approach to both the architecture and teacher pre-training in one shot, and originally we didn’t separate these two factors, and we agree that it made it difficult to infer the effects of each factor. **While this analysis did not directly affect the conclusions in our paper (it was an ablation), to alleviate the concerns and isolate the effects of the two factors, we now repeated the analysis for 1) teacher pre-training with contrastive learning, 2) uni-modal accelerometer pre-training with contrastive learning, 3) distilling the Teacher encoder from 1 mentioned here to student accelerometer encoder, where the encoder architectures are Transformers, and we provided these numbers in the new Appendix Table 13. Overall, we observed no meaningful and qualitative difference in contrastive learning using EfficientNet and Transformers, and all of our original conclusions remain intact.** Note that we now have the combinations of (Transformer, MAE & MAE $\rightarrow$ KD), (Transformer, CL & CL $\rightarrow$ KD), (EfficientNet, CL & CL $\rightarrow$ KD) in the paper. The 4th combination (EfficientNet, MAE) is not practically doable because masked autoencoding, as known in [1], is specific to Transformer architecture due to MAE’s components such as masking and dropping..
> > >
> > > [1]: He, K., Chen, X., Xie, S., Li, Y., Dollár, P., & Girshick, R. (2022). Masked autoencoders are scalable vision learners. In Proceedings of the IEEE/CVF conference on computer vision and pattern recognition (pp. 16000-16009).
> > >
> > > > Re W2.2
> > >
> > > This is a good point. All ablation studies, unless otherwise specified, have been done with our main method demonstrated in Figure 1. The other analysis for pre-training the teacher (or uni-modal accelerometer) with contrastive learning (with EfficientNets originally and now with Transformers as well given your comment above) was an ablation to demonstrate that our idea and distillation framework is not unique to MAE and Transformers. **However, we still agree with the reviewer that this can be communicated better. To clarify further, we have now added a clarifying sentence to the manuscript in the beginning of Section 5.5**: “*Unless otherwise specified, we performed the following ablation studies with “Accel-KD via PPG-MAE” as it is our main method (Figure 1), without loss of generalization given similar qualitative conclusions to “Accel-KD via PPG-CL” (Section 5.2 and Appendix Table 13). We also performed several other ablations presented in Appendix A.3.*”

---

> > > > ### Author Response · Authors · 2024-11-20
> > > > **Response to [R-3EkY] (Part 4/n)**
> > > >
> > > > > Re Q1
> > > >
> > > > We would like to thank the reviewer for this important point. First of all, we would like to point out that as part of the revision, **to provide more detailed information about the importance of individual augmentations, we have now performed a new ablation analysis to quantify the importance of single augmentation functions in our stochastic augmentation module**: “*In addition to comparing knowledge distillation with and without augmentations (Section 5.5), we studied the importance of individual augmentations during knowledge distillation. To this end, we performed the knowledge distillation from PPG to accelerometer, where we only kept one of the augmentation functions during distillation (applied in every forward pass), one at a time. This was done while maintaining all other training choices the same to control for the effect of augmentations. We observed that: 1) our stochastic augmentation module achieved the highest accuracy (Appendix Table 17), likely because it captures more diverse range of distortions during training, 2) among the individual augmentations, ”add Gaussian noise” and ”Cut out” had the highest importance, while ”Time warp” had the least importance.*”. **We have now added this ablation to our Appendix Table 17, with its corresponding text in Appendix A3.** To answer the reviewer’s question: “*In general, our hypothesis for why augmentations are important for knowledge distillation across PPG and accelerometer is that given the relationship between arterial blood flow present in PPG and ballistocardiogram in accelerometer (Inan et al., 2015; Kim et al., 2016), particularly for aligned PPG-accelerometer segments during low-motion periods which is the focus of our work, knowledge distillation without augmentations is a relatively easier pre-training task compared to that with augmentations. Therefore, we think distillation without augmentations, and even very simple augmentations as shown by individual augmentations results in Appendix Table 17, leads to capturing less minute information, which is relatively similar to why and how the amount and type of augmentations in uni-modal contrastive learning is critical as demonstrated in prior work (Chen et al., 2020).*”, **which we have now also added to the Appendix Section A3 for more clarity.**
> > > >
> > > > [1] Inan, O. T., Migeotte, P. F., Park, K. S., Etemadi, M., Tavakolian, K., Casanella, R., ... & Di Rienzo, M. (2014). Ballistocardiography and seismocardiography: A review of recent advances. IEEE journal of biomedical and health informatics, 19(4), 1414-1427.
> > > >
> > > > [2] Kim, C. S., Ober, S. L., McMurtry, M. S., Finegan, B. A., Inan, O. T., Mukkamala, R., & Hahn, J. O. (2016). Ballistocardiogram: Mechanism and potential for unobtrusive cardiovascular health monitoring. Scientific reports, 6(1), 31297.
> > > >
> > > > [3] Chen, T., Kornblith, S., Norouzi, M., & Hinton, G. (2020, November). A simple framework for contrastive learning of visual representations. In International conference on machine learning (pp. 1597-1607). PMLR.

---

> > > > > ### Author Response · Authors · 2024-11-20
> > > > > **Response to [R-3EkY] (Part 5/n)**
> > > > >
> > > > > > Re Q2
> > > > >
> > > > > We thank the reviewer for their suggestion and raising this important point. We would like to apologize if our previous Section 5.1 did not convey the message properly.
> > > > > * We agree with the reviewer that “Accel-random” did not provide additional insight and **have now removed “Accel-random” from the previous Table 1,  previous Appendix Table 9 and previous Figure 2 (now Table 1, Appendix Table 8, and Appendix Figure 4) according to the reviewer’s feedback.**
> > > > > * We would like to respectfully emphasize that “random guessing” was meant to provide the chance performance (not baseline) so it would be clear how much above chance performance the other numbers are. **Therefore, we kept that row and changed to “Chance-level performance” for more transparency.**
> > > > > * We believe it is important for us to emphasize that the reason we had provided “Accel-MAE” comparison with Procrustes transformation (which learns the optimal rotation, shifting, and scaling that makes one embedding as close as possible to another embedding) was to rule out that the near perfect alignment of accelerometer and PPG after distillation was simply achievable with uni-modal encoders (again not baseline, but just to provide context). **All in all, we also want to clarify that, we had performed the alignment analysis in an unsupervised manner, to demonstrate that we can match accelerometer embeddings to their correct PPG embedding pair with 99.2% top-1 accuracy, therefore, our main purpose was to showcase this absolute performance (99.2% top-1 accuracy for retrieval), as opposed to drawing comparisons; although, we provided some extra numbers to contextualize this absolute performance.** The reason for the importance of this absolute alignment performance is that if the distilled accelerometer embeddings are aligned with the PPG embeddings, it means we can expect improved performance for distilled accelerometer embeddings (as the low-fidelity modality) for any generic downstream target due to alignment to PPG (as the high-fidelity modality).
> > > > > * **We have now modified section 5.1 to better convey all these points, updated Table 1 and Appendix Table 8 for more clarification and transparency, and here we quote part of it**: “*To rule out that this near perfect retrieval performance was simply achievable with uni-modal encoder embeddings, we repeated the retrieval analysis using the uni-modal accelerometer embeddings from “Accel-MAE” and applying optimal translation, rotation and scaling via Procrustes alignment (Krzanowski, 2000) to make them as close as possible to "PPG-MAE'“ embeddings, and observed marked difference in the retrieval performance (Table 1 and Appendix Table 8). Overall, very high retrieval performance (e.g., 99.17 top-1 accuracy) demonstrates the effectiveness of our representational knowledge distillation framework and how well the distilled accelerometer encoder embeddings match with PPG teacher encoder embeddings. Importantly, these results indicate that distilled accelerometer embeddings may achieve improved performance for predicting downstream targets due to their high alignment with the high-fidelity PPG embeddings, which we will investigate in the next two sections.*”

---

> ### Comment · Reviewer_3EkY · 2024-11-23
> **Response to authors**
>
> I appreciate the softened language around the claims of the paper, as well as the increased amount of downstream targets, showing the increased utility of the embedding alignment. While the scientific usefulness of this work is (in my opinion) quite compelling, I think that the primary concern of the other reviewers is rooted in that ICLR may not be be the best venue to show this off, due to the limited machine learning methodological advancements, benchmarks, reproducibility, and very high specificity in it's application.
>
> Nevertheless, I believe that this is interesting work that could be accepted into ICLR due to the strong empirical results shown, so I have adjusted my score accordingly from weak reject (5) to a weak accept (6).

---

> > ### Author Response · Authors · 2024-11-25
> > **Response to [R-3EkY] (part 6/n)**
> >
> > Dear Reviewer,
> >
> > We are truly grateful for your constructive comments throughout this process, your positive reassessment of our manuscript and your kind words about the "*scientific usefulness*" of our work.
> >
> > Thanks!

---

### Official Review · Reviewer_GDg8 · 2024-11-01

**Soundness:** 2
**Presentation:** 3
**Contribution:** 1
**Rating:** 3
**Confidence:** 4

**Summary:**

This paper presents a framework to distill representational knowledge from PPG to accelerometer data. As the deployment of PPG (optical) sensors are expensive, the authors proposed to use PPG during training without labels and use accelerometer after deployment for heart rate and heart rate variability. The authors used 20 million minutes of unlabeled data collected from ∼172K participants in the Apple Heart and Movement Study under informed consent for self-supervised training.

**Strengths:**

The paper used a large scale dataset for experiments. The topic of interest is important as the wearable devices are widely used. The motivation is well framed with a good amount of reference to previous works.

**Weaknesses:**

The main idea and the specific components of the paper are not novel. Obtaining a better representation from a low-fidelity signal using a high-fidelity one is already explored [1]. All the components of the framework, masking, augmentations, loss functions, distillation, tokenizing, are known to the machine learning community. No application specific novel modification is introduced.


The main weakness of the paper is its evaluation. The used dataset is private without an open access, therefore, it is hard to find the details. But the website indicates the study only requires Apple Watch and iPhone [2]. This arises the significant question of how ground truth heart rate and heart rate variability are obtained. If the ground truth values are obtained from Apple Watch, the current experiments and the presented results are significantly limited. There are several works that show the HR and HRV from Apple Watch are not highly correlated with ground truth HR values that are obtained from gold standard ECG signals [3-4-5], especially during the motion.
In my opinion, if the evaluation of the presented framework is performed with the values that are prone to errors, this invalidates the claims in the paper.


[1] Pritam Sarkar. CardioGAN: Attentive Generative Adversarial Network with Dual Discriminators for Synthesis of ECG from PPG. AAAI 2021.

[2] https://appleheartandmovementstudy.bwh.harvard.edu/

[3] Daniel Fuller, Reliability and Validity of Commercially Available Wearable Devices for Measuring Steps, Energy Expenditure, and Heart Rate: Systematic Review. JMIR 2020.

[4] Brinnae Bent, Investigating sources of inaccuracy in wearable optical heart rate sensors, NPJ digital medicine 2020.

[5] Benjamin W. Nelson, Guidelines for wrist-worn consumer wearable assessment of heart rate in biobehavioral research, NPJ digital medicine, 2020.


The baseline comparison is also extremely limited. Out of three comparisons, two of them are random guessing and random weight models. I would expect at least two more methods from signal conversion (low to high) to show and support the claim that the proposed framework is better than previous methods in non-private datasets.

As a minor weakness, the title claims "Across Wearable Biosignals" but, the paper only focuses from PPG to IMUs. The title should be modified. The current version misleads the reader.

**Questions:**

1) For the downstream task of HR estimation, why not reporting RMSE and Pearson correlation? These metrics are widely used in HR estimation [1-3-4] problem and give detailed information about the data and model.

2) The importance of the augmentations are known in self-supervised learning, did the authors explore specific augmentations for specific modality? In literature, there are several examples [2-3] showed that the augmentations chosen by the authors are not effective for the modalities.

3) During the first stage of the training, why all PPG channels (4-channels) are used? And, for 60-second long? Is there an ablation study for that?

4) What does the statement "demonstrated robustness to the number of available training labels" show us about the data and method? Normally, in self-supervised learning methods, when the amount of training labels increased, the performance also increases [1] and the variation decreases. But, in this case, even though the amount of labels increased by 1000, the improvement is close to 0.


[1] Yuzhe Yang, SimPer: Simple Self-Supervised Learning of Periodic Targets, ICLR 2023.

[2] Hangwei Qian, What makes good contrastive learning on small-scale wearable-based tasks? In Proceedings of the 28th ACM SIGKDD 2022.

[3] Berken Utku Demirel, Finding Order in Chaos: A Novel Data Augmentation Method for Time Series in Contrastive Learning, NeurIPS 2023.

[4] Jeremy Speth, Non-Contrastive Unsupervised Learning of Physiological Signals from Video. CVPR 2024

---

> ### Author Response · Authors · 2024-11-20
> **Response to [R-GDg8] (Part 1/n)**
>
> > Summary
>
> We appreciate the reviewer for detailing our manuscript’s strengths, and their thoughtful suggestions. We have responded to the reviewer’s comment below, and have addressed them accordingly in the uploaded revision of the paper. We believe the reviewer’s comments and suggestions have significantly strengthened our manuscript. **We are looking forward to the reviewer’s revised assessment of our manuscript given the improvements suggested by all the reviewers, which we summarized in the "Summary of revision" on top.**

---

> ### Author Response · Authors · 2024-11-20
> **Response to [R-GDg8] (Part 2/n)**
>
> > Re W1 "*The main idea and the specific components of the paper are not novel. Obtaining a better representation from a low-fidelity signal using a high-fidelity one is already explored [1].*" (part 1)
>
> We would like to thank the reviewer for this important point. **We have now discussed the mentioned manuscript in various locations in our manuscript (see last item below) but we believe the reference mentioned by the reviewer [1], while interesting and of great importance, does not negatively impact the contributions of our work due to major differences.** In summary, [1] presents a cycleGAN to reconstruct ECG from PPG, and the authors show that heart rate calculated directly from the reconstructed ECG from PPG provides a good estimate of the ground truth heart rate. We would like to emphasize that:
>
> * **We respectfully believe our methodology, our application, our dataset, and most importantly our empirical findings significantly differ from [1] (and any prior work), and our manuscript provides transformative evidence of using accelerometer for diverse health applications/diagnoses to the wearable research community.**
>     * **Methodology**: Methodology used in our work is distinct from [1]. We investigate transferring representational knowledge (compact embeddings) from an already trained embedding encoder for PPG to another embedding encoder for accelerometer, which falls under knowledge distillation [2, 3]. This is while the prior work investigates cross-modal reconstruction of ECG from PPG in the output domain with a cycleGAN.  In our original submission’s title/abstract/text, we have frequently emphasized that our work is “representational knowledge distillation”, which is a narrow variation of knowledge distillation referring to transferring embeddings/representations from a network to another network [2], and we disagree that [1] fits in this category, nor this prior work has investigated creating **generalist/foundation** embedding models like we have. Also, in details, almost all training and modeling choices significantly differ between the two works.
>     * **Application**: We have repeatedly mentioned in our original submission that our goal is to build models for low-cost power-efficient biosignals that are available in any wearable without requiring any optical sensors, which is why we used accelerometer. **This is a transformative application/finding for wearable research and industry to use accelerometer for a wide array of health/diagnosis applications.** The mentioned paper by the reviewer [1] (**and any prior work**), does not show this, and in fact we disagree with the reviewer that [1] has a “low-fidelity” biosignal and we believe PPG is high-fidelity and predictive of a wide array of health targets, in many cases better than ECG [4].
>     * **Dataset**:  Prior work [1] uses 4 public datasets that in total have **125** participants. **Our dataset includes 172,318 participants with the data collected under natural daily life conditions covering 4 years of data. This is 1378x larger in terms of number of participants.** We hope that the reviewer, like we do, appreciates the challenges of modeling large-scale datasets and the generalizability of the findings in our manuscript to a broad range of demographics, which is important for health applications.
>     * **Findings**: In our revised manuscript, we show that **a single distilled accelerometer encoder is readily (without fine-tuning) predictive of heart rate, heart rate variability, age, body mass index (BMI), biological sex, as well as 46 other health related targets spanning targets for underlying health conditions, use of medications and lifestyle habits.** This is all done by only linear probing of the embeddings of single accelerometer encoder, which means a single compact embedding vector is predictive of all these targets. **We respectfully believe, this is significantly different than the mentioned work by the reviewer which only looks at heart rate by counting the number of peaks from the reconstructed ECG.** In addition, we have done several ablation studies for sweeping the number of labels, compressing the model sizes, different architecture, teacher pre-training framework, and etc. These are all distinct from prior work.
>
> [1] Sarkar, Pritam, and Ali Etemad. "Cardiogan: Attentive generative adversarial network with dual discriminators for synthesis of ecg from ppg." Proceedings of the AAAI Conference on Artificial Intelligence. Vol. 35. No. 1. 2021.
>
> [2] Hinton, G. (2015). Distilling the Knowledge in a Neural Network. arXiv preprint arXiv:1503.02531.
>
> [3] Tian, Y., Krishnan, D., & Isola, P. (2019). Contrastive representation distillation. arXiv preprint arXiv:1910.10699.
>
> [4] Abbaspourazad, S., Elachqar, O., Miller, A. C., Emrani, S., Nallasamy, U., & Shapiro, I. (2023). Large-scale training of foundation models for wearable biosignals. arXiv preprint arXiv:2312.05409.

---

> > ### Author Response · Authors · 2024-11-20
> > **Response to [R-GDg8] (Part 3/n)**
> >
> > > Re W1 "*The main idea and the specific components of the paper are not novel. Obtaining a better representation from a low-fidelity signal using a high-fidelity one is already explored [1].*" (part 2)
> >
> > * **Despite the above item, to more precisely state our main contribution, we have now softened the language used in our abstract, contribution list and related work in Introduction**:
> >     * **Abstract**: “*While multi-modal modeling and cross-modal reconstruction of biosignals have been done before, here, we demonstrate that we can distill representational knowledge across biosignals with different levels of fidelity, i.e., from PPG to accelerometer, using 20million minutes of unlabeled data collected from ∼172K participants in the Apple Heart and Movement Study under informed consent*”.
> >     * **Contribution list in Introduction**: “*1) Representational knowledge distillation across wearable biosignals in a large-scale dataset: While multi-modal modeling and cross-modal reconstruction of biosignals have been done before, we study representational knowledge distillation across wearable biosignals using a dataset at such large scale with 20M minutes of multi-modal sensor data from ∼172K participants.*”.
> >     * **Related work**: “*All in all, while multi-modal modeling and cross-modal reconstruction of biosignals have been done before, our work studies unsupervised representational knowledge distillation across wearable biosignals using large-scale data collected from wearable consumer devices, to develop a single encoder for accelerometer that is readily predictive of a wide array of downstream health targets.*”
> >
> > * **We have now discussed the mentioned work by the reviewer in our manuscript**:
> >     * **In Introduction**: “*Similarly for health applications, there has been a growing interest in cross-modal reconstruction of biosignals (Sarkar & Etemad, 2020), or modeling and pre-training multiple biosignal modalities simultaneously (Deldari et al., 2022; Liu et al., 2024;Deldari et al., 2024; Thapa et al., 2024).*”.
> >     * **Related work**: “*In line with our motivation, there has been prior work on leveraging asymmetric information in biosignals, by cross-modal reconstruction of electrocardiogram (ECG) from PPG, for a more accurate estimation of heart rate (Sarkar & Etemad, 2020).*”
> >     * **Discussion**: “*Another caveat is that while this work can be used for knowledge transfer or retrieval of the high-fidelity modality embeddings, it does not provide a generative model across modalities (Sarkar & Etemad, 2020); future work can consider recent techniques to incorporate generative capabilities using unified encoder and decoder Transformers (Mizrahi et al., 2023; Meta, 2024).*”

---

> > > ### Author Response · Authors · 2024-11-20
> > > **Response to [R-GDg8] (Part 4/n)**
> > >
> > > > Re W1 "*All the components of the framework, masking, augmentations, loss functions, distillation, tokenizing, are known to the machine learning community. No application specific novel modification is introduced.*" (part 3)
> > >
> > > This is an important point. We want to clarify that:
> > >
> > > *  We agree that the individual modeling components used in our paper are not themselves novel in the deep learning domain, and we never claimed we created new individual components (e.g. masked autoencoding, contrastive learning, knowledge distillation, knowledge distillation across pre-trained models) — in fact, even in our original manuscript we said: “*We combine and adopt techniques inspired by uni-modal and multi-modal pre-training frameworks from other domains of deep learning to create a fully unsupervised distillation framework for biosignal time-series, which is particularly crucial for health applications where labeled data is limited*”. **That said, we respectfully believe figuring out how best to combine and adopt existing techniques in order to solve real-world problems is extremely non-trivial, particularly in such large-scale datasets and novel applications for accelerometer signals. We are not aware of any prior work that does the modeling similar to how we do for wearable biosignals.**
> > > * We believe our technical implementation, solution to the problem and empirical findings are valuable to researchers and engineers in research and industry labs working with data from wearable devices. Practical applications of how to combine such techniques are non-trivial, and we would like to respectfully state that according to ICLR reviewer guidelines (https://iclr.cc/Conferences/2025/ReviewerGuide), “Submissions bring value to the ICLR community when they convincingly demonstrate new, relevant, impactful knowledge (incl., empirical, theoretical, for practitioners, etc).”, and **we believe our manuscript does fit well in this category, which is why we originally submitted our manuscript to the “Applications” track.**
> > > * In line to the above point, thanks to the reviewer’s, and other reviewers’, feedback, we have now added significantly more empirical evidence about the efficacy of our distilled accelerometer encoders **including 49 new downstream evaluation targets** (see next comment and summary comment above), and many new ablations. **To the best of our knowledge, this is the first work demonstrating that a single accelerometer encoder can predict demographic variables with remarkably high accuracy, e.g., 4.04 years of error for age, 0.99 classification of sex, 2.46 Kg/m2 error for BMI, as well as accuracy/robustness we show for HR/HRV in our manuscript, and being readily predictive of other health related targets including health conditions, use of medications and lifestyle habits.** We hope the reviewer also shares this sentiment with us that this is significantly valuable for wearable research community as accelerometer is one of the most easy to collect wearable biosignals, is available in any wearable device, and yet under-explored for such applications and we show such strong performance that can unlock new opportunities for research and industry labs working on wearable devices.

---

> > > > ### Author Response · Authors · 2024-11-20
> > > > **Response to [R-GDg8] (Part 5/n)**
> > > >
> > > > > Re W2 (part 1)
> > > >
> > > > We would like to thank the reviewer for this critical suggestion. In summary, to address the reviewer’s comment, **we have now added 49 new downstream targets: age, body mass index (BMI) and biological sex prediction as well as 46 health targets derived from AHMS survey questions (see item 2 below).** We would like to clarify that:
> > > >
> > > > * **Our dataset only contains periods of low-motion, where HR/HRV are of high quality and reliably detected, and the HR/HRV derived from Apple Watch during low-motion periods are the basis of FDA-approved features**: We thank the reviewer for their thoughtful questions. As the reviewer surmises, the ground truth values (for heart rate, SDNN and RMMSD) are indeed derived from PPG signals. More specifically they are computed from tachograms, which are a sequence of the time differences between heartbeats [1] and extracted from PPG. **The same tachogram algorithms enable the Irregular Rhythm Notification Feature (IRNF) [1], which has undergone thorough clinical validation and has been cleared by the FDA.** We would like to clarify that the research presented in this paper **uses only those sensor segments during low-motion periods where the tachograms were reliably generated and in fact provide accurate heart rate values [2]**, and we disagree with the reviewer’s claim that “*HR and HRV from Apple Watch are not highly correlated with ground truth HR values that are obtained from gold standard ECG signals*”, particularly for low-motion periods. Therefore, resulting segments tend to coincide with periods of low motion, where ground truth values are of very high quality, and we had originally mentioned in the manuscript: “*Apple Watch intermittently and passively records simultaneous green PPG and accelerometer signals during low-motion periods multiple times per day.*”. These periods also happen to be the regime where accelerometer contains Ballistocardiogram signals informative of an individual’s health status [3,4]. **To address the reviewer’s comment, we now have clarified in the paper Appendix A3 that**: “*These targets are from Apple Watch's generated values during low-motion periods where PPG peaks are reliably detected, resulting in accurate prediction of heart rate and heart rate variability.*”
> > > >
> > > > [1] Using Apple Watch for Arrhythmia Detection
> > > > https://www.apple.com/ca/healthcare/docs/site/Apple_Watch_Arrhythmia_Detection.pdf
> > > >
> > > > [2] Using Apple Watch to measure heart rate, calorimetry, and activity: https://www.apple.com/health/pdf/Heart_Rate_Calorimetry_Activity_on_Apple_Watch_November_2024.pdf
> > > >
> > > > [3] Inan, O. T., Migeotte, P. F., Park, K. S., Etemadi, M., Tavakolian, K., Casanella, R., ... & Di Rienzo, M. (2014). Ballistocardiography and seismocardiography: A review of recent advances. IEEE journal of biomedical and health informatics, 19(4), 1414-1427.
> > > >
> > > > [4] Kim, C. S., Ober, S. L., McMurtry, M. S., Finegan, B. A., Inan, O. T., Mukkamala, R., & Hahn, J. O. (2016). Ballistocardiogram: Mechanism and potential for unobtrusive cardiovascular health monitoring. Scientific reports, 6(1), 31297.

---

> > > > > ### Author Response · Authors · 2024-11-20
> > > > > **Response to [R-GDg8] (Part 6/n)**
> > > > >
> > > > > > Re W2 (part 2)
> > > > >
> > > > > * **Most importantly, to mitigate the concerns regarding the downstream targets, we have added 49 new downstream targets: age, body mass index (BMI) and biological sex prediction as well as 46 health targets derived from AHMS survey questions**: Despite our argument in the previous item, we totally agree with the reviewer that a more diverse set of downstream evaluations provides more context about the information in the embeddings, demonstrates their generalizability, and can shed more light on the efficacy of our application and message. **To this end, we added various new downstream evaluations including age, biological sex, body mass index (BMI), and a wide array of health targets collected from surveys in Apple Heart and Movement Study (AHMS) containing targets such as health conditions, use of medications, and lifestyle habits, to provide more context**. These targets evaluate different aspects in the embeddings, e.g., morphological information (as opposed to pulse timing information that may be enough for heart rate and heart rate variability), just as an example, it is well-known that PPG morphological information changes with age [5]. In addition, they do not suffer from errors mentioned by the reviewer. We observed that our distilled accelerometer encoder is significantly better predictive of demographic variables (age, BMI, and biological sex) and health targets compared to the baseline uni-modal accelerometer encoders, indicating that waveform information are also transferred to accelerometer embeddings during knowledge distillation. **In addition, to the best of our knowledge, this is the first work demonstrating that a single accelerometer encoder can predict demographic variables with remarkably high accuracy, e.g., 4.04 years of error for age, 0.99 classification ROC AUC for sex, 2.46 $\text{kg}/\text{m}^2$ error for BMI, and is readily predictive of a variety of health targets. We have now added these results to the paper in new Table 2, new Appendix Table 12, new Results Section 5.3, and new supporting Appendix Tables 20, 21, 22, and 23.**
> > > > >
> > > > > * **Our alignment analysis is agnostic to downstream targets and we observed near perfect alignment between accelerometer and PPG embeddings, i.e., our accelerometer encoder can capture PPG-like embeddings irrespective of the downstream target**: We would like to emphasize that in our original submission, we had performed an alignment analysis in an unsupervised manner, demonstrating that we can match accelerometer embeddings to their correct PPG embedding pair with 99.2% top-1 accuracy. This analysis is agnostic to the downstream target and holistically estimates how well these embeddings are aligned irrespective of the downstream target. If the distilled accelerometer embeddings are aligned with the PPG embeddings, it means we can expect improved performance for distilled accelerometer embeddings (as the low-fidelity modality) for any generic downstream target due to alignment to PPG (as the high-fidelity modality).

---

> ### Author Response · Authors · 2024-11-20
> **Response to [R-GDg8] (Part 7/n)**
>
> > Re W3
>
> This is an important point. **We respectfully want to clarify that the reviewer that “*Out of three comparisons, two of them are random guessing and random weight models.*” is inaccurate.** We respectfully want to mention that **our original submission already had comparisons versus: 1) self-supervised accelerometer encoders trained with contrastive learning and masked auto encoding, with two different architectures of Transformer and EfficientNet (now even more evidence in Figure 2, Table 2, Appendix Tables 9, 10, 11, 12, 13), 2) supervised accelerometer encoders (now even more evidence in Figure 2, Appendix Tables 9, 10, 11), 3) simultaneous multi-modal contrastive learning of PPG and accelerometer (as an alternative multi-modal method, Table 3).** This is while **we never claimed that the message of our paper was proposing a novel method but rather the application of transferring knowledge from PPG to accelerometer and we used our 2-stage distillation method (Fig. 1) as a means to achieve this.** Also, we originally only had the random models for the alignment analysis purely for visualization purposes (**which we now moved to the new Appendix Figure 4**). In addition, we also would like to respectfully emphasize that we originally put “random guessing” in only one of our tables (Table 1) to provide the chance-level performance (not baseline) so it would be clear how much above chance performance the other numbers are. **Therefore, we kept that row and changed to “*Chance-level performance*” for more transparency in the new Table 1, and Appendix Table 8.**
>
>  Regarding the reviewer’s comment “*I would expect at least two more methods from signal conversion (low to high) to show and support the claim that the proposed framework is better than previous methods in non-private datasets.*”, as mentioned above **our message is to demonstrate the efficacy and application of transferring knowledge from a PPG encoder to an accelerometer encoder, and creating one accelerometer encoder that is predictive of a wide array of downstream targets. While we have already provided several comparisons, we respectfully do not believe additional comparisons are required to support the paper’s message because our main message is not about the specific method itself but the application.**
>
>  > Re W4
>
> We would like to thank the reviewer for this suggestion. We will change the title for the camera ready submission by explicitly mentioning accelerometer, but during the review period we keep it as to avoid confusion for the AC and other reviewers.
>
> > Re Q1
>
> We would like to thank the reviewer for raising this important point. We agree with the reviewer that root mean squared error (RMSE) and Pearson Correlation (Pearson’s R) are important metrics and provide additional information regarding HR/HRV estimations versus ground truth. **To address this, we have now reported these metrics, in addition to our original mean absolute error metric for HR/HRV predictions in new Appendix Tables 9, 10 and 11.** We would like to emphasize that the same order between the models still holds for HR/HRV predictions with these new evaluation metrics, i.e., distilled accelerometer encoders are still better than supervised and uni-modal accelerometer encoders, and **all our conclusions remain the same.** We believe these new metrics suggested by the reviewer provide additional information about our evaluations.

---

> > ### Author Response · Authors · 2024-11-20
> > **Response to [R-GDg8] (Part 8/n)**
> >
> > > Re Q2
> >
> > We thank the reviewer for making this point. We agree that augmentations play an important role in self-supervised learning (SSL) with joint embedding architectures. This is particularly important for contrastive SSL approaches where two views of the same input are created to form a positive pair, and the choice and amount of augmentations greatly influence the quality of learned representations [1]. Therefore, while we agree with the reviewer that “*importance of the augmentations are known in* **[uni-modal]** *self-supervised learning*”, there’s no prior work for that for multi-modal knowledge distillation with contrastive learning for biosignals. We do use contrastive loss for knowledge distillation, but the **input data in the joint embedding architecture is fundamentally different** with uni-modal contrastive learning: **the two "views" are created by choosing aligned segments from two different modalities.** We have now clarified this in our manuscript: "*While prior works have investigated augmentations for uni-modal foundation models of biosignals, the effect of augmentations for multi-modal knowledge distillation of biosignals remains unknown*". While we have used an extensive list of augmentations guided by prior work on time-series, we respectfully believe modality specific augmentations are out of scope for the message of this paper and can be guided to future work as we now mention in our manuscript's Discussion section citing the reviewer’s suggested references: "*In addition, another interesting area of investigation for future work could be experimenting with modality specific augmentations (Qian et al., 2022; Demirel & Holz, 2023).*".
> >
> > However, to still provide more detailed information about the importance of individual augmentations, we have now performed a new ablation analysis to quantify the importance of single augmentation functions in our stochastic augmentation module: “*In addition to comparing knowledge distillation with and without augmentations (Section 5.5), we studied the importance of individual augmentations during knowledge distillation. To this end, we performed the knowledge distillation from PPG to accelerometer, where we only kept one of the augmentation functions during distillation (applied in every forward pass), one at a time. This was done while maintaining all other training choices the same to control for the effect of augmentations. We observed that: 1) our stochastic augmentation module achieved the highest accuracy (Appendix Table 17), likely because it captures more diverse range of distortions during training, 2) among the individual augmentations, ”add Gaussian noise” and ”Cut out” had the highest importance, while ”Time warp” had the least importance.*”. **We have now added this ablation to our Appendix Table 17, with its corresponding text in Appendix A3.**
> >
> > [1] Chen, T., Kornblith, S., Norouzi, M., & Hinton, G. (2020, November). A simple framework for contrastive learning of visual representations. In International conference on machine learning (pp. 1597-1607). PMLR.
> >
> > > Re Q3
> >
> > This is an important point to clarify. Apple Watch records PPG segments passively and intermittently in 60-s durations by design. Also by design, it records 4 different green PPG channels given different combinations of emitting and receiving diodes on the back of the Watch. Therefore, we used the Apple Watch PPG data as is, 4-channels and 60-second, without removing any temporal and spatial content. This was to gain as much information as possible from the existing data. However, To still address the reviewer’s question and provide some evidence, we trained the first stage of training (“PPG-MAE”) only with one channel and demonstrated that the performance metrics for downstream evaluations of heart rate and heart rate variability drops as expected. **We have now added this to the paper in the new Appendix Table 15, with its corresponding text.**

---

> > > ### Author Response · Authors · 2024-11-20
> > > **Response to [R-GDg8] (Part 9/n)**
> > >
> > > > Re Q4
> > >
> > > This is an important. We would like to clarify a few points regarding the reviewer’s comment in “*Normally, in self-supervised learning methods, when the amount of training labels increased*”:
> > >
> > > * Given that self-supervised learning by definition does not typically require explicit training labels, the reviewer may be pointing to amount of unlabeled data. In that case, we agree to say that with more amount of unlabeled data (and usually model size / compute), the accuracy of self-supervised models increases, which is widely known as scaling laws [1, 2]. However, we respectfully believe that it is not the scope of our work and future work can look into that.
> > > * If the reviewer refers to **the amount of linear probing training labels**, we originally mentioned in our manuscript, **what our label efficiency analysis shows (Figure 2), is that our pre-trained models are very robust to the number of explicit target labels after pre-trainig during linear probing. This is the main motivation behind training our models on a very large-scale unlabeled dataset: to be able to predict a target accurately even with very limited amount of labeled data, and that is in line with what we showed in our label efficiency analysis (Figure 2).** This is actually a strong point about our results, for example, other reviewers find it “compelling” (e.g., [R-3EkY]).
> > > * Also, we would like to clarify that although we have relative robustness to the number of linear probing training labels, the changes are **by no means close to 0** and are still meaningful, as we originally had in our Appendix Tables (now Appendix Tables 9, 10, 11), **for example 0.41 and 0.7 drop of performance in SDNN and RMSSD**, respectively, for “Accel-KD via PPG-MAE” going from 100% to 0.1% data.
> > >
> > >
> > >
> > >
> > > [1]: Kaplan, J., McCandlish, S., Henighan, T., Brown, T. B., Chess, B., Child, R., ... & Amodei, D. (2020). Scaling laws for neural language models. arXiv preprint arXiv:2001.08361.
> > >
> > > [2]: Zhai, X., Kolesnikov, A., Houlsby, N., & Beyer, L. (2022). Scaling vision transformers. In Proceedings of the IEEE/CVF conference on computer vision and pattern recognition (pp. 12104-12113)..

---

> ### Comment · Reviewer_GDg8 · 2024-11-22
>
> I thank the authors for their detailed response and their efforts.
>
> While the novelty of this work is a secondary concern, the evaluation methodology and underlying motivation raise significant questions.
>
> First, sampling signals exclusively during low-motion periods appears to inherently bias the dataset. By excluding segments where subjects are running, walking, or climbing stairs, the dataset is probably limited to resting HR/HRV values. This narrow scope restricts the generalizability of the evaluation and risks overfitting to the specific conditions of the data.
>
> Second, the authors suggest that Ballistocardiogram (BCG) signals, detectable by the accelerometer during low motion, are informative [3,4]. While I agree with this, it raises an important question: if these BCG signals are present, why not use a signal processing pipeline to directly extract HR/HRV values? What specifically justifies deploying a large (2M-5M-parameter for a wearable device) neural network instead of a more efficient and interpretable method? This also contradicts the authors' stated focus on low-power, resource-constrained settings. If signal processing pipelines fail—which is doubtful given the low-motion sampling—why are such methods not included as baseline comparisons?
>
> Third, while I understand that downstream tasks (e.g., medication use prediction) may not be feasible with signal processing alone, traditional machine learning methods (e.g., random forests, SVMs) using simple features (mean, power, median, HR/HRV from signal processing) should be provided as a baseline. Including these with confidence intervals would highlight the relative improvement offered by the neural network model. The comparisons provided (e.g., Table 12) fail to establish the superiority of the proposed method because no traditional ML baselines with simple features are included (or a mean predictor). Furthermore, without reporting confidence intervals, it is difficult to assess whether the observed differences in metrics—some of which are very close—are significant.
>
>
> Given the limitations of the current evaluation methodology and the contradictory motivation of the manuscript, the discussion of novelty remains difficult and unnecessary at this stage. Therefore, I will maintain my original rating.

---

> > ### Author Response · Authors · 2024-11-23
> > **Response to [R-GDg8] (Part 10/n)**
> >
> > Dear reviewer,
> >
> > Please find our response to your comments in the following thread:
> >
> > > First, sampling signals exclusively during low-motion periods appears to inherently bias the dataset. By excluding segments where subjects are running, walking, or climbing stairs, the dataset is probably limited to resting HR/HRV values. This narrow scope restricts the generalizability of the evaluation and risks overfitting to the specific conditions of the data.
> >
> > Apple Watch **by design** collects these segments at low-motion periods to reliably perform predictions about an individual health and resting HR/HRV (e.g., Arrhythmia detection, [1]). **We have data from 172K participants over 4 years under natural living conditions (asleep and awake periods throughout the day), and we disagree about the concerns for generalizability of our findings to other activities because the idea is literally to model accelerometer at sedentary settings for health applications where it captures ballistocardiogram signals.** This is not doable with accelerometer during gross motion. **We want to re-emphasize that this exact type of low-motion recorded segments  has undergone regulatory approvals for FDA-cleared features on the Apple Watch [1], which means it has in fact been deemed to be generalizable from the FDA regulatory team, and it is deployed on the Apple Watch.**
> >
> > We never claimed we modeled activity data, and estimating targets such as health conditions and HRV at resting periods are extremely impactful applications. In fact, **most protocols for measuring HRV in particular strongly suggest measuring it during motionless periods that are as noise-free as possible** — “*preferentially, short-term recordings which are free of ectopy, missing data, and noise should be used*” is how [2] describes it in the European Heart Journal’s guidelines on measuring HRV for physiological interpretation and clinical use. Even for the application of tracking training loads in athletes, it is HRV during rest or after exercise that is typically used in prior works [3]. Additionally, **accurate prediction of resting HR from any wearable during free living conditions, opportunistically collected at low-motion periods, can be extremely impactful**, as resting HR is a well-known independent predictor of cardiovascular and all-cause mortality, and may be a potential therapeutic target [4]. In addition to these, **we are demonstrating predictability of a wide array of downstream conditions that could all be individually impactful on their own (e.g., 0.86 ROC AUC for predicting heart failure).**
> >
> > [1] Using Apple Watch for Arrhythmia Detection
> > https://www.apple.com/ca/healthcare/docs/site/Apple_Watch_Arrhythmia_Detection.pdf
> >
> > [2] Electrophysiology, Task Force of the European Society of Cardiology the North American Society of Pacing. "Heart rate variability: standards of measurement, physiological interpretation, and clinical use." Circulation 93.5 (1996): 1043-1065. https://www.escardio.org/static-file/Escardio/Guidelines/Scientific-Statements/guidelines-Heart-Rate-Variability-FT-1996.pdf
> >
> > [3] D. J. Plews, P. B. Laursen, J. Stanley, A. E. Kilding, and M. Buchheit, “Training Adaptation and Heart Rate Variability in Elite Endurance Athletes: Opening the Door to Effective Monitoring,” Sports Med, vol. 43, no. 9, pp. 773–781, Sep. 2013, doi: 10.1007/s40279-013-0071-8.
> >
> > [4] Fox, Kim, et al. "Resting heart rate in cardiovascular disease." Journal of the American College of Cardiology 50.9 (2007): 823-830. https://www.jacc.org/doi/abs/10.1016/j.jacc.2007.04.079

---

> ### Author Response · Authors · 2024-11-23
> **Response to [R-GDg8] (Part 11/n)**
>
> > Second, the authors suggest that Ballistocardiogram (BCG) signals, detectable by the accelerometer during low motion, are informative [3,4]. While I agree with this, it raises an important question: if these BCG signals are present, why not use a signal processing pipeline to directly extract HR/HRV values? What specifically justifies deploying a large (2M-5M-parameter for a wearable device) neural network instead of a more efficient and interpretable method? This also contradicts the authors' stated focus on low-power, resource-constrained settings. If signal processing pipelines fail—which is doubtful given the low-motion sampling—why are such methods not included as baseline comparisons?
>
> There is no contradiction here, **we show that a single accelerometer encoder is simultaneously predictive of 51 health targets, which in fact is a power-efficient solution.** For estimating HRV, traditional algorithms for PPG/ECG operate by first estimating locations of peaks corresponding to individual heart beats, followed by generating statistics over inter-beat intervals [1]. While this may be lot easier to achieve with high-fidelity modalities (such as ECG or PPG), it is an extremely challenging problem with wrist-based accelerometer signals **because peak detection methods do not work for wrist-based accelerometer like they do for PPG/ECG. An example of how different the signals are in PPG and accelerometer are depicted in Figure 1** — there are no clear peaks in accelerometer as for PPG.  This is to be expected — accelerometer picks up extremely minute movements, and the overall signal-to-noise ratio to find the specific movements corresponding to heartbeats from such data is many orders of magnitude weaker than PPG during even low-motion segments.
>
> In addition, there is no prior work demonstrating prediction of all these health conditions and HR/HRV simultaneously using accelerometer, not only with the “*signal processing pipelines*” the reviewer refer to without providing references, but also with any other modeling framework. In spite of that, we already have comparisons with supervised algorithms for accelerometer for HR/HRV and showed that our models are better not only at the full data availability regime, but also with only a limited amount of labels. **Given the large volume of data available to us, it should be clear that a flexible enough neural network trained via direct supervision should greatly outperform simple hand-crafted features from a “*signal processing pipeline*” — and even then, we see that our supervised accelerometer models are still worse than our proposed distilled accelerometer embeddings for predicting HR and HRV.**
>
> With that said, we are also demonstrating that a single accelerometer encoder is predictive of all these other targets, **a much more efficient approach than building 51 separate “*signal processing pipelines*” (or supervised encoders) for each target, likely requiring extensive manual tuning in order to get it to work (if it is even possible at all).**
>
> This work is the first work demonstrating 1) accelerometer-derived signals are predictive of all these health conditions — **this is an important scientific finding**, and 2) **a single accelerometer encoder can predict all these targets simultaneously with one embedding vector. Our results are indeed in line with our motivations** because accelerometer is 1) power efficient to collect, 2) is available in almost all wearables, particularly those without optical sensors for PPG, 3) a single accelerometer encoder can simultaneously predict 51 targets.
>
> [1] F. Shaffer and J. P. Ginsberg, “An Overview of Heart Rate Variability Metrics and Norms,” Front. Public Health, vol. 5, p. 258, Sep. 2017, doi: 10.3389/fpubh.2017.00258.

---

> ### Author Response · Authors · 2024-11-23
> **Response to [R-GDg8] (Part 12/n)**
>
> > Third, while I understand that downstream tasks (e.g., medication use prediction) may not be feasible with signal processing alone, traditional machine learning methods (e.g., random forests, SVMs) using simple features (mean, power, median, HR/HRV from signal processing) should be provided as a baseline. Including these with confidence intervals would highlight the relative improvement offered by the neural network model. The comparisons provided (e.g., Table 12) fail to establish the superiority of the proposed method because no traditional ML baselines with simple features are included (or a mean predictor). Furthermore, without reporting confidence intervals, it is difficult to assess whether the observed differences in metrics—some of which are very close—are significant.
>
> As we mentioned above and as the reviewer agrees, there’s no well-known “*signal processing pipeline*” for these targets, our work is the first of its own demonstrating accelerometer is predictive of all these demographics variables and health targets with such high accuracy.
>
> However, **to still address the reviewer’s concern, we used even a more accurate measure of HR/HRV as if we hypothetically had an accurate “*signal processing pipeline*” to measure HR/HRV from accelerometer, and showed that still we cannot predict the health targets accurately.** To this end, as baseline feature streams, we used HR/HRV from the values logged on Apple Watch (HR/HRV from PPG [1], which are better than accelerometer) + some derivative accelerometer features (amplitude / power) to predict our demographic variables and health targets. It is worth mentioning that for these baseline feature streams, we included their mean/median/std, so 3 values per feature stream as suggested by the reviewer. Also, we trained the predictor model from this final feature vector to the target using 1) linear regression, 2) random forest, and report the better performance of the two. **As expected, we demonstrate that our distilled accelerometer encoder embeddings are significantly and by a large margin better in predicting these targets. The number of held out participants in these evaluations is very large (~17k unseen participants) so even marginal differences are significant, but for the camera ready version we will add significance p-values for more transparency. These results rule out the hypothetical scenario the reviewer raises, and further validate how performant our models are.**
>
> [1] Using Apple Watch to measure heart rate, calorimetry, and activity: https://www.apple.com/health/pdf/Heart_Rate_Calorimetry_Activity_on_Apple_Watch_November_2024.pdf

---

> ### Author Response · Authors · 2024-11-23
> **Response to [R-GDg8] (Part 13/n)**
>
> 1. **Prediction of demographic variables, age, biological sex, and BMI.** Please note that the chance-level performance (mean predictor that the reviewer refers to) for age is 11.06, BMI is 5.68, and chance-level performance for biological sex (ROC AUC) is 0.5. The gap between our method and the baseline the reviewer suggests is extremely larger compared to the gap between the baseline and the mean predictor, which is an indication of how performant our models are.
>
>
> | Target      |  Accel KD via PPG-MAE (linear probing)  | HR/HRV/accel features (best of linear probing and random forest)  |
> |--------|--------|--------|
> | Age (mean absolute error) | **4.04** | 9.32|
> | BMI (mean absolute error) | **2.48**| 5.34|
> | Biological Sex (ROC AUC) | **0.99**| 0.656|
>
>
> 2. **Prediction of other health targets.** Similar to above, our embeddings are significantly better than the baseline the reviewer suggests.
>
> | Target (ROC AUC)     |  Accel KD via PPG-MAE (linear probing)  | HR/HRV/accel features  (best of linear probing and random forest)  |
> |--------|--------|--------|
> | ACE-inhibitors | **0.802**| 0.642|
> | Active alcohol user |**0.681** | 0.552|
> | Active smoking | **0.810**| 0.687|
> | Afib | **0.816**| 0.677 |
> | Allergy | **0.652**| 0.580|
> | Anti-anxiety | **0.713**| 0.612|
> | Anti-psychotics |**0.796** | 0.674|
> | Anticoagulants |**0.818** |0.651 |
> | Antidepressants |**0.795** | 0.616|
> | Antiplatelets |**0.784** | 0.662|
> | Anxiety | **0.767**| 0.628|
> | Artery disease | **0.880**| 0.684|
> | Arthritis | **0.781**| 0.665|
> | Asthma | **0.634**|0.560|
> | Beta-blockers | **0.759**| 0.619|
> | Blood pressure | **0.798**|0.658|
> | Blood pressure med | **0.710**| 0.603|
> | Calcium-channel blockers | **0.772**| 0.632|
> | Cancer | **0.800**|0.691|
> | Chemotherapy | **0.735**| 0.555|
> | Cholesterol | **0.755**| 0.647|
> | Chronic bronchitis | **0.725**| 0.624|
> | Depression | **0.740**|0.625 |
> | Diabetes | **0.829**| 0.704|
> | Diuretics |**0.756** | 0.627|
> | Hearing | **0.719** |0.627 |
> | Heart attack | **0.835**| 0.704|
> | Heart disease | **0.857**| 0.711|
> | Heart failure |**0.857**| 0.683|
> | Heart rhythm |**0.678** | 0.571|
> | Hip/Knee | **0.844**| 0.706|
> | Kidney | **0.694**| 0.599|
> | Liver | **0.729**| 0.626|
> | Lower back | **0.685**| 0.605|
> | Neck disorder | **0.724**| 0.637|
> | Neuropathy | **0.802**| 0.676|
> | Opioid painkillers | **0.769**| 0.624|
> | Osteoporosis | **0.854**| 0.707|
> | Pacemaker | **0.910**|0.797 |
> | Painkillers | **0.602**| 0.551|
> | Sleep apnea | **0.798**| 0.635|
> | Sleep meds | **0.673**| 0.592|
> | Stroke or TIA |**0.790**| 0.662|
> | Thyroid | **0.750**| 0.607|
> | Urinary | **0.799**| 0.643|
> | Vision |**0.657** | 0.587|

---

> ### Author Response · Authors · 2024-11-26
> **Response to [R-GDg8] (Part 14/n)**
>
> Dear reviewer,
>
> Hope this message finds you well. We are thankful for your suggestions to improve the quality of our manuscript. We have now added statistical significance p-values to the Appendix Table 12 to demonstrate that our distilled accelerometer encoders are **statistically** better than the baseline acceleroemter encoders, as you suggested.
>
> With this change, we have now carefully addressed your questions and concerns from your original and follow up feedback, and incorporated them appropriately in the revised manuscript. We would be grateful if you could consider reassessing our manuscript and raising your score. We believe, like other reviewers do, our manuscript has improved significantly thanks to your and other reviewers' suggestions. Thank you!
>
> > Furthermore, without reporting confidence intervals [in Appendix Table 12], it is difficult to assess whether the observed differences in metrics—some of which are very close—are significant.
>
> We have now added statistical significance p-values to the Appendix Table 12 as you suggested. As we mentioned above, the number of data points in our comparisons is very high (~17K) so even marginal differences are likely significant. To demonstrate this, we have now reported statistical significance p-value for "Accel-KD via PPG-MAE" being better than "Accel-CL" (as the best uni-modal accelerometer encoder). We observed in 44/46 comparisons, the distilled accelerometer encoder had statistically better performance. **We added additional text to the Appendix Table 12 with statistical significance comparison p-values**: "*Asterisks indicate statistical significance for the comparison between the best distilled accelerometer encoder (“Accel-KD via PPG-MAE”) versus the best uni-modal accelerometer encoder (“Accel-CL”). For statistical significance, we calculated 200 bootstrapped ROC AUC values for each evaluation, and then performed one-sided Wilcoxon Rank-Sum test to compute the p-value of the statistical comparis. We report ”\*\*\*” for P < 5e−4, ”\*\*” for 5e−4 ≤ P < 5e−3, ”\*” for 5e−3 ≤ P < 5e−2 and ”n.s.” for P ≥ 5e−2, where P is the p-value of the comparison.*".

---

> > ### Comment · Reviewer_GDg8 · 2024-11-28
> >
> > I would like to thank the authors for addressing my questions and incorporating additional baselines into the paper. I apologize for the delayed response, as I needed some time to review additional results that are added during the discussion time with related works.
> >
> > I agree with the authors that accelerometer-based HR/HRV measurement is prone to errors and challenging. However, wasn’t the primary motivation of this research to address those challenges and obtain HR/HRV from IMUs? Even the introduction emphasizes this goal.
> >
> > My previous concern was about the signal processing approaches in this direction. A literature search reveals several works that have successfully used wrist-based accelerometers to obtain HR/HRV [1-3], with some even reconstructing pulse waves for direct comparison to R peaks in ECG. These studies show that beats can be detected while excluding motion artifacts, and recent methods demonstrate strong performance. I wonder why the authors did not compare their results with these approaches.
> >
> > While I understand that these methods may not address health-related targets, considering the HR/HRV results presented in the manuscript, a future reader might see this as a fair question. Without such comparisons, the claim that *"a flexible neural network trained via direct supervision should greatly outperform simple hand-crafted features from a signal processing pipeline"* does not appear to be scientifically justified.
> >
> >
> > [1] Reconstruction of pulse wave and respiration from wrist accelerometer during sleep,” IEEE Transactions on Biomedical Engineering, 2021.
> >
> >
> > [2] Detection and analysis of pulse waves during sleep via wrist-worn actigraphy. PloS One, 2019.
> >
> >
> > [3] Nightbeat: Heart Rate Estimation From a Wrist-Worn Accelerometer During Sleep, IEEE-EMBS International Conference on Biomedical and Health Informatics, 2024.
> >
> >
> >
> > Regarding the statement *"We want to re-emphasize that this exact type of low-motion recorded segments has undergone regulatory approvals for FDA-cleared features on the Apple Watch [1], which means it has in fact been deemed to be generalizable from the FDA regulatory team,"* was this approval specifically for HR/HRV features or general heart rate? Please correct me if I’m wrong. As a reader, I can accept this in the context of HR/HRV since it aligns with the initial submission. However, as far as I know (and with the literature search), there is no FDA verification for the health-related targets (e.g., medication use, cholesterol) mentioned in the additional metrics. If such FDA clearances exist, please report and clarify them. I am not an expert on FDA clearances, but the script also misses the information.
> >
> >
> > If not, I wonder how one can confidently claim that the private dataset is verified as generalizable and valid for all tasks. Does this not raise the risk of overfitting to the specific conditions of the dataset for additional targets?
> >
> >
> > I would like to thank the authors for adding significance tests to the additional metrics. In Table 20, it is stated that the third column indicates the number of left-out participants for evaluation—for example, 26,784 for Cirrhosis of the liver. Does this mean 26,784 people were used for evaluation? Does it also imply that 26,784 segments (windows/inputs) were used?
> >
> > I noticed the explanation added to the revised text: *"In all these subject-related targets that do not vary from segment to segment, we perform the linear probing at participant granularity: we mean-aggregate all the embeddings associated with each participant so that each participant contributes one and only one sample in the downstream training/evaluation."* If I understand correctly, this means there were 26,784 points (one per participant) used in the evaluation. Is that correct?
> >
> > Additionally, the authors stated: *"The number of held-out participants in these evaluations is very large (~17k unseen participants)."* However, this number is different from what is reported in Table 20. Could you clarify why these numbers differ?
> >
> >
> > Please do not get me wrong, I am not trying to overshadow the efforts put into the paper. I am simply asking the questions I would consider as a reader, as future readers might also wonder why these points were not raised by the reviewer.
> >
> >
> > While I largely agree with Reviewer 3EkY’s feedback on limited methodological advancements, lack of benchmarks, reproducibility issues, and narrow application focus, I found the baseline comparisons for HR/HRV tasks particularly weak. The additional experiments introduced during the discussion phase raised further questions regarding the evaluation and soundness of the paper. Therefore, I am trying to clarify them.

---

> ### Author Response · Authors · 2024-11-29
> **Response to [R-GDg8] (Part 15/n)**
>
> Dear reviewer,
>
> We would like to thank you for investing the time to review our manuscript and we are more than happy to answer your questions. Please find our responses below:
>
> > I agree with the authors that accelerometer-based HR/HRV measurement is prone to errors and challenging. However, wasn’t the primary motivation of this research to address those challenges and obtain HR/HRV from IMUs? Even the introduction emphasizes this goal.
>
> This is an important point to clarify. **In our manuscript's Introduction/Abstract, we never say our primary goal is to predict HR/HRV** but if there's a specific point that has caused the reviewer to say that, we will be more than happy to address in the camera ready version (the window to update the pdf has ended). **In turn, we say our motivation is** "*developing digital biomarkers from any wearable device with lower-fidelity biosignals, and help individuals track their health more frequently and conveniently*", and in line with that, we demonstrate that a single accelerometer embedding model can capture rich embeddings that are readily predictive of a variety of targets (a very efficient solution) -- with all the new evidence, only 2 out of 51 targets we present in our revised manuscript are HR/HRV. Also, our method can work for any other downstream targets, we provided evidence with the example labels that were available to us, but future work can look into other downstream targets as well.
>
> > My previous concern was about the signal processing approaches in this direction. A literature search reveals several works that have successfully used wrist-based accelerometers to obtain HR/HRV [1-3], with some even reconstructing pulse waves for direct comparison to R peaks in ECG. These studies show that beats can be detected while excluding motion artifacts, and recent methods demonstrate strong performance. I wonder why the authors did not compare their results with these approaches.
> > While I understand that these methods may not address health-related targets, considering the HR/HRV results presented in the manuscript, a future reader might see this as a fair question. Without such comparisons, the claim that "a flexible neural network trained via direct supervision should greatly outperform simple hand-crafted features from a signal processing pipeline" does not appear to be scientifically justified.
>
> This is very important to clarify. Given our response in the previous round (see "Response to [R-GDg8] (Part {12,13,14}/n)", **we provided strong evidence that even if there was an accurate estimation of HR/HRV from accelerometer, we would still not be able to predict all the other 49 targets including demographics variables, health conditions, use of medications and lifestyle habits (even the reviewer agrees with this in their previous response, that's why they asked for the additional evidence which we provided)**. In fact, the gap of performance is dramatically large which represents that our embeddings extract significantly richer information that just HR/HRV. Given all the evidence we have provided, we believe an additional signal processing baseline for HR/HRV is out of scope for our manuscript, and future work can look into that (we can add this as a discussion to the camera ready version). As a side note, as we mentioned in our prior responses, our work covers low-motion periods throughout daily living, which is significantly more challenging than just modeling overnight accelerometer data the reviewer has provided references for. As an extra side note, we respectfully want to re-emphasize that our motivation was building efficient solutions for predicting health targets from available biosignals on all wearables such as accelerometer, and we demonstrate that a single accelerometer encoder is predictive of all these other targets. We believe this is a much more efficient approach than building separate signal processing pipelines (or supervised encoders) for each target -- in fact, we originally provided supervised evaluations to contextualize our label efficiency analysis and not as an alternative efficient solution.

---

> > ### Author Response · Authors · 2024-11-29
> > **Response to [R-GDg8] (Part 16/n)**
> >
> > > Regarding the statement "We want to re-emphasize that this exact type of low-motion recorded segments has undergone regulatory approvals for FDA-cleared features on the Apple Watch [1], which means it has in fact been deemed to be generalizable from the FDA regulatory team," was this approval specifically for HR/HRV features or general heart rate? Please correct me if I’m wrong. As a reader, I can accept this in the context of HR/HRV since it aligns with the initial submission. However, as far as I know (and with the literature search), there is no FDA verification for the health-related targets (e.g., medication use, cholesterol) mentioned in the additional metrics. If such FDA clearances exist, please report and clarify them. I am not an expert on FDA clearances, but the script also misses the information.
> >
> > Thanks for raising this question. Our reponse in "Response to [R-GDg8] (Part 10/n)" was to address your question/concern that "*sampling signals exclusively during low-motion periods appears to inherently bias the dataset. By excluding segments where subjects are running, walking, or climbing stairs, the dataset is probably limited to resting HR/HRV values. This narrow scope restricts the generalizability of the evaluation and risks overfitting to the specific conditions of the data.*". **We did not claim that Apple Watch had FDA approval for all the downstream targets presented in this paper, however, our argument was that**: Apple Watch by design collects data at low-motion periods to reliably make predictions about an individual's health. This does not reduce the generalizability of evaluations/evidence in this study, because the ultimate goal is to provide accurate predictions regarding individuals' health by opportunistically leveraging the low-motion periods throughout the day. We originally provided an example of this that is an FDA approved feature for atrial fibrillation (AFib, a significant underlying health condition that can lead to heart failure) detection (https://www.apple.com/ca/healthcare/docs/site/Apple_Watch_Arrhythmia_Detection.pdf). **FDA has approved using the exact same segments during low-motion to predict this condition, and is already deployed on the watch.** FDA has an extensive regulatory process to approve such features given their reliability and generalizability. In addition, FDA has just recently deemed this very AFib detection feature as a qualified tool in the Medical Device Development Tools, which is a testament to the quality and generalizability (https://healthmanagement.org/c/cardio/News/apple-watch-receives-fda-approval-as-mddt). Given this, we argued that we believe there are no significant concerns about generalizability that the reviewer raised because the point is to opportunistically leverage the low-motion periods, to reliably make predictions about an individual's health.

---

> > > ### Author Response · Authors · 2024-11-29
> > > **Response to [R-GDg8] (Part 17/n)**
> > >
> > > > I would like to thank the authors for adding significance tests to the additional metrics. In Table 20, it is stated that the third column indicates the number of left-out participants for evaluation—for example, 26,784 for Cirrhosis of the liver. Does this mean 26,784 people were used for evaluation? Does it also imply that 26,784 segments (windows/inputs) were used?
> > > > I noticed the explanation added to the revised text: "In all these subject-related targets that do not vary from segment to segment, we perform the linear probing at participant granularity: we mean-aggregate all the embeddings associated with each participant so that each participant contributes one and only one sample in the downstream training/evaluation." If I understand correctly, this means there were 26,784 points (one per participant) used in the evaluation. Is that correct?
> > > > Additionally, the authors stated: "The number of held-out participants in these evaluations is very large (~17k unseen participants)." However, this number is different from what is reported in Table 20. Could you clarify why these numbers differ?
> > >
> > > We would like to apologize for this confusion, we should have been more accurate with our terminology and the example number provided here. The numbers in the manuscript are correct but we should not have had an example number here, which apparantly caused confusion, and we totally understand and again apologize about that. Given the unique nature of the labels in Apple Heart and Movement Study, the total number of participants that end up in the test evaluation are different for each downstream target for the health targets we report and here we explain why. First of all, the column N in Tables 20/21/23 are for the number of participants and not segments as the reviewer inferred correctly. Second, We have a 80/20 splitting protocol stratified by participants for our downstream linear probing but given that the labels come from different survey questions (see the questions in Tables 20/21/22), some participants 1) have only partially responded to the survey questions, or 2) have answered "I do not know" or "I prefer not to answer", which are removed given our acceptable "yes/no" labels explained in the caption of those tables. Also, some of these labels come from different surveys (e.g., medications and health conditions are asked in different surveys), therefore, the frequency/availability of labels could be different. For these reasons, there's a wide variance in the number of held out participants that actually had labels for each downstream target (e.g., ranging from ~11k to ~26k in Tables 20/21/23), and there was no exact number for us to refer to here and we apologize for the misreference. We hope this further clarifies the reviewer's confusion. Our evidence/claims for the significance p-values are in tact, and we will add a paragraph further explaining these details for more clarity in the camera ready version.

---

> ### Author Response · Authors · 2024-11-29
> **Response to [R-GDg8] (Part 18/n)**
>
> > Please do not get me wrong, I am not trying to overshadow the efforts put into the paper. I am simply asking the questions I would consider as a reader, as future readers might also wonder why these points were not raised by the reviewer.
>
> We appreciate your time and energy for giving us valuable feedback, and helping us improve our manuscript.
>
> > While I largely agree with Reviewer 3EkY’s feedback on limited methodological advancements, lack of benchmarks, reproducibility issues, and narrow application focus, I found the baseline comparisons for HR/HRV tasks particularly weak. The additional experiments introduced during the discussion phase raised further questions regarding the evaluation and soundness of the paper. Therefore, I am trying to clarify them.
>
> Thanks for raising this point. We respectfully request you to evaluate our manuscript from all aspects. Overall, We believe our manuscript has breadth of information for practitioners and researchers working with wearable biosignals, and even if we don't introduce novel components (like a new loss function), we believe our approach to the problem (2-stage masked autoencoding and then distillation across modalities for an unsupervised end-to-end method), the implementation details, and empirical findings have several novelties and are valuable for the community. Even [R-3EkY] that is being referred to, has mentioned several strengths of our work in their initial and final evaluation, including "*Although cross-modality knowledge distillation has been studied before, applying this onto biosignals is a novel and interesting problem.*" and "*scientific usefulness of this work is (in my opinion) quite compelling*", and **has given us an above acceptance score.**
>
> We understand your initial concerns but we did our best to address all of them, and we respectfully believe our manuscript has improved significantly over the course of the rebuttal period from your first evaluation; we genuinely request you to evaluate the current state of our manuscript **from all aspects** with a degree of openness and for a fair evaluation, thank you!

---

> ### Author Response · Authors · 2024-12-02
> **A friendly reminder**
>
> Dear Reviewer,
>
> We hope this message finds you well. Thank you once again for your feedback on our paper and for your thoughtful comments throughout the discussion period.
>
> We have done our best to address your questions and incorporate your initial and follow-up feedback into the manuscript. We sincerely believe that the manuscript has improved significantly as a result.
>
> As we approach the end of the discussion period, we would greatly appreciate it if you could take a moment to reassess our manuscript from all aspects in light of the rebuttal, and we would be grateful if you could consider updating your score to reflect the improvements since your pre-rebuttal evaluation/score.
>
> Thank you for your time and support, and we look forward to hearing from you.

---

### Official Review · Reviewer_hLgx · 2024-11-02

**Soundness:** 3
**Presentation:** 3
**Contribution:** 2
**Rating:** 8
**Confidence:** 4

**Summary:**

The paper addresses the challenge of creating low-power, high-fidelity biosignal encoders in wearable devices by distilling knowledge from high-fidelity photoplethysmography (PPG) signals to accelerometer signals. This work is based on a large-scale dataset from the Apple Heart and Movement Study (AHMS), containing 20 million minutes of data from approximately 172,000 participants. Key contributions of the paper include:

* A representational knowledge distillation framework across biosignals, aiming to leverage high-fidelity signals for improved lower-fidelity signal representation.

* Demonstrating significant improvements in representation quality, evaluated through heart rate and heart rate variability predictions.

**Strengths:**

* **Relevant Problem:** The paper addresses a significant real-world problem with direct applications in wearable health technology.
* **Effective Solution:** The proposed method effectively improves the representation of accelerometer signals for health applications.
* **Detailed Ablation Studies:** The ablation studies provide a comprehensive analysis of the method’s behavior under various configurations.
* **Clear Writing:** The paper is well-structured, making it accessible and easy to follow.

**Weaknesses:**

* **Limited Novelty:** While the approach is effective, the novelty is somewhat limited as similar representational knowledge distillation methods exist for biosignals.
* **Contribution Clarification:** Some contributions, such as ablation studies, are essential for verifying robustness rather than stand-alone contributions.

**Questions:**

## Major Comments

* **Misleading Contribution Title:**
The first listed contribution, “Representational Knowledge Distillation across Biosignals,” could be misleading. While it is possibly a novel application to wearable signals, other biosignal domains have explored knowledge distillation. Clarifying this distinction could help prevent misunderstandings about the paper’s contributions.

* **Contrastive Learning Experiment Setup:**
In Section 3.1.2, where contrastive learning is applied as an alternative to masked autoencoding, the architecture changes alongside the learning paradigm. This confounds the results, making it difficult to isolate the effects of each factor. It may be beneficial to conduct experiments with a fixed architecture across learning paradigms to achieve a more controlled comparison.

* **Justification for Segment-Level Pair Selection:**
In Section 3.1.2, participant-level positive pairs are replaced with segment-level pairs to enhance the model’s ability to learn segment-specific information. While this adjustment is intuitive, an experiment supporting this decision would add robustness to the claim.

* **Confusing Results in Table 4:**
The results in Table 4 appear to contradict the discussion on page 10, where it is claimed that simultaneous training on both modalities causes a significant performance drop (132%, 66%, and 50%). These values do not align with the reported figures in Table 4, creating confusion. Clarifying these discrepancies would improve result interpretation.

## Minor Comments

* **Citation for AHMS Dataset:**
When the AHMS dataset is introduced in the introduction (page 2, line 077), it should be properly cited to enhance clarity and give credit to the dataset source.

* **Ablation Studies as a Contribution:**
The fifth contribution, “Ablation Studies,” may be better categorized as part of the robustness evaluation rather than a unique contribution. Demonstrating the necessity of ablation studies supports the robustness of the findings but may not stand alone as a contribution.

* **Reference for SDNN and RMSSD:**
In Section 4.2, line 307, SDNN and RMSSD are mentioned as commonly used targets without any citation. Adding references for their relevance in prior works would provide supporting context for their selection.

* **Purpose of Untrained Model Results:**
In Table 1, the results for “Accel-random” and random selection are reported without clear context. These numbers may be of limited utility, as comparing an untrained model or random selections provides little practical insight. Justifying their inclusion would improve result interpretation.

POST REBUTTAL EDIT: Changed Score from 6 -> 8

---

> ### Author Response · Authors · 2024-11-20
> **Response to [R-hLgx] (Part 1/n)**
>
> > Summary
>
> We appreciate the reviewer’s feedback, their kind words about the strengths of our work, and their careful examination of our manuscript. We have responded to the reviewer’s comments below (**W**: **W**eakness, **Q**: **Q**uestion), and have addressed them accordingly in the uploaded revision of the paper. We believe the reviewer’s comments have positively improved our manuscript. **We are looking forward to the reviewer’s updated assessment of our manuscript given the improvements suggested by all the reviewers, which we summarized in the "Summary of revision" on top.**
>
> > Re W1 & W2 & Q1
>
> We would like to thank the reviewer for raising this point. We want to clarify that to the best of our knowledge, our work is the first that demonstrates unsupervised knowledge distillation of representations across wearable biosignals, particularly so in a dataset at the scale of what we have used in our manuscript. **We are not familiar with any prior work on representational knowledge distillation across wearable biosignals, but if the reviewer is aware of such work, we would appreciate any pointers they have so that we can update our related work and be sure to better contextualize our contributions.** That said, we totally agree with the reviewer to further clarify our contributions to avoid misunderstandings, **therefore to more precisely state our main contribution, we have now softened the language used in our abstract, contribution list and related work in Introduction**:
>
> * **Abstract**: “*While multi-modal modeling and cross-modal reconstruction of biosignals have been done before, here, we demonstrate that we can distill representational knowledge across biosignals with different levels of fidelity, i.e., from PPG to accelerometer, using 20million minutes of unlabeled data collected from ∼172K participants in the Apple Heart and Movement Study under informed consent*”.
> * **Contribution list in Introduction**: “*1) Representational knowledge distillation across wearable biosignals in a large-scale dataset: While multi-modal modeling and cross-modal reconstruction of biosignals have been done before, we study representational knowledge distillation across wearable biosignals using a dataset at such large scale with 20M minutes of multi-modal sensor data from ∼172K participants.*”.
> * **Related work**: “*All in all, while multi-modal modeling and cross-modal reconstruction of biosignals have been done before, our work studies unsupervised representational knowledge distillation across wearable biosignals using large-scale data collected from wearable consumer devices to develop a single encoder for accelerometer that is readily predictive of a wide array of downstream health targets.*”
>
> > Re Q2
>
> We would like to thank the reviewer for pointing this out. The analysis for contrastive learning and EfficientNet was done to provide an ablation to the generalizability of the approach to both the architecture and teacher pre-training in one shot, and originally we didn’t separate these two factors, and we agree that it made it difficult to infer the effects of each factor. **While this analysis did not directly affect the conclusions in our paper (it was an ablation), to alleviate the concerns and isolate the effects of the two factors, we now repeated the analysis for 1) teacher pre-training with contrastive learning, 2) uni-modal accelerometer pre-training with contrastive learning, 3) distilling the Teacher encoder from 1 mentioned here to student accelerometer encoder, where the encoder architectures are Transformers, and we provided these numbers in the new Appendix Table 13.** Overall, we observed no meaningful and qualitative difference in contrastive learning using EfficientNet and Transformers, and all of our original conclusions remain intact. Note that we now have the combinations of (Transformer, MAE & MAE $\rightarrow$ KD), (Transformer, CL & CL $\rightarrow$ KD), (EfficientNet, CL & CL $\rightarrow$ KD) in the paper. The 4th combination (EfficientNet, MAE) is not practically doable because masked autoencoding, as known in [1], is specific to Transformer architecture due to MAE’s components such as masking and dropping.
>
> [1]: He, K., Chen, X., Xie, S., Li, Y., Dollár, P., & Girshick, R. (2022). Masked autoencoders are scalable vision learners. In Proceedings of the IEEE/CVF conference on computer vision and pattern recognition (pp. 16000-16009).
>
> > Re Q3
>
> This is a great point. **We have now added new Appendix Table 14 providing evidence for this selection by comparing downstream performance of these two positive pair selections for targets such as heart rate and heart rate variability.** As shown, the pre-trained teacher PPG encoder with segment-level positive pairs has significantly better prediction performance.

---

> > ### Author Response · Authors · 2024-11-20
> > **Response to [R-hLgx] (Part 2/n)**
> >
> > > Re Q4
> >
> > We would like to thank the reviewer for pointing this out. The old Table 4 only contained the simultaneous multi-modal contrastive learning performance numbers in 2 cases (where the PPG encoder was initialized 1) from scratch, 2) with the "PPG-MAE" model, and the comparisons with distilled accelerometer encoder was implicitly made from the numbers in other Tables (e.g., previous Tables 2/3 and previous Appendix Table 10). We totally understand that this caused confusion, and **we therefore, modified Result section 5.5 of the manuscript explicitly added the baseline performance numbers to previous Table 4 (new Table 3) and better labeled the comparisons with “Simultaneous multi-modal contrastive learning vs. Cross-modal knowledge distillation (ours)”. To keep the paper below 10 pages, we moved part of previous Table 4 to new Appendix Table 19, and accordingly updated the Results text in section 5.5. We believe these changes made the performance numbers more clear in the new Table 3 and new Appendix Table 19, and Section 5.5.**
> >
> > > Re Q5
> >
> > Thanks for pointing this out. **We previously had the AHMS dataset citation in the dataset section but have now added to Introduction as well**: “*In this paper, using unlabeled sensor data collected under informed consent from the large longitudinal Apple Heart and Movement Study (AHMS) (MacRae, 2021; Truslow et al., 2024),...*”.
> >
> > > Re W2 & Q6
> >
> > This is an important point, and thanks for raising it. We agree with you and we have now removed “*Ablation studies*” from contributions, and instead state the reviewer’s suggestion in Introduction: “*In addition to these contributions, to support the robustness of the findings, we perform ablation studies...*”.
> >
> > > Re Q7
> >
> > We thank the reviewer for this suggestion. We agree that adding references in this context will clarify the choice of using SDNN and RMSSD as targets. **To this end, we have added the references to the manuscript and clarified**: “*We perform linear probing for predicting heart rate (HR), and two popular measures of heart rate variability: standard deviation of normal-to-normal intervals (SDNN) and root mean square of successive differences (RMSSD). These targets are chosen because they are widely used in wearable devices (Natarajan et al., 2020) and are indicative of health status (Shaffer & Ginsberg, 2017), training load in athletes (Plews et al., 2013) and stress levels (Kim et al., 2018).*”.
> >
> > [1] A. Natarajan, A. Pantelopoulos, H. Emir-Farinas, and P. Natarajan, “Heart rate variability with photoplethysmography in 8 million individuals: a cross-sectional study,” The Lancet Digital Health, vol. 2, no. 12, pp. e650–e657, Dec. 2020, doi: 10.1016/S2589-7500(20)30246-6.
> >
> > [2] F. Shaffer and J. P. Ginsberg, “An Overview of Heart Rate Variability Metrics and Norms,” Front. Public Health, vol. 5, p. 258, Sep. 2017, doi: 10.3389/fpubh.2017.00258.
> >
> > [3] D. J. Plews, P. B. Laursen, J. Stanley, A. E. Kilding, and M. Buchheit, “Training Adaptation and Heart Rate Variability in Elite Endurance Athletes: Opening the Door to Effective Monitoring,” Sports Med, vol. 43, no. 9, pp. 773–781, Sep. 2013, doi: 10.1007/s40279-013-0071-8.
> >
> > [4] H.-G. Kim, E.-J. Cheon, D.-S. Bai, Y. H. Lee, and B.-H. Koo, “Stress and Heart Rate Variability: A Meta-Analysis and Review of the Literature,” Psychiatry Investig, vol. 15, no. 3, pp. 235–245, Mar. 2018, doi: 10.30773/pi.2017.08.17.

---

> > > ### Author Response · Authors · 2024-11-20
> > > **Response to [R-hLgx] (Part 3/n)**
> > >
> > > > Re Q8
> > >
> > > We thank the reviewer for their suggestion and raising this important point. We would like to apologize if our previous Section 5.1 did not convey the message properly.
> > > * We agree with the reviewer that “Accel-random” did not provide additional insight and **have now removed “Accel-random” from the previous Table 1,  previous Appendix Table 9 and previous Figure 2 (now Table 1, Appendix Table 8, and Appendix Figure 4) according to the reviewer’s feedback.**
> > > * We would like to respectfully emphasize that “random guessing” was meant to provide the chance performance (not baseline) so it would be clear how much above chance performance the other numbers are. **Therefore, we kept that row and changed to “Chance-level performance” for more transparency.**
> > > * We believe it is important for us to emphasize that the reason we had provided “Accel-MAE” comparison with Procrustes transformation (which learns the optimal rotation, shifting, and scaling that makes one embedding as close as possible to another embedding) was to rule out that the near perfect alignment of accelerometer and PPG after distillation was simply achievable with uni-modal encoders (again not baseline, but just to provide context). **All in all, we also want to clarify that, we had performed the alignment analysis in an unsupervised manner, to demonstrate that we can match accelerometer embeddings to their correct PPG embedding pair with 99.2% top-1 accuracy, therefore, our main purpose was to showcase this absolute performance (99.2% top-1 accuracy for retrieval), as opposed to drawing comparisons; although, we provided some extra numbers to contextualize this absolute performance.** The reason for the importance of this absolute alignment performance is that if the distilled accelerometer embeddings are aligned with the PPG embeddings, it means we can expect improved performance for distilled accelerometer embeddings (as the low-fidelity modality) for any generic downstream target due to alignment to PPG (as the high-fidelity modality).
> > > * **We have now modified section 5.1 to better convey all these points, updated Table 1 and Appendix Table 8 for more clarification and transparency, and here we quote part of it**: “*To rule out that this near perfect retrieval performance was simply achievable with uni-modal encoder embeddings, we repeated the retrieval analysis using the uni-modal accelerometer embeddings from “Accel-MAE” and applying optimal translation, rotation and scaling via Procrustes alignment (Krzanowski, 2000) to make them as close as possible to "PPG-MAE'“ embeddings, and observed marked difference in the retrieval performance (Table 1 and Appendix Table 8). Overall, very high retrieval performance (e.g., 99.17 top-1 accuracy) demonstrates the effectiveness of our representational knowledge distillation framework and how well the distilled accelerometer encoder embeddings match with PPG teacher encoder embeddings. Importantly, these results indicate that distilled accelerometer embeddings may achieve improved performance for predicting downstream targets due to their high alignment with the high-fidelity PPG embeddings, which we will investigate in the next two sections.*”
> > > * **We would also like to refer the reviewer to our new 49 downstream targets (see the summary on top).**

---

> > > > ### Comment · Reviewer_hLgx · 2024-11-22
> > > >
> > > > I have carefully reviewed your response and the changes made to the manuscript. I would like to thank you for addressing most of my concerns and incorporating the suggested modifications into the paper. The revisions have significantly improved the clarity, rigor, and overall quality of the work.
> > > >
> > > > While I still hold the view that the method itself is not highly novel, I recognize that its application to the domain of wearable biosignals is unique and impactful. The thorough investigation and practical implications presented in the manuscript outweigh the limitations in novelty, and I believe that this work will benefit the community. Taking all aspects into consideration, I have decided to increase my score.

---

> > > > > ### Author Response · Authors · 2024-11-22
> > > > > **Response to [R-hLgx] (Part 4/n)**
> > > > >
> > > > > Dear reviewer,
> > > > >
> > > > > We are grateful for your initial constructive feedback, your reassessment of our manuscript, and your kind words about "*clarity, rigor, and overall quality of the work*" and the importance of the application that "*this work will benefit the community*". We have attempted our best to cite proper references for any component of our method or similar related work, however, if there are any additional references you would like us to cover or any claims you would like us to revise, please let us know and we will be happy to address to fairly communicate our message. In the meantime, we will make sure to proactively further modify potentially strong claims that we might have missed, if any, for the camera ready version.
> > > > >
> > > > > Thanks!

---

### Official Review · Reviewer_AiqZ · 2024-11-06

**Soundness:** 3
**Presentation:** 3
**Contribution:** 3
**Rating:** 8
**Confidence:** 4

**Summary:**

This paper presents a novel approach to learning representations from wearable sensor data by combining accelerometer and PPG signals. The authors propose a two-stage training process: first, a student model learns to predict heart rate from accelerometer data, guided by a teacher model that uses PPG signals; second, a shared embedding is learned from both modalities. This approach shows promising results on heart rate and heart rate variability estimation tasks.

**Strengths:**

- The proposed method cleverly leverages two modalities in a cooperative way. Using accelerometer data to guide the learning of heart rate estimation from less reliable acceleration data is a novel and practical approach.

- The authors demonstrate the effectiveness of their method on benchmark datasets, achieving competitive performance on HR and HRV estimation tasks.

- The paper clearly articulates the motivation behind the proposed approach and provides a comprehensive analysis of its benefits.

**Weaknesses:**

-  While the authors acknowledge some related work, a more in-depth comparison to models that directly use acceleration as input and heart rate as output is missing. This would help to better understand the specific advantages of the proposed distillation-based approach.

- The largest model evaluated has 6.3M parameters, which might be insufficient to fully leverage the large dataset. Exploring the impact of scaling the model further could reveal additional performance gains.

- The evaluation focuses solely on HR and HRV estimation. Exploring a wider range of downstream tasks, such as sleep stage classification, activity recognition, or stress detection, would provide a more comprehensive assessment of the learned representations.

**Questions:**

- How does the proposed method compare to models that directly map acceleration to heart rate without any distillation, such as those presented in Hallgrímsson et al. (2018), Spathis et al. (2021), and Ni et al (2019)?

-- Hallgrímsson, Haraldur T., et al. "Learning individualized cardiovascular responses from large-scale wearable sensors data." arXiv preprint arXiv:1812.01696 (2018).

-- Spathis, Dimitris, et al. "Self-supervised transfer learning of physiological representations from free-living wearable data." Proceedings of the Conference on Health, Inference, and Learning. 2021.

-- Ni, Jianmo, Larry Muhlstein, and Julian McAuley. "Modeling heart rate and activity data for personalized fitness recommendation." The World Wide Web Conference. 2019.

- What is the impact of scaling the model size on the performance of both the student and the teacher models? Did you observe any evidence of underfitting with the current model sizes?

- Why did you choose to focus only on HR and HRV estimation tasks? Have you considered evaluating the learned representations on other downstream tasks, such as those explored in Abbaspourazad et al. and Spathis et al., to demonstrate the generalizability of the learned embeddings?

- How does the choice of different teacher network architectures affect the student's performance and the quality of the learned embeddings?

---

> ### Author Response · Authors · 2024-11-20
> **Response to [R-AiqZ] (Part 1/n)**
>
> > Summary
>
> We appreciate the reviewer’s feedback, their kind words about the strengths of our work, and their careful examination of our manuscript. We have responded to the reviewer’s comments below (**W**: **W**eakness, **Q**: **Q**uestion), and have addressed them accordingly in the uploaded revision of the paper. We believe the reviewer’s comments have positively improved our manuscript. **We are looking forward to the reviewer’s revised assessment of our manuscript given the improvements suggested by all the reviewers, which we summarized in the "Summary of revision" on top.**
>
>
> > Regarding W1 & Q1
>
> We would like to thank the reviewer about this comment and pointing out these references. **Our original submission already included various competitive baselines for accelerometer-based prediction methods**: We originally had detailed comparisons across various available labeled data regimes compared to 1) self-supervised accelerometer encoders trained with contrastive learning and masked auto encoding, with two different architectures, 2) supervised accelerometer encoders, which are in line with the reviewer’s comment about “models that directly use acceleration as input and heart rate as output”, as they cover a diverse set of ways to predict downstream targets from accelerometer with unsupervised (1) and supervised methods (2).
>
> Regarding the mentioned references, the general theme across the references [1-3] is modeling the dynamical relationship between gross body motion and the user's heart rate. Activity metrics from these prior works include steps, speed, sleep, or accelerometer statistics at a very slow sampling rate (0.067Hz in [2]) which capture very slow changes in activity/movement. The modeling is done using sparse inputs (one sample every few seconds) over longer time scales (typically up to a few hours) and can include periods of high motion. While such models can effectively learn a measure of cardiovascular activity and fitness, the heart rate estimates are just approximations since the modeling inputs do not directly contain any (heart) beat-level information. In contrast, our paper learns a representation of the accelerometer signal over a short time scale of 60s using the raw signals at 64 Hz during periods of low-motion and at rest. **This allows the model to examine the minute fluctuation of the body which are a direct result of the heart function. This approach is commonly referred to as Ballistocardiography [4, 5], and an example of the associated low-amplitude features can be observed in the periodic fluctuations shown in the sample of accelerometer signals depicted in Figure 1.** Since this captures signatures of individual heart beats, we expect the heart rate estimates during the downstream evaluation to be of much higher quality compared to [1-3], albeit only limited to low motion scenarios. **We have now added these references to the Introduction and Discussion section of the manuscript**:
> * Introduction: "*Also, our work focuses on accelerometer during low-motion periods where it captures minute cardiovascular signals such as the ballistocardiogram, which is distinct from modeling slower changes in accelerometer during gross motion for health and fitness (Hallgrimsson et al., 2018; Ni et al.,2019; Spathis et al.,2021).*"
> * Discussion: “*Our work primarily focused on accelerometer signals during low-motion and sedentary periods (Section 4.1), where accelerometer captures ballistocardiogram and therefore minute cardiovascular-related information (Inan et al., 2015; Kim et al., 2016). Future work can investigate models that take slower-scale activity metrics on wearable devices such as steps, speed, sleep, and slow changes in accelerometer, as well as minute changes in accelerometer at sedentary settings to improve the performance for downstream targets (Hallgrimsson et al., 2018; Spathis et al., 2021; Ni et al., 2019).*”.
>
> [1] Hallgrímsson, Haraldur T., et al. "Learning individualized cardiovascular responses from large-scale wearable sensors data." arXiv preprint arXiv:1812.01696 (2018).
>
> [2] Spathis, Dimitris, et al. "Self-supervised transfer learning of physiological representations from free-living wearable data." Proceedings of the Conference on Health, Inference, and Learning. 2021.
>
> [3] Ni, Jianmo, Larry Muhlstein, and Julian McAuley. "Modeling heart rate and activity data for personalized fitness recommendation." The World Wide Web Conference. 2019.
>
> [4] H.-G. Kim, E.-J. Cheon, D.-S. Bai, Y. H. Lee, and B.-H. Koo, “Stress and Heart Rate Variability: A Meta-Analysis and Review of the Literature,” Psychiatry Investig, vol. 15, no. 3, pp. 235–245, Mar. 2018, doi: 10.30773/pi.2017.08.17.
>
> [5] Inan, O. T., Migeotte, P. F., Park, K. S., Etemadi, M., Tavakolian, K., Casanella, R., ... & Di Rienzo, M. (2014). Ballistocardiography and seismocardiography: A review of recent advances. IEEE journal of biomedical and health informatics, 19(4), 1414-1427.

---

> > ### Author Response · Authors · 2024-11-20
> > **Response to [R-AiqZ] (Part 2/n)**
> >
> > > Re W2 & Q2
> >
> > We would like to thank the reviewer for pointing this out. Estimating scaling laws for the pre-training is definitely an avenue for further improvements and investigations. We would like to clarify that: “*We made several optimizations to keep our model sizes small for feasibility on running wearable devices with power and resource constraints. Interestingly, we observed signs of overfitting as we increased our encoder size, which is why our encoder sizes are not larger. This could be due to the fact that one needs to scale the model and data size simultaneously to gain benefits of scaling laws (Kaplan et al., 2020; Zhai et al., 2022). As an example, when we increased the encoder size in “PPG-MAE” (from 6.3M to 12.7M) by increasing the number of layers from 8 to 16, we observed initial signs of overfitting as shown in Table 14. We believe future work can investigate the scaling laws for encoder models of wearable biosignals by growing the encoder and data size simultaneously (Kaplan et al., 2020; Zhai et al., 2022)*”. **We have now added this new evidence to the new Appendix Table 16 and its corresponding text in Appendix A3.**
> >
> > [1]: Kaplan, J., McCandlish, S., Henighan, T., Brown, T. B., Chess, B., Child, R., ... & Amodei, D. (2020). Scaling laws for neural language models. arXiv preprint arXiv:2001.08361.
> >
> > [2]: Zhai, X., Kolesnikov, A., Houlsby, N., & Beyer, L. (2022). Scaling vision transformers. In Proceedings of the IEEE/CVF conference on computer vision and pattern recognition (pp. 12104-12113).

---

> > > ### Author Response · Authors · 2024-11-20
> > > **Response to [R-AiqZ] (Part 3/n)**
> > >
> > > > Re W3 & Q3 (part 1)
> > >
> > > We would like to thank the reviewer for raising this important point. Here we clarify that:
> > >
> > > * **We originally used HR/HRV as example downstream targets because of their significance in wearable devices due to their information about an individual’s of health status, training loads and stress level, and because wearable devices typically predict HR/HRV frequently throughout the day**: We originally used HR/HRV as our downstream targets due to the importance of predicting them frequently using wearable devices throughout the day and in various environments, which matched well with our motivation of creating powerful encoders for low-fidelity but easy-to-collect biosignals such as accelerometer that are available on all wearables (see our Introduction). These targets, particularly HRV (SDNN/RMSSD) are widely used in wearable devices [1], and are known to be indicative of health status [2], training load in athletes [3] and stress levels [4]. **We have now clarified further significance of HR/HRV in the manuscript**: “*We perform linear probing for predicting heart rate (HR), and two popular measures of heart rate variability: standard deviation of normal-to-normal intervals (SDNN) and root mean square of successive differences (RMSSD). These targets are chosen because they are widely used in wearable devices (Natarajan et al., 2020) and are indicative of health status (Shaffer & Ginsberg, 2017), training load in athletes (Plews et al., 2013) and stress levels (Kim et al., 2018).*”.
> > > * **Our alignment analysis was agnostic to downstream targets and we observed near perfect alignment between accelerometer and PPG embeddings, i.e., our accelerometer encoder can capture PPG-like embeddings irrespective of the downstream target**: In addition, we had performed an alignment analysis in an unsupervised manner, demonstrating that we can match accelerometer embeddings to their correct PPG embedding pair with 99.2% top-1 accuracy. This analysis is agnostic to the downstream target and holistically estimates how well these embeddings are aligned irrespective of the downstream target. If the distilled accelerometer embeddings are aligned with the PPG embeddings, it means we can expect improved performance for distilled accelerometer embeddings (as the low-fidelity modality) for any generic downstream target due to alignment to PPG (as the high-fidelity modality).
> > >
> > > [1] A. Natarajan, A. Pantelopoulos, H. Emir-Farinas, and P. Natarajan, “Heart rate variability with photoplethysmography in 8 million individuals: a cross-sectional study,” The Lancet Digital Health, vol. 2, no. 12, pp. e650–e657, Dec. 2020, doi: 10.1016/S2589-7500(20)30246-6.
> > >
> > > [2] F. Shaffer and J. P. Ginsberg, “An Overview of Heart Rate Variability Metrics and Norms,” Front. Public Health, vol. 5, p. 258, Sep. 2017, doi: 10.3389/fpubh.2017.00258.
> > >
> > > [3] D. J. Plews, P. B. Laursen, J. Stanley, A. E. Kilding, and M. Buchheit, “Training Adaptation and Heart Rate Variability in Elite Endurance Athletes: Opening the Door to Effective Monitoring,” Sports Med, vol. 43, no. 9, pp. 773–781, Sep. 2013, doi: 10.1007/s40279-013-0071-8.
> > >
> > > [4] H.-G. Kim, E.-J. Cheon, D.-S. Bai, Y. H. Lee, and B.-H. Koo, “Stress and Heart Rate Variability: A Meta-Analysis and Review of the Literature,” Psychiatry Investig, vol. 15, no. 3, pp. 235–245, Mar. 2018, doi: 10.30773/pi.2017.08.17.

---

> > > > ### Author Response · Authors · 2024-11-20
> > > > **Response to [R-AiqZ] (Part 4/n)**
> > > >
> > > > > Re W3 & Q3 (part 2)
> > > >
> > > > * **Most importantly, to mitigate the concerns regarding the downstream targets, we have added 49 new downstream targets: age, body mass index (BMI) and biological sex prediction as well as 46 health targets derived from AHMS survey questions**: We totally agree with the reviewer that additional downstream evaluations provide more context about the information in the embeddings and shed more light on their generalizability. **To this end, we added various new downstream evaluations including age, biological sex, body mass index (BMI), and a wide array of health targets collected from surveys in Apple Heart and Movement Study (AHMS) containing targets such as health conditions, use of medications, and lifestyle habits, to provide more context (as suggested by the reviewer’s reference).** These targets evaluate different aspects in the embeddings, e.g., morphological information (as opposed to pulse timing information that may be enough for heart rate and heart rate variability), just as an example, it is well-known that PPG morphological information changes with age [5]. We observed that our distilled accelerometer encoder is significantly better predictive of demographic variables (age, BMI, and biological sex) and health targets compared to the baseline uni-modal accelerometer encoders, indicating indicating its generalizability. **In addition, to the best of our knowledge, this is the first work demonstrating that a single accelerometer encoder can predict demographic variables with remarkably high accuracy, e.g., 4.04 years of error for age, 0.99 classification ROC AUC for sex, 2.46 Kg/m2 error for BMI, and is readily predictive of a variety of health targets. We have now added these results to the paper in new Table 2, new Appendix Table 12, new Results Section 5.3, and new supporting Appendix Tables 20, 21, 22, and 23.**
> > > >
> > > > [5]: Charlton, P. H., Paliakaitė, B., Pilt, K., Bachler, M., Zanelli, S., Kulin, D., ... & Marozas, V. (2022). Assessing hemodynamics from the photoplethysmogram to gain insights into vascular age: a review from VascAgeNet. American Journal of
> > > > Physiology-Heart and Circulatory Physiology, 322(4), H493-H522.

---

> > > > > ### Author Response · Authors · 2024-11-20
> > > > > **Response to [R-AiqZ] (part 5/n)**
> > > > >
> > > > > > Re Q4
> > > > >
> > > > >
> > > > > This is a good point. **We would like to emphasize that we had already done this ablation in the original manuscript** in the original Figure 2, where, as an experiment to change the student/teacher architecture and the pre-training of the teacher, we had redone all the analyses with an EfficientNet architecture where the teacher pre-training was done with contrastive learning (look at “PPG-CL”, “Accel-CL”, “Accel-KD via PPG-CL” where all models are with EfficientNets). We observed similar qualitative improvements across our downstream targets from uni-modal “Accel-CL” to distilled “Accel-KD via PPG-CL” in original Figure 2. **As part of the revision, we have now also repeated the contrastive learning ablations with Transformer, and showed that there are meaningful differences for models trained with contrastive learning when we change the architecture and all our prior conclusions remain the same. We have added this new result to Appendix Table 13.** Note that we now have the combinations of (Transformer, MAE & MAE $\rightarrow$ KD), (Transformer, CL & CL $\rightarrow$ KD), (EfficientNet, CL & CL $\rightarrow$ KD) in the paper. The 4th combination (EfficientNet, MAE) is not practically doable because masked autoencoding, as known in [1], is specific to Transformer architecture due to MAE’s components such as masking and dropping.
> > > > >
> > > > > [1]: He, K., Chen, X., Xie, S., Li, Y., Dollár, P., & Girshick, R. (2022). Masked autoencoders are scalable vision learners. In Proceedings of the IEEE/CVF conference on computer vision and pattern recognition (pp. 16000-16009).

---

> > > > > > ### Comment · Reviewer_AiqZ · 2024-11-26
> > > > > > **Response to authors**
> > > > > >
> > > > > > I want to thank the authors for the thorough response and detailed updates to the paper. I am quite happy with the current state of the work, thanks for addressing all my comments.

---

> > > > > > > ### Author Response · Authors · 2024-11-26
> > > > > > > **Response to [R-AiqZ] (part 6/n)**
> > > > > > >
> > > > > > > Dear reviewer,
> > > > > > >
> > > > > > > We are genuinely grateful for your constructive feedback, your kind words about the strengths of our work, and acknowledging that we have addressed all your comments in our revised manuscript.
> > > > > > >
> > > > > > > Thanks!

---

### Author Response · Authors · 2024-11-20
**Summary of revision (Part 1/n)**

We would like to thank the reviewers for supporting our manuscript and their thoughtful feedback. Overall the reviewers agreed with the strengths of our work with “*This paper presents a novel approach to learning representations from wearable sensor data by combining accelerometer and PPG signals*” [R-AiqZ], “*The paper addresses a significant real-world problem with direct applications in wearable health technology.*” [R-hLgx], “*The motivation is well framed with a good amount of reference to previous works*” [R-GDg8], “*Although cross-modality knowledge distillation has been studied before, applying this onto biosignals is a novel and interesting problem.*” [R-3EkY] & “*The plots showing how distilling ppg encoder information into the accelerometry encoder improves performance is compelling, especially in regards to label efficiency compared to the supervised model*” [R-3EkY], and we are grateful for their positive comments.

We also thank the reviewers for their constructive comments. We have now incorporated all the reviewers’ suggestions and made several important changes and additions. We believe the improvements have significantly strengthened our message, addressed the reviewers's concerns, and improved the quality of our manuscript, thanks to the reviewers' feedback. We have also uploaded a revised version of our manuscript incorporating these changes, where **all the *main* additions are shown in green**; **for the new tables, we changed only the caption to green for better readability** (Table 2, and Appendix Tables 12, 13, 14, 15, 16, 20, 21, 22, 23). We have a point by point response to the reviewers and here we provide summary of all changes:

* **Added 49 new downstream evaluation targets including age, body mass index, biological sex, health conditions, use of medications and lifestyle habits**: In response to the reviewers’ concerns about the generalizability of our evaluations to other downstream targets and the potential concerns regarding the quality of heart rate and heart rate variability targets, we have now added 49 new evaluation targets encompassing a wide array of health related targets including age, body mass index (BMI), biological sex, and 46 other health targets including health conditions, use of medications and lifestyle habits from the questions of Apple Heart & Movement Study. We would like to emphasize that we now have: 1) alignment analysis demonstrating near perfect alignment of PPG and accelerometer embeddings, 2) prediction of heart rate and heart rate variability with detailed label efficiency analysis, 3) as well as all these new 49 targets including age, BMI and biological sex, and health conditions. We believe these evaluations are sufficient to provide evidence about the efficacy of our application: Distillation of information from PPG to accelerometer. **In addition, to the best of our knowledge, this is the first work demonstrating that a single accelerometer encoder can predict demographic variables and health targets with such high accuracy, e.g., 4.04 years of error for age, 0.99 classification of sex, 2.46 $\text{kg}/\text{m}^2$ error for BMI. We have now added these results to the paper in new Table 2, new Appendix Table 12, new Results Section 5.3, and new supporting Appendix Tables 20, 21, 22, and 23.**
* **Added an ablation experiment for using Transformer models when the pre-training is with contrastive learning**: In response to the reviewers’ concerns for simultaneously changing the architecture and teacher pre-training in one of our ablations, we now repeated the analysis for 1) PPG teacher pre-training with contrastive learning, 2) uni-modal accelerometer pre-training with contrastive learning, 3) distilling the PPG teacher encoder from number 1 to an accelerometer student encoder, where the encoders are Transformers. **We showed that contrastive learning can be done with Transformer architecture similar to the EfficientNet without meaningful differences and all conclusions in our manuscript remain unchanged. We have now added this experiment to the new Appendix Table 13.**

* **Ablation study for individual augmentation functions**: While augmentations have been studied before for uni-modal contrastive learning of biosignals, their impact on knowledge distillation across biosignals remains unknown. **To provide a more detailed experiment, we have now studied the effect of individual augmentation functions during knowledge distillation and added the results to the new Appendix Table 17.**

---

> ### Author Response · Authors · 2024-11-20
> **Summary of revision (Part 2/n)**
>
> * **Modified the language about our contributions**: There were few  concerns about the language used in the manuscript for our contributions. **To more precisely state our main contribution, we have now softened the language used in our abstract, contribution list and related work in Introduction**, for example we list our contribution as: “*While multi-modal modeling and cross-modal reconstruction of biosignals have been done before, we study representational knowledge distillation across wearable biosignals using a dataset at such large scale with 20M minutes of multi-modal sensor data from ∼172K participants.*”.
>
> * **Provided additional evidence and ablations**: We also provided additional evidence about:
>     * Root mean squared error and Pearson’s R metrics for heart rate and heart rate variability predictions in **new Appendix Tables 9, 10, and 11**, and showed that our conclusions were unchanged.
>     * Our positive pair selection in contrastive learning in **new Appendix Table 14**.
>     * Why we did not scale our model sizes beyond 6.3M parameters in **new Appendix Table 16**.
>     * The effect of number of PPG channels in teacher pre-training in **new Appendix Table 15**.
> * **Modified text and additional references**: We also modified and added text in various sections of the manuscript, and provided additional references, as suggested by reviewers throughout the manuscript.
> * **Modified the organization of the manuscript**: we slightly modified the organization of the manuscript to fit the manuscript within the 10-page limit while incorporating the new changes.
>
>
> While the above are the changes we made to the manuscript, we would like to clarify few points regarding concerns raised about novelty and comparisons:
>
> * While a couple of reviewers claim that: “*The baseline comparison is also extremely limited. Out of three comparisons, two of them are random guessing and random weight models.*” [R-GDg8] and “*The paper lacks comparisons with state-of-the-art (SOTA) cross-modal distillation methods, as well as accelerometer-based prediction methods*” [R-9r1K], **we respectfully want to mention that our original submission already had comparisons versus**: 1) self-supervised accelerometer encoders trained with contrastive learning and masked auto encoding, with two different architectures of Transformer and EfficientNet (now even more evidence in Figure 2, Table 2, Appendix Tables 9, 10, 11, 12, 13), 2) supervised accelerometer encoders (now even more evidence in Figure 2, Appendix Tables 9, 10, 11), 3) simultaneous multi-modal contrastive learning of PPG and accelerometer (as an alternative multi-modal method, Table 3). **This is while we never claimed that the message of our paper was proposing a novel method but rather the application of transferring knowledge from PPG to accelerometer and  we used our 2-stage distillation method (Figure 1) as a means to achieve this.**
> * We agree that the individual modeling components used in our paper are not themselves novel in the deep learning domain, and we never claimed we created new individual components (e.g. masked autoencoding, contrastive learning, knowledge distillation, knowledge distillation across pre-trained models) — in fact, even in our original manuscript we said: “*We combine and adopt techniques inspired by uni-modal and multi-modal pre-training frameworks from other domains of deep learning to create a fully unsupervised distillation framework for biosignal time-series, which is particularly crucial for health applications where labeled data is limited*”. **That said, we respectfully believe figuring out how best to combine and adopt existing techniques in order to solve real-world problems is extremely non-trivial, particularly in such large-scale datasets and novel applications for accelerometer signals. We are not aware of any prior work that does the modeling similar to how we do for wearable biosignals.**
> * We believe our technical implementation, solution to the problem and empirical findings are valuable to researchers and engineers in research and industry labs working with data from wearable devices. Practical applications of how to combine such techniques are non-trivial, and we would like to respectfully state that according to ICLR reviewer guidelines (https://iclr.cc/Conferences/2025/ReviewerGuide), “Submissions bring value to the ICLR community when they convincingly demonstrate new, relevant, impactful knowledge (incl., empirical, theoretical, for practitioners, etc).”, and **we believe our manuscript does fit well in this category, which is why we originally submitted our manuscript to the “Applications” track.**

---

> ### Author Response · Authors · 2024-11-20
> **Summary of revision (Part 3/n)**
>
> * While three reviewers are very positive about novelty and impact of our approach and findings: “*This paper presents a novel approach to learning representations from wearable sensor data*” [R-AiqZ], “*The paper addresses a significant real-world problem with direct applications in wearable health technology.*” [R-hLgx] and “*Although cross-modality knowledge distillation has been studied before, applying this onto biosignals is a novel and interesting problem.*” [R-3EkY], the other 2 reviewers raise concerns about novelty, where only one of them [R-GDg8] explicitly mentions a related work [1] in terms of these concerns for the novelty. However, we respectfully believe **our methodology**, **our application**, **our dataset**, and most importantly **our empirical findings** significantly differ from [1] (**and any prior work**), and **our manuscript provides transformative evidence of using accelerometer for diverse health applications/diagnoses to the wearable research community.**
>     * **Methodology**: Methodology used in our work is distinct from [1]. We investigate transferring representational knowledge (compact embeddings) from an already trained embedding encoder for PPG to another embedding encoder for accelerometer, which falls under knowledge distillation [2, 3]. This is while the prior work investigates cross-modal reconstruction of ECG from PPG in the output domain with a cycleGAN.  In our original submission’s title/abstract/text, we have frequently emphasized that our work is “representational knowledge distillation”, which is a narrow variation of knowledge distillation referring to transferring embeddings/representations from a network to another network [2], and we disagree that [1] fits in this category, nor this prior work has investigated creating **generalist/foundation** embedding models like we have. Also, in details, almost all training and modeling choices significantly differ between the two works.
>     * **Application**: We have repeatedly mentioned in our original submission that our goal is to build models for low-cost power-efficient biosignals that are available in any wearable without requiring any optical sensors, which is why we used accelerometer. **This is a transformative application/finding for wearable research and industry to use accelerometer for a wide array of health/diagnosis applications.** The mentioned paper by the reviewer [1] (**and any prior work**), does not show this, and in fact we disagree with the reviewer that [1] has a “low-fidelity” biosignal and we believe PPG is high-fidelity and predictive of a wide array of health targets, in many cases better than ECG [4].
>     * **Dataset**:  Prior work [1] uses 4 public datasets that in total have **125** participants. **Our dataset includes 172,318 participants with the data collected under natural daily life conditions covering 4 years of data. This is 1378x larger in terms of number of participants.** We hope that the reviewers, like we do, appreciate the challenges of modeling large-scale datasets and the generalizability of the findings in our manuscript to a broad range of demographics, which is important for health applications.
>     * **Findings**: In our revised manuscript, we show that **a single distilled accelerometer encoder is readily (without fine-tuning) predictive of heart rate, heart rate variability, age, body mass index (BMI), biological sex, as well as 46 other health related targets spanning targets for underlying health conditions, the use of medications and lifestyle habits.** This is all done by only linear probing of the embeddings of single accelerometer encoder, which means a single compact embedding vector is predictive of all these targets. **We respectfully believe, this is significantly different than the mentioned work by the reviewer which only looks at heart rate by calculating it directly from the reconstructed ECG.** In addition, we have done several ablation studies for sweeping the number of labels, compressing the model sizes, different architecture, teacher pre-training framework, and etc. These are all distinct from prior work.
>
> [1] Sarkar, Pritam, and Ali Etemad. "Cardiogan: Attentive generative adversarial network with dual discriminators for synthesis of ecg from ppg." Proceedings of the AAAI Conference on Artificial Intelligence. Vol. 35. No. 1. 2021.
>
> [2] Hinton, G. (2015). Distilling the Knowledge in a Neural Network. arXiv preprint arXiv:1503.02531.
>
> [3] Tian, Y., Krishnan, D., & Isola, P. (2019). Contrastive representation distillation. arXiv preprint arXiv:1910.10699.
>
> [4] Abbaspourazad, S., Elachqar, O., Miller, A. C., Emrani, S., Nallasamy, U., & Shapiro, I. (2023). Large-scale training of foundation models for wearable biosignals. arXiv preprint arXiv:2312.05409.

---

### Comment · Area_Chair_sDZD · 2024-11-21
**Rebuttal**

Dear Reviewers,

I encourage you to review the rebuttal and reach out to the authors with any additional questions or requests for clarification.

Best,\
AC

---

### Meta-Review · Area_Chair_sDZD · 2024-12-20

**Metareview:**

The paper proposes a framework to distil information from PPG to accelerometer data. They then demonstrate that this enables accelerometers to predict heart rate, heart rate variability, demographics, and health outcomes. The paper uses a very large-scale dataset, the Apple Heart and Movement Study, for both training and testing. The paper has several notable strengths as follow. It applies knowledge distillation in an innovative manner to leverage accelerometers for physiological and health monitoring - this has never been done before. It is easy to understand and well-presented. The results are strong in both breadth and depth. The key weakness which has resulted in my recommendation is its explicit reliance on a private dataset without additional evaluation on publicly available datasets, limiting reproducibility and generalization. Please see below for a detailed account of the discussions and my reasons for rejection.

**Additional Comments On Reviewer Discussion:**

The paper has received highly diverse scores: 3, 3, 6, 8, 8.  Reviewers AiqZ, hLgx, and 3EkY strongly supported the paper given its interesting proposition, strong implications for the field, and breadth of empirical downstream results, all of which I agree with. Reviewers GDg8 and 9r1K, however, were critical of the work, citing lack of baselines, possible dataset bias, and limited novelty. I respectfully disagree with these points. I believe that while the baselines could have indeed been more diverse, this is not a critical issue. I also don’t see any likely issues with bias in their dataset (although I cannot confirm this since it is not public). Regarding the reviewer comments that the data have been selected from less-noisy and stationary data –this is common practice for any real-world product and in-the-wild setting. So I don’t seen an issue with this either. I also don’t see the novelty as a concern – while the novelty does appear incremental from a methodological standpoint, wholistically evaluating the paper, the concept/problem-statement itself is novel enough.

My only concern with the paper, which aligns with the critical reviewers, is reproducibility given the ‘explicit’ use of a private dataset (no public dataset is used) and no release of code or model weights. What added to my worries was the vague reproducibility statement in the paper that states:\
"*The aggregated data that support the findings of this study can be made available on request from the corresponding author. Request for data will be evaluated and responded to in a manner consistent with the specific language in the study protocol and informed consent form (ICF). Similarly, code for all data analyses may be available upon request from the corresponding author. Requests for code will be evaluated and responded to in a manner consistent with policies intended to protect participant confidentiality and language in the study protocol and ICF.*"

This leaves it unclear as to whether the actual dataset or the code or the model can be shared or not, leaving reproducibility of the work highly questionable. I personally admire this work and believe the paper can have positive and interesting implications for our field. But upon considering all the angles, I very sadly realize that **without any avenue for reproducibility**, I cannot, in good conscience, recommend acceptance.

To ensure reproducibility, the paper could either: (1) perform everything in two separate/parallel experiment settings: one where training and testing are done on private data, and one where training and testing are done on public data; or (2) only use private data for training, but then instead release the model code and weights, plus also test on a few (even though small) public datasets. Either of these two paths would enable reproducibility of the work in my opinion. But the current approach uses a private dataset for training and only presents results on private data, with no public release of the code or weights. *This makes it impossible to even attempt to replicate the work given no clear way to move toward the direction of the provided results.* Given that ICLR is an ML conference, there is a heavy emphasis on reproducibility. Based on this, unfortunately I cannot recommend acceptance. I understand that the internal policies of the author’s organization or acquired consent forms may not allow them to release the code, weights, or data. This is where public datasets would need to come in as a parallel experiment setting. Alternatively, another way to address this issue would be to re-cast the paper as a study with a research question in the lines of “Can health metrics such as X which are generally extracted from PPG, also be extracted from Accelerometer data?”. In this approach, the ML piece would simply be seen as a tool and not the main contribution of the work, similar to many other health-related studies whose “tools” are far less important than the findings – but then the paper would need more analysis, discussions, and statistical validation of the findings.

---

### Decision · Program_Chairs · 2025-01-22

Reject